# The chromatin factors SET-26 and HCF-1 oppose the histone deacetylase HDA-1 in longevity and gene regulation in *C. elegans*

Felicity J. Emerson [1], Caitlin Chiu [1], Laura Y. Lin [1], Christian G. Riedel [2], Ming Zhu[3] & Siu Sylvia Lee [1] ✉

SET-26, HCF-1, and HDA-1 are highly conserved chromatin factors with key roles in development and aging. Here we present mechanistic insights into how these factors regulate gene expression and modulate longevity in *C. elegans*. We show that SET-26 and HCF-1 cooperate to regulate a common set of genes, and both antagonize the histone deacetylase HDA-1 to limit longevity. HCF-1 localization at chromatin is largely dependent on functional SET-26, whereas SET-26 is only minorly affected by loss of HCF-1, suggesting that SET-26 could recruit HCF-1 to chromatin. HDA-1 opposes SET-26 and HCF-1 on the regulation of a subset of their common target genes and in longevity. Our findings suggest that SET-26, HCF-1, and HDA-1 comprise a mechanism to fine-tune gene expression and longevity and likely have important implications for the mechanistic understanding of how these factors function in diverse organisms, particularly in aging biology.

Chromatin factors provide a layer of regulatory information for the genome by influencing the surrounding chromatin environment. They impact gene expression by modulating histone marks, DNA methylation, accessibility of chromatin, and the composition of protein complexes at chromatin. As such, chromatin factors have an essential role in biology and have been implicated in diverse biological processes from cell division[1] to aging[2]. Of particular interest, chromatin and aging are intricately linked. Many studies show correlations between aging and an altered chromatin environment (see ref. 3 for review), but causative studies are challenging in humans and mammalian model organisms, which often have a relatively long natural lifespan. One of the best systems with which to study this relationship is the worm *C. elegans*, which shares many critical chromatin regulators with mammals[4] and is a robust model for aging due to its short lifespan and well-characterized aging pathways.

In *C. elegans*, manipulation of specific histone readers, writers, erasers, adapters, and chromatin remodelers has been shown to influence longevity[5–14], supporting a causative link between chromatin and aging. Chromatin factors that influence longevity have been demonstrated to work through many well-studied longevity pathways, including induction of the mitochondrial unfolded protein response (mitoUPR)[9,11,13,14], lipid metabolism[12], and insulin signaling[6,7,10]. Understanding the precise mechanism by which chromatin factors act to modulate aging will help us better understand the aging process itself and may guide efforts to improve healthy aging.

SET-26 and HCF-1 are two highly conserved chromatin factors that our lab has studied that influence longevity in *C. elegans*. SET-26, the *C. elegans* homolog of MLL5 and SETD5, is an epigenetic reader that binds to H3K4me3, a mark typically associated with active chromatin, via its PHD domain[15]. HCF-1, the *C. elegans* homolog of HCF-1, is a chromatin adapter protein, well characterized in mammalian cells for recruiting histone-modifying complexes to chromatin[16]. Loss of either SET-26 or HCF-1 leads to a long lifespan and increased stress resistance[5,8,15,17]. Evidence indicates that SET-26 and HCF-1 operate within somatic cells to limit lifespan[5,15], and they function within the germline to regulate germline development and fertility[15,18]. Both SET-26 and HCF-1 require DAF-16 to modulate lifespan[5,8,15,17], and HCF-1 also interacts with SIR-2.1 to affect lifespan[17]. Whereas SET-26 and HCF-1 share many similarities in how they impact longevity, their working relationship was not previously investigated.

[1]Department of Molecular Biology and Genetics, Cornell University, Ithaca, NY, USA. [2]Department of Biosciences and Nutrition, Karolinska Institutet, Huddinge, Sweden. [3]National Institute of Biological Sciences, Beijing, China. ✉e-mail: sylvia.lee@cornell.edu

Chromatin factors lacking enzymatic activity themselves, like SET-26 and HCF-1, often partner with additional enzymatic components that directly alter the chromatin environment[16]. Histone deacetylases are a major group of enzymes that remove histone acetylation, a mark typically associated with active chromatin[19]. The histone deacetylase HDA-1, the *C. elegans* homolog of HDAC1 and HDAC2, is a critical regulator of development, neurodegeneration, and the mitoUPR[13,14,20,21], a stress response program that communicates mitochondrial stress to the nucleus and initiates transcriptional activation of mitochondrial chaperones and proteases[22]. HDA-1 is required for the long lifespan of a model of mitoUPR activation[13], but its role in the lifespan of other longevity models has not yet been explored.

Here we show that SET-26 and HCF-1 are functional partners in longevity and chromatin regulation in *C. elegans*. We find that SET-26 and HCF-1 operate in the same genetic pathway to regulate lifespan, they share many common binding sites at chromatin, and they influence somatic gene expression in similar ways. Our data show that SET-26 is largely required for HCF-1 binding to chromatin in *C. elegans* somatic cells, whereas HCF-1 is dispensable for most, but not all, SET-26 binding to chromatin. We therefore hypothesize that SET-26 recruits HCF-1 to chromatin, and, at a subset of binding sites, we hypothesize that HCF-1 plays a minor role in stabilizing SET-26 binding as well. We find that the long lifespan of both the *set-26(-)* and *hcf-1(-)* mutants depends on HDA-1, and we posit that all three factors co-regulate a common set of genes, likely those involved in mitochondrial and lipid metabolism, with HDA-1 antagonizing SET-26 and HCF-1 for control over gene expression.

## Results

### SET-26 and HCF-1 work together to modulate lifespan

Previous work has demonstrated that loss of either SET-26[8,15] or HCF-1[5,17] leads to a longer lifespan in *C. elegans*, however the mechanism of lifespan extension for either of these mutants is incompletely understood, and functional cooperative partners for these proteins have not been identified. Analysis of HCF-1 binding partners in *C. elegans* through two independent immunoprecipitation-mass spectrometry (IP-Mass Spec)[17] experiments identified SET-26 as a high-confidence interactor (Supplementary Data 1), suggesting the two proteins could be part of the same complex. Consistent with our IP-Mass Spec observation, MLL5 and HCF-1, the human homologs of SET-26 and HCF-1, are able to physically interact and form a complex via an "HCF-1 binding motif" present in the MLL5 protein[23], which is well conserved from worms to humans (Supplementary Fig. 1a). Interestingly, SET-26 has two human homologs, MLL5 and SETD5, which both contain similar SET domains as SET-26[24,25]. MLL5 appears to be more similar to SET-26 in its ability to bind H3K4me3 via its PHD domain[26]. SETD5, which lacks a PHD domain[25], shares a similar degree of sequence similarity to SET-26 and has also been recently reported to interact with human HCF-1 in hematopoietic stem cells[27], indicating that both human homologs of SET-26 can interact with human HCF-1.

We reasoned that if SET-26 and HCF-1 are functional partners, they should operate in the same genetic pathway to modulate longevity. We tested the *set-26(tm2467) hcf-1(pk924)* double mutant and observed that loss of *set-26* did not significantly further extend the long lifespan of the *hcf-1(-)* mutant (Fig. 1a). This non-additive effect is consistent with the hypothesis that the two genes operate in the same genetic pathway to modulate lifespan. However, as it is possible that a non-additive phenotype could be observed in the double mutant due to alternative explanations, such as two separate longevity pathways being fully activated and unable to extend lifespan further, we pursued additional experiments to characterize the actions of SET-26 and HCF-1 directly at chromatin.

### SET-26 and HCF-1 bind common gene targets

Next, we wondered whether SET-26 and HCF-1 occupy similar binding sites at chromatin, as would be expected if they are part of the same protein complex. To test this, we first constructed CRISPR knock-in strains containing tagged SET-26 or HCF-1. Notably, the tagged strains for both SET-26 and HCF-1 had normal lifespans (Supplementary Fig. 1b, c), indicating that the tags did not disrupt the normal protein function.

We then performed the chromatin profiling technique, CUT&RUN (Cleavage under targets and release using nuclease) with tagged SET-26 or HCF-1. Since our previous work indicated that SET-26 and HCF-1 act in somatic cells to modulate lifespan[5,15], we specifically profiled SET-26 and HCF-1 genomic binding in somatic cells using a germline-less mutant background. This approach was especially critical because SET-26 and HCF-1 both have separate functions in the germline to regulate development and reproduction[15,18], and due to the large size of the *C. elegans* germline[28], utilizing only wildtype worms would risk diluting any somatic-specific signal that could be the most relevant for longevity. The CUT&RUN data indicated that the biological replicates were well correlated with each other (Supplementary Fig. 1d), indicating the results were highly reproducible. As previously shown, the SET-26 binding profile obtained by CUT&RUN was highly similar to the previous ChIP-seq profile for SET-26 obtained from our lab[15,29]. SET-26 and HCF-1 peaks were both highly overlapping with promoter regions in the *C. elegans* genome (Supplementary Fig. 1e, f), with high enrichment immediately upstream of the transcription start site (TSS) (Fig. 1b, c). This is consistent with previous ChIP-seq data of HCF-1 from mammalian cells[30,31], as well as our lab's previous SET-26 ChIP-seq data in *C. elegans*, which indicated that SET-26 is an H3K4me3 reader and binds to many active promoters containing H3K4me3[15].

The somatic binding profiles of SET-26 and HCF-1 were highly similar (Fig. 1d–f; Supplementary Data 2), with 76% of HCF-1 peaks overlapping with SET-26 peaks and 55% of SET-26 peaks overlapping with HCF-1 peaks. Moreover, we found that within these common peak regions, SET-26 and HCF-1 binding were centered directly on top of each other (Fig. 1g). Importantly, we found that SET-26 and HCF-1 profiles were highly overlapping even after excluding blacklisted regions[32] and highly occupied target (HOT) regions[33] (Supplementary Fig. 1g; Supplementary Data 2), which are regions commonly identified as peaks across many genome-wide binding profiles for transcription factors[33,34]. Interestingly, we found that SET-26 and HCF-1 bind to thousands of regions in the genome (with 13,422 and 9961 peaks called, respectively), indicating that the factors are highly ubiquitous across the genome. As expected for factors enriched preceding TSSs, SET-26 and HCF-1 profiles largely overlapped with active chromatin, including regions marked by H3K4me3[35] and accessible regions identified through ATAC-seq[36] (Supplementary Fig. 1h; Supplementary Data 2) which were identified at similar ages in germline-less worms. This indicates that SET-26 and HCF-1 are both highly prevalent factors at active chromatin, and likely also interact with many additional factors that also bind to these sites.

We next associated SET-26 and HCF-1 peaks to their closest promoters[37] to identify the putative target genes regulated by SET-26 and HCF-1. As expected, the majority of their putative target genes were shared (Fig. 1h; Supplementary Data 2). Functional enrichment analysis indicated that the 8824 SET-26 and HCF-1 commonly-bound genes were enriched for many biological processes, with the most highly significant being mitochondrial metabolism (253 genes), mRNA processing (214 genes), and ribosome subunits (82 genes) (Fig. 1i; Supplementary Data 2). Taken together, these results suggest that SET-26 and HCF-1 both bind to the promoters of many genes important for basic biological functions in worms.

To confirm that the main findings from this analysis also extended to wild-type worms, we performed the same CUT&RUN analysis in the wildtype background and obtained similar results (Supplementary Fig. 1i–l; Supplementary Data 2). Interestingly, while the majority of genes bound by SET-26 and HCF-1 were identified in both wildtype and somatic-specific analysis (Supplementary Fig. 1m, n; Supplementary

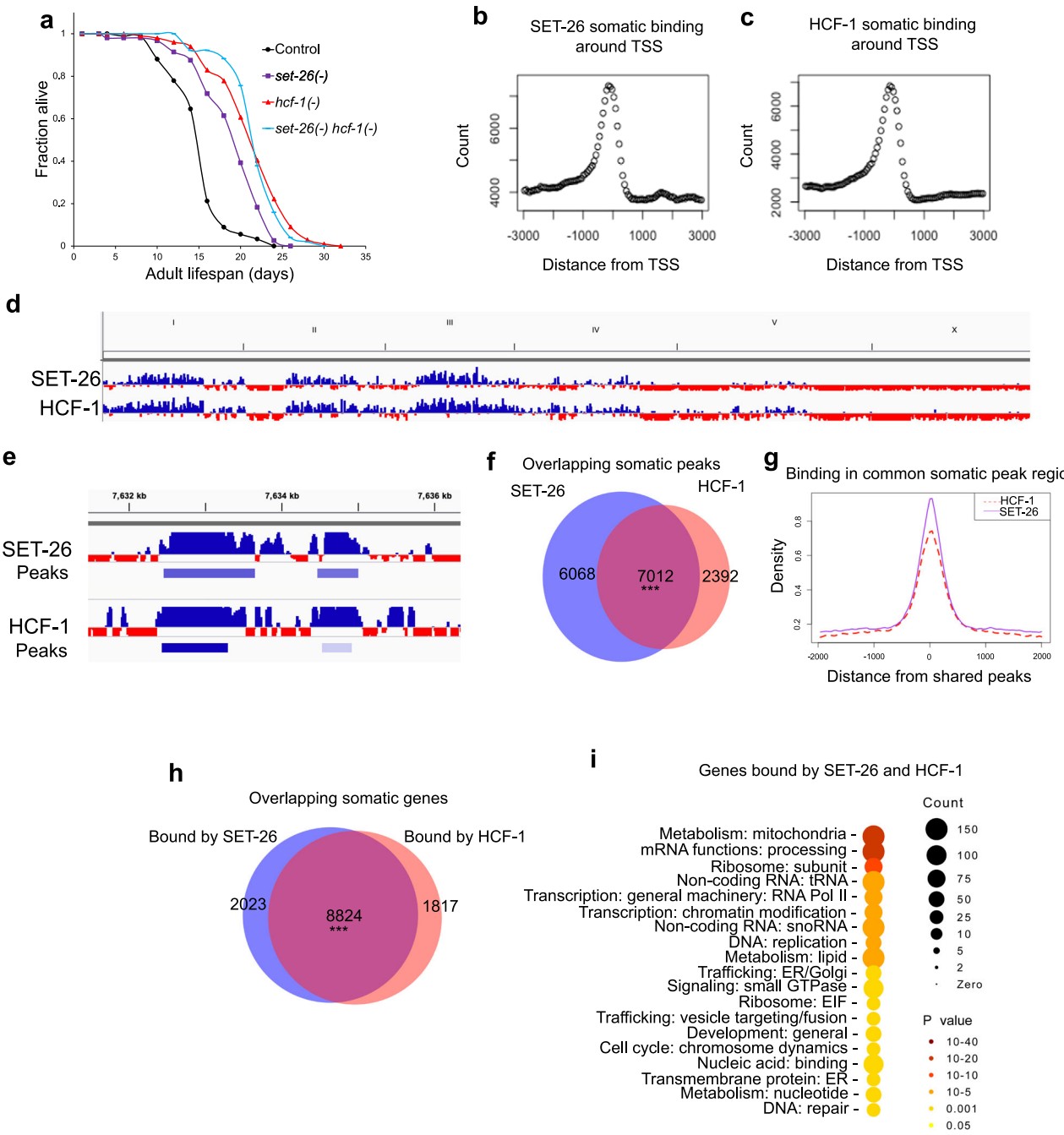

**Fig. 1 | SET-26 and HCF-1 operate in the same genetic pathway to modulate lifespan and have overlapping binding profiles at chromatin. a** Survival curves for wildtype controls, *set-26(-)*, *hcf-1(-)*, and *set-26(-) hcf-1(-)* double mutants from one representative experiment ($n = 101, 103, 105$, and 106 worms, respectively). Source data are provided as a Source Data file. $N = 2$ biological replicates. Distribution of somatic (**b**) SET-26 or (**c**) HCF-1 peaks relative to the transcription start sites (TSS) of associated genes within 3 kb. Count represents the number of somatic peaks within each bin. Data were obtained from CUT&RUN assays ($N = 2$) in either *glp-1(-); set-26::ha* or *glp-1(-);hcf-1::gfp::3xflag* tagged strains. d Screenshot from the Integrated Genomics Viewer (IGV) showing a genome-wide view of somatic SET-26 or HCF-1 binding. Binding is displayed as normalized log2 signal, where blue represents enrichment, and red represents depletion, of factor binding relative to control. **e** This shows the same binding data as in (**d**) but for a portion of Chromosome IV. Binding is displayed on the top track, and peaks called by MACS2 are displayed

on the bottom track, where the darker blue represents more statistically significant peaks as determined by MACS2. **f** Venn diagram showing somatic SET-26 and HCF-1 peaks in the *glp-1(-)* mutant background, and 7012 peaks showing an overlap of 1 bp or more. **g** Metaplots of normalized SET-26 and HCF-1 binding signals within the 7012 overlapping peak regions identified in (**f**). **h** Venn diagram showing the number of genes associated with somatic SET-26 or HCF-1 binding, and 8824 genes that are commonly bound by both factors. **i** Gene ontology (GO) term analysis for the 8824 genes bound by both SET-26 and HCF-1 in somatic cells as determined by Wormcat. In (**f, h**) *** indicates $p < 1 \times 10^{-15}$ and the overlap is higher than expected by chance, as calculated by one-sided hypergeometric test for peak overlap (in **f**) and one-sided Fisher's Test for gene overlap (in **h**). In (**i**), Wormcat $p$ values are determined by one-sided Fisher test with FDR correction. Peaks and gene sets are provided in Supplementary Data 2.

Data 2), the genes identified as bound by SET-26 and HCF-1 only in wildtype worms containing germlines were enriched for additional functional groups, including cell cycle (12 and 14 genes, respectively) (Supplementary Fig. 1o, p; Supplementary Data 2), consistent with important roles for SET-26 and HCF-1 in the cell cycle as has been demonstrated in mammalian cells[16,23]. Whole worm SET-26 and HCF-1 binding was also enriched for transcription: chromatin structure (48 and 39 genes), small nuclear RNAs (20 and 44 genes), and mRNA binding (16 and 26 genes).

**SET-26 and HCF-1 regulate expression of a common set of genes**
To better understand how SET-26 and HCF-1 regulate the genes to which they bind, we performed RNA-seq of germline-less *set-26(-)* and *hcf-1(-)* mutants at the same time point as our CUT&RUN experiments (day 1 adulthood) (Supplementary Fig. 2a). Differential expression analysis revealed highly overlapping RNA expression changes in *set-26(-)* and *hcf-1(-)* mutants (Fig. 2a–c; Supplementary Data 3), suggesting similar pathways are activated and repressed in both longevity mutants. Pathway enrichment analysis indicates that the 485 genes commonly upregulated in germline-less *set-26(-)* and *hcf-1(-)* mutants are most enriched for mitochondrial metabolism (55 genes, median fold change 3.60 and 2.69), and collagen/extracellular material (31 genes, median fold change 30.59 and 118.53) (Supplementary Fig. 2b; Supplementary Data 3), while the 602 commonly downregulated

genes are most significantly enriched for pathogen stress response (41 genes, median fold change 0.31 and 0.24) and lipid metabolism (56 genes, median fold change 0.31 and 0.31) (Supplementary Fig. 2c; Supplementary Data 3).

We focused on the putative direct targets of SET-26 and HCF-1 that are likely to have biological relevance under our assaying condition, which are the genes bound by each of the factors that also showed expression changes when each of the factors was lost. Notably, similar numbers of SET-26 and HCF-1 direct targets showed up- (206 genes) or downregulated (239 genes) mRNA expression in the mutants (Fig. 2d; Supplementary Data 3), suggesting that SET-26 and HCF-1 could have both activating and repressive roles depending on the gene. The SET-26 and HCF-1 targets commonly upregulated in *set-26(-)* and *hcf-1(-)* mutants were enriched for mitochondrial metabolism (46 genes, median fold change 3.60 and 2.57), ribosome biogenesis (12 genes, median fold change 2.53 and 2.98), and mRNA binding (7 genes, median fold change 2.21 and 3.07) (Fig. 2e; Supplementary Data 3), whereas the downregulated common targets were enriched for lipid metabolism (23 genes, median fold change 0.38 and 0.40), short chain dehydrogenase metabolism (5 genes, median fold change 0.44 and 0.46), and pathogen stress response (9 genes, median fold change 0.37 and 0.38) (Fig. 2f; Supplementary Data 3), suggesting SET-26 and HCF-1 could have direct roles in regulating these pathways.

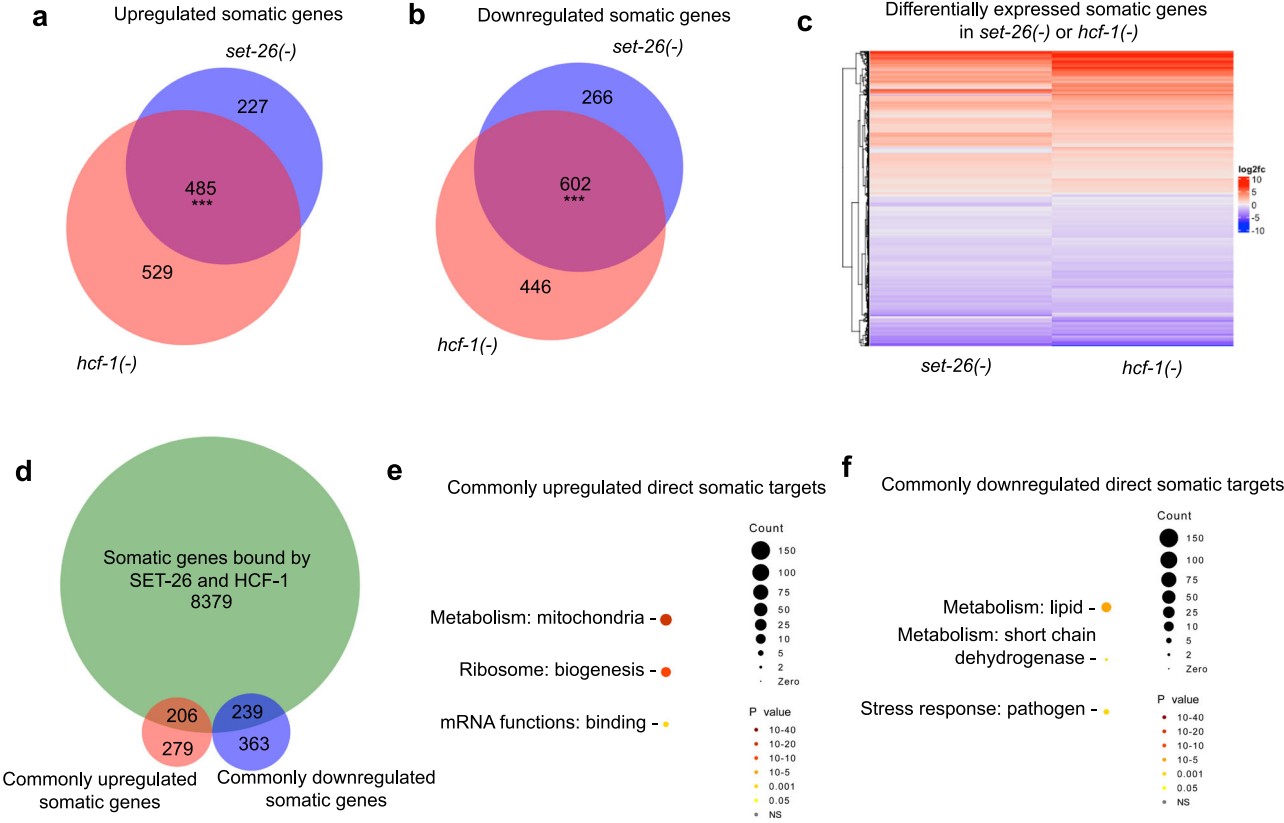

**Fig. 2 | SET-26 and HCF-1 regulate a common set of genes.** Venn diagrams showing genes (**a**) upregulated or (**b**) downregulated in RNA expression in germline-less day 1 adult *glp-1(-);set-26(-)* or *glp-1(-);hcf-1(-)* double mutants compared to *glp-1(-)* single mutants and the overlapping 485 upregulated, or 602 downregulated genes with significant expression changes in both mutants. **c** Heatmap showing hierarchical clustering of all differentially expressed genes in either *glp-1(-);set-26(-)* or *glp-1(-);hcf-1(-)* double mutants compared to *glp-1(-)* single mutants; The log2 fold change of RNA expression in each double mutant versus control was used for the clustering analysis and is shown as indicated on the scale. **d** Venn diagram showing the genes bound by SET-26 and HCF-1 as determined in

Fig. 1h and the genes commonly upregulated or downregulated in RNA expression when *set-26* or *hcf-1* is inactivated as determined in (**a**, **b**). 445 genes are bound by both SET-26 and HCF-1 and show significant RNA expression change in both mutants. GO term analysis by Wormcat of SET-26 and HCF-1 direct somatic targets that are (**e**) upregulated (206) or (**f**) downregulated (239) in RNA expression in (**d**). In (**a**) and (**b**), *** indicates $p < 1 \times 10^{-15}$ and the overlap is higher than expected by chance, as calculated by one-sided Fisher's Test. In (**e**) and (**f**), Wormcat *p* values are determined by one-sided Fisher test with FDR correction. Gene sets are provided in Supplementary Data 3. *N* = 2 for RNA-seq.

## SET-26 is required for HCF-1's recruitment to chromatin at the majority of HCF-1 binding sites

We next wondered what relationship SET-26 and HCF-1 have with each other at chromatin. SET-26 and its' mammalian homolog, MLL5, have both been previously described to bind to the active H3K4me3 histone mark[15,26], which could account for their method of recruitment to chromatin. HCF-1 however, does not possess any DNA or chromatin binding domains itself, and the mammalian homolog is often recruited by other chromatin factors, including MLL5[16,38], to large protein complexes. To determine whether SET-26 could recruit HCF-1 to chromatin in somatic cells of *C. elegans*, we obtained the chromatin binding profile of HCF-1 using our HCF-1-tagged strain in germline-less worms with and without functional *set-26*. In order to appropriately compare HCF-1 binding profiles between genotypes, we additionally surveyed H3 binding in each genotype in parallel and normalized HCF-1 binding to H3 within each genotype. By surveying H3 in parallel within each genotype, we were able to use H3 as a type of internal control for the immunoprecipitation process itself within each genotype. We found

that normalizing to antibody background as in Fig. 1 or to H3 made little difference in the appearance of the HCF-1 profile or the location of peaks called by MACS2 (Supplementary Fig. 3a), and the H3 profiles were highly consistent between controls and *set-26(-)* mutants (Supplementary Fig. 3b). We therefore used H3 as normalization for further differential analysis.

A survey of the HCF-1 peak regions revealed a consistent and dramatic decrease in HCF-1 binding in somatic cells of the *set-26(-)* mutant (Fig. 3a), such that the number of HCF-1 somatic peaks was reduced by 88% in the *set-26(-)* background (Fig. 3b; Supplementary Data 4). To visualize this genome-wide, we plotted HCF-1 binding at all HCF-1 somatic peak regions and noticed a drastic decrease in binding in the *set-26(-)* mutant at most HCF-1 peaks (Fig. 3c). While there is still some enrichment of HCF-1 at binding sites in the *set-26(-)* mutant above background levels, we conclude that SET-26 is required for the majority of proper HCF-1 patterning at chromatin genome-wide in somatic cells. *hcf-1* RNA and protein levels are not noticeably changed in the germline-less *set-26(-)* mutant (Supplementary Fig. 3c–e),

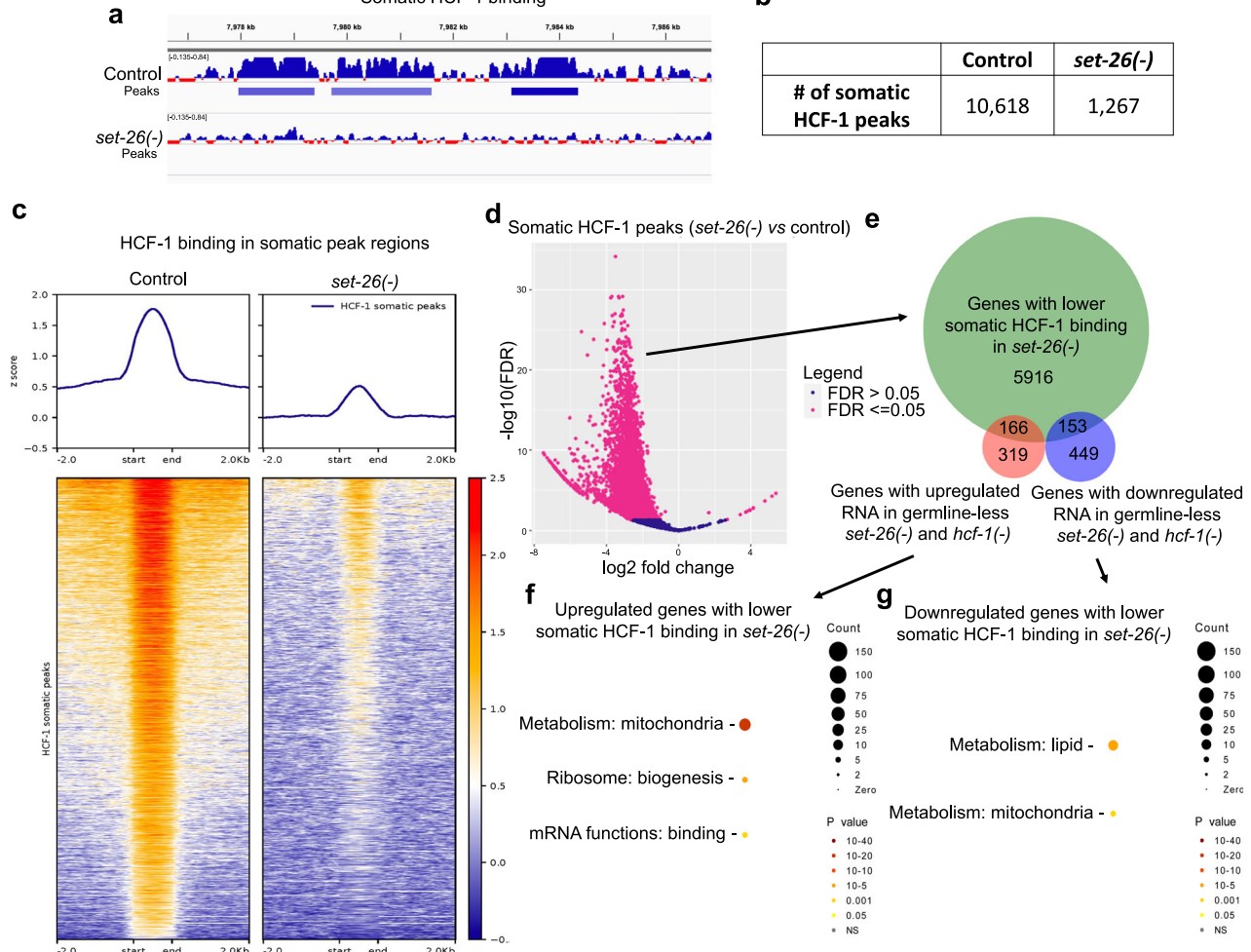

**Fig. 3 | HCF-1 recruitment to chromatin in somatic cells requires SET-26.**
**a** Screenshot from IGV shows normalized somatic HCF-1 binding and peak calls in a portion of Chromosome V (captured by CUT&RUN of *hcf-1::gfp::3xflag* worms, *N* = 2) in control worms or *set-26(−)* mutants grown on *glp-1* RNAi. **b** Number of HCF-1 peaks called in combined replicates of controls or *set-26(−)* mutants grown on *glp-1* RNAi. **c** Metaplot (top) and heatmap (bottom) of z-scores representing normalized HCF-1 signal in somatic HCF-1 binding sites and surrounding 2 kb up- and downstream in either controls or *set-26(−)* mutants grown on *glp-1* RNAi. **d** Volcano plot of HCF-1 binding regions determined by DiffBind to be significantly different (pink, FDR < = 0.05) or unchanged (blue, FDR > 0.05) in *set-26(−)* mutants compared

to controls grown on *glp-1* RNAi. DiffBind FDR values are calculated using DESeq2. **e** Venn diagram of genes with significantly lower HCF-1 binding in *set-26(-)* mutants grown on *glp-1* RNAi (as determined in **d**) and the overlap with genes up- or downregulated in RNA expression in germline-less *set-26(-)* and *hcf-1(-)* mutants (from Fig. 2a, b). Wormcat GO enrichment analysis for genes with lower HCF-1 binding in *set-26(-)* mutants on *glp-1* RNAi that are either (**f**) upregulated or (**g**) downregulated in RNA expression in both *set-26(-)* and *hcf-1(-)* germline-less mutants (from **e**). Wormcat *p* values are determined by one-sided Fisher test with FDR correction. Gene sets and differential peaks are provided in Supplementary Data 3 and 4.

suggesting that the decrease of HCF-1 binding to chromatin is not due to a defect in HCF-1 production in set-26(-) mutants.

We next asked which HCF-1 somatic binding regions were statistically significantly decreased in the set-26(-) mutant compared to controls, and we observed that 82% of HCF-1 peaks exhibited significantly lower binding in the set-26(-) mutant (Fig. 3d; Supplementary Data 4), with a substantial median fold change decrease (0.18). We next asked which of these set-26-dependent HCF-1-bound regions were associated with genes that showed significant RNA expression change in somatic cells of set-26(-) and hcf-1(-) mutants (Fig. 3e; Supplementary Data 3). We found that the expression of 319 of the set-26-dependent HCF-1-bound genes showed significant RNA expression change in set-26(-) and hcf-1(-) mutants, with similar numbers of up- and downregulated genes. The upregulated genes were strongly enriched for mitochondrial metabolism (42 genes, median fold change of 3.65 and 2.60), with less significant enrichment for ribosome biogenesis (9 genes, median fold change of 2.44 and 3.14) and mRNA binding (7 genes, median fold change of 2.21 and 3.07) (Fig. 3f; Supplementary Data 3), while the downregulated genes were enriched for lipid (18 genes, median fold change 0.36 and 0.38) and mitochondrial metabolism (10 genes, median fold change 0.36 and 0.39) (Fig. 3g; Supplementary Data 3). The data suggest that SET-26 could play a critical role in either recruiting or stabilizing HCF-1 binding at chromatin, where the two factors work together to fine-tune gene expression, particularly of mitochondrial and lipid metabolism genes. To confirm that the dramatic germline-less results we obtained were not due to the RNAi system used to produce germline-less worms (glp-1 RNAi), we repeated the experiments with a genetic perturbation to produce germline-less worms (glp-1(e2141)) and obtained similar results (Supplementary Fig. 3f, g), with 80% of HCF-1 peaks being significantly decreased by loss of set-26 (Supplementary Fig. 3h; Supplementary Data 4).

When we repeated these experiments in whole worms containing germlines, we observed a moderate loss of HCF-1 recruitment to chromatin in the set-26(-) mutant worms (Supplementary Fig. 3i, j), with only 9% of HCF-1 peaks being significantly lowered in worms lacking set-26 (Supplementary Fig. 3k; Supplementary Data 4). As SET-26 primarily operates in somatic cells to modulate lifespan[15], we focused on the somatic results, which we believe to be the most relevant to potentially help us understand the lifespan phenotypes of the mutants.

## HCF-1 is dispensable for most SET-26 recruitment but does contribute to SET-26 stabilization at a subset of genes

We next tested the opposite hypothesis, examining whether SET-26 recruitment to chromatin also required HCF-1. We performed CUT&RUN targeting SET-26 in germline-less worms with and without functional hcf-1 (Supplementary Fig. 4a), and we did not observe a dramatic difference in SET-26 recruitment to chromatin (Fig. 4a–c; Supplementary Data 4). We wondered whether individual SET-26 peak regions at local sites may still reach the threshold for being significantly changed in the hcf-1 mutant. We identified 37% of SET-26 somatic peaks were significantly decreased by loss of hcf-1 (Fig. 4d; Supplementary Data 4), with a modest median fold change decrease (0.54). These binding differences were largely unique to somatic cells as well, as, when we repeated the experiments in whole worms containing germlines, only 2% of SET-26 binding regions were identified as significantly altered in hcf-1(-) mutants (Supplementary Fig. 4b–d; Supplementary Data 4). We asked which of the hcf-1-dependent SET-26-bound regions identified in somatic cells were associated with genes that showed significant RNA expression change in somatic cells of set-26(-) and hcf-1(-) mutants (Fig. 4e; Supplementary Data 3). 238 hcf-1-dependent SET-26-bound genes showed significant expression changes in the somatic set-26(-) and hcf-1(-) mutants, again with a similar number of up- and down-regulated genes.

The upregulated genes were enriched for mitochondrial metabolism (38 genes, median fold change 3.72 and 2.71), ribosome biogenesis (8 genes, median fold change 3.23 and 3.54), and mRNA binding (6 genes, median fold change of 2.16 and 3.23) (Fig. 4f; Supplementary Data 3), while the downregulated genes were enriched for lipid metabolism (11 genes, median fold change 0.43 and 0.41) (Fig. 4g; Supplementary Data 3).

Since these genes represent similar biological categories to the genes identified in Fig. 3e–g as having decreased HCF-1 binding in the set-26(-) germline-less mutant and exhibiting expression changes in the mutants, we asked how many of the genes from the two analyses overlapped. We found that the majority of the genes did overlap (Fig. 4h, i; Supplementary Data 3), with 122 upregulated and 79 downregulated genes in common. As expected, the 122 upregulated genes were enriched for mitochondrial metabolism (36 genes, median fold change of 3.72 and 2.71), ribosome biogenesis (6 genes, median fold change 3.23 and 3.54), and mRNA binding (5 genes, median fold change 2.21 and 3.40), (Supplementary Fig. 4e; Supplementary Data 3) while the 79 downregulated genes were enriched for lipid metabolism (8 genes, median fold change of 0.42 and 0.41) (Supplementary Fig. 4f; Supplementary Data 3). Overall, the data suggest that SET-26 and HCF-1 could co-stabilize each other on a subset of direct target genes involved in key biological processes.

## HDA-1 is required for the full longevity of set-26(-) and hcf-1(-) mutants

Because SET-26 and HCF-1 homologs are well known to work with additional chromatin factors[23,30,38,39], we wondered whether we could identify additional protein factors that might work together with SET-26 and HCF-1 at chromatin in C. elegans. We noted that the histone deacetylase HDA-1, homolog of mammalian HDAC1 and HDAC2, was a possible interactor of HCF-1 based on the IP-Mass Spec data (Supplementary Data 1), even though the data suggested only a weak interaction. Homologs of SET-26 and HCF-1 in flies and humans, respectively, have been suggested to recruit HDAC1 homologs to chromatin[16,40], supporting a possible conserved functional interaction between these three proteins in various species.

We next tested whether hda-1 depletion would impact the lifespan of our set-26(-) and hcf-1(-) mutants. Previous studies have found that, depending on conditions, depletion of hda-1 can either reduce[13], or have no effect[41,42] on lifespan in C. elegans. However, the standard method of initiating hda-1 RNAi at egglay in C. elegans leads to a profound sterility and developmental phenotype, resulting in worms with a sterile and disorganized gonad and protruding vulva[21]. To avoid these pleiotropic developmental effects and to focus on the effect of hda-1 depletion during aging, we initiated hda-1 RNAi on day 1 of adulthood in our lifespan studies. We found that under these conditions, depletion of hda-1 with two different RNAi constructs (Supplementary Fig. 5a, b) did not impact the lifespan of wildtype worms, as previously reported for RNAi initiated at the L4 stage[41]. Interestingly, depletion of hda-1 in adulthood specifically decreased the lifespan of set-26(-) and hcf-1(-) mutants (Fig. 5a and Supplementary Fig. 5c). Repeating this experiment in a germline-less mutant background, we obtained similar results (Fig. 5b), suggesting that HDA-1 is required in somatic cells of set-26(-) and hcf-1(-) mutants, where SET-26 and HCF-1 operate to modulate longevity.

## HDA-1 co-occupies promoters bound by SET-26 and HCF-1

Although histone deacetylases are typically associated with gene silencing by removing active acetylation marks, the mammalian HDAC1 has been shown to localize at active gene promoters[19,43]. Recent ChIP-seq data in C. elegans from germline-less mutants also supports the localization of HDA-1 to active promoters in the worm[13]. To repeat this observation in our own hands, we tagged HDA-1 with GFP and HA. We performed CUT&RUN with this HDA-1-tagged strain in germline-

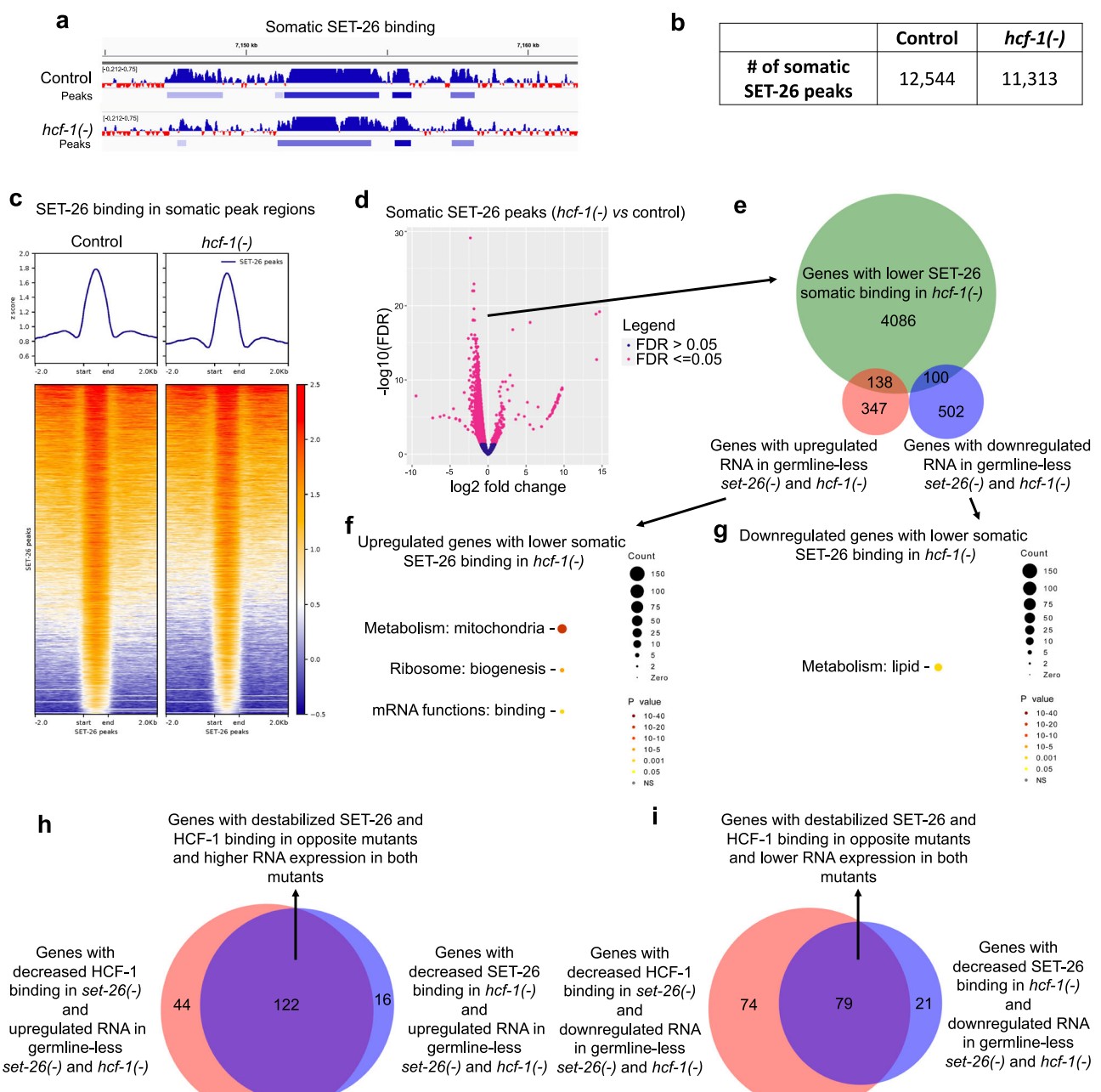

**Fig. 4 | HCF-1 is dispensable for most SET-26 recruitment genome-wide but facilitates SET-26 binding at a subset of somatic binding sites. a** Screenshot from IGV shows normalized somatic SET-26 binding and peak calls in a portion of Chromosome II (captured by CUT&RUN of *set-26::ha* worms, *N* = 2) in controls or *hcf-1(-)* mutants grown on *glp-1* RNAi. **b** Number of SET-26 peaks called in combined replicates of controls or *hcf-1(−)* mutants grown on *glp-1* RNAi. **c** Metaplot (top) and heatmap (bottom) of z-scores representing normalized SET-26 signal in somatic SET-26 binding sites and surrounding 2 kb up- and downstream in either controls or *hcf-1(-)* mutants grown on *glp-1* RNAi. **d** Volcano plot of SET-26 binding regions determined by DiffBind to be significantly different (pink, FDR < = 0.05) or unchanged (blue, FDR > 0.05) in *hcf-1(-)* mutants compared to controls grown on *glp-1* RNAi. DiffBind FDR values are calculated using DESeq2. **e** Venn diagram

showing the overlap of the somatic genes with significantly lower SET-26 binding in *hcf-1(-)* mutants grown on *glp-1* RNAi (as determined in **d**) and those with up- or downregulated RNA expression in germline-less *set-26(-)* and *hcf-1(-)* mutants (as seen in Fig. 2a, b). Wormcat GO enrichment analysis for genes with lower SET-26 binding in *hcf-1(-)* mutants on *glp-1* RNAi that are either (**f**) upregulated or (**g**) downregulated in RNA expression in *set-26(-)* and *hcf-1(-)* mutants (from **e**). Wormcat *p* values are determined by one-sided Fisher test with FDR correction. h-i Venn diagram showing the number of genes with decreased SET-26 or HCF-1 binding in the opposite mutant and the (**h**) 122 that overlap and are upregulated or (**i**) 79 that overlap and are downregulated in RNA expression in both mutants as determined in (**e**) and Fig. 3e. Gene sets and differential peaks are provided in Supplementary Data 3 and 4.

less mutants (Supplementary Fig. 5d) and noticed that HDA-1 peaks were often (81% of the time) overlapping with promoter regions (Supplementary Fig. 5e), and, like SET-26 and HCF-1, the HDA-1 signal was enriched preceding the TSS (Fig. 5c). HDA-1 co-occupied many of the same promoters as SET-26 and HCF-1 (Fig. 5d−f; Supplementary Data 2), with 54% of HDA-1 peaks overlapping both SET-26 and HCF-1

somatic peaks, 49% of SET-26 peaks overlapping both HCF-1 and HDA-1 peaks, and 70% of HCF-1 peaks overlapping both SET-26 and HDA-1 peaks. This suggests that all three factors could regulate a common set of genes. Similar results were obtained by repeating the experiment in whole worms containing a germline (Supplementary Fig. 5f−i; Supplementary Data 2).

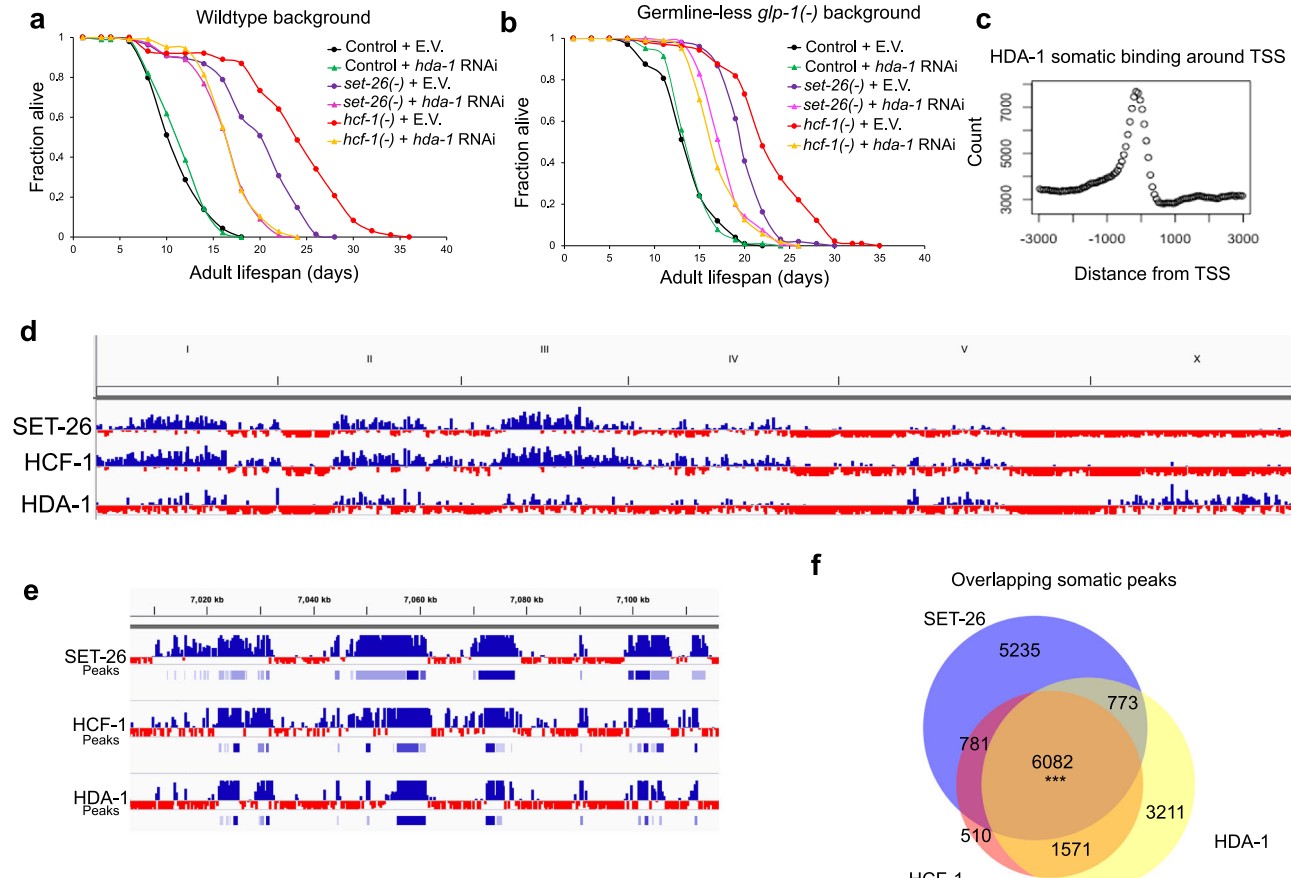

**Fig. 5 | HDA-1 is required for the full longevity of *set-26(-)* and *hcf-1(-)* mutants and co-occupies many binding sites with SET-26 and HCF-1. a** Survival curves for wildtype controls, *set-26(-)*, and *hcf-1(-)* mutants on E.V. control RNAi and *hda-1* RNAi from one representative experiment (*n* = 108, 102, 107, 105, 104, and 106 worms, respectively). **b** Survival curves for *glp-1(-)* germline-less controls, *glp-1(-);set-26(-)*, and *glp-1(-);hcf-1(-)* mutants on E.V. control RNAi and *hda-1* RNAi from one representative experiment (n = 104, 104, 106, 104, 106, and 105 worms, respectively). *hda-1* RNAi was initiated on day 1 of adulthood to avoid developmental defects caused by initiating *hda-1* RNAi from egglay. *N* = 2 biological replicates for lifespan experiments. **c** Distribution of somatic HDA-1 peaks relative to the transcription start sites (TSS) of associated genes within 3 kb. Count represents the number of

somatic peaks within each bin. Data were obtained from CUT&RUN assays in the *glp-1(-);hda-1::gfp::ha* tagged strain, *N* = 2. Screenshots from IGV show normalized somatic SET-26, HCF-1 (as in Fig. 1), and HDA-1 binding either in a (**d**) whole genome view, or a (**e**) close-up of a section of Chromosome II, with peaks called by MACS2 underneath the signal track for each factor. **f** Venn diagram showing somatic SET-26, HCF-1, and HDA-1 peaks in the *glp-1(−)* mutant background and 6082 peaks that overlap by 1 bp or more. In (**f**), *** indicates *p* < 1 × 10⁻¹⁵ and the overlap is higher than expected by chance, as calculated by one-sided hypergeometric test. Source data are provided as a Source Data file. Gene sets are provided in Supplementary Data 2.

## HDA-1 binding to chromatin is not detectably changed in *set-26(-)* or *hcf-1(-)* mutants

We next wondered whether HDA-1 localization to chromatin was dependent on SET-26 or HCF-1. We performed CUT&RUN using germline-less HDA-1-tagged worms in *set-26(-)* and *hcf-1(-)* mutants (Supplementary Fig. 6a). We did not detect major differences in HDA-1 recruitment in *set-26(-)* or *hcf-1(-)* germline-less mutants (Fig. 6a–c; Supplementary Data 4), although the overall somatic HDA-1 signal and number of peaks was somewhat lower, particularly in *set-26(-)* mutants. This was accompanied by the identification of 26% of HDA-1 somatic peaks as being significantly lower in the *set-26(-)* mutant (Fig. 6d; Supplementary Data 4), while only 5% of peaks were lower in the *hcf-1(-)* mutant (Fig. 6e; Supplementary Data 4). When we repeated the experiment in whole worms containing a germline (Supplementary Fig. 6b, c), the effect of *set-26* loss on HDA-1 recruitment to chromatin was more dramatic, with 64% of HDA-1 peaks showing decreased binding (Supplementary Fig. 6d; Supplementary Data 4), whereas loss of *hcf-1* continued to have a minimal effect on HDA-1 recruitment to chromatin (Supplementary Fig. 6e; Supplementary Data 4). The data raise the intriguing possibility that germline SET-26 could have a role in HDA-1 recruitment. However, given that SET-26 operates in somatic

cells to regulate lifespan, our data suggest that any HDA-1 recruitment defect observed here is unlikely to be related to the longevity phenotype of the *set-26(-)* mutant. This notion is further supported by the observation that only a small number of genes with lower somatic HDA-1 binding in *set-26(-)* and *hcf-1(-)* mutants also showed gene expression changes in those mutants (Fig. 6f; Supplementary Data 3). Altogether, we suggest that altering HDA-1 recruitment in early adulthood is not the main mechanism through which *set-26(-)* and *hcf-1(-)* mutants extend lifespan.

## HDA-1 regulates expression of a subset of SET-26 and HCF-1 targets in opposing directions in the soma of *set-26(-)* and *hcf-1(-)* mutants

We next wondered whether the interaction between SET-26, HCF-1, and HDA-1 could be at the level of gene regulation. Given the antagonistic relationship between HDA-1 with SET-26 and HCF-1 in terms of lifespan modulation, we hypothesized that HDA-1 would oppose SET-26 and HCF-1 in gene regulation. We performed RNA-seq of germline-less *set-26(-)* and *hcf-1(-)* mutants and controls grown to adulthood on empty vector control bacteria and aged on either *hda-1* RNAi or empty vector control to determine *hda-1*-dependent

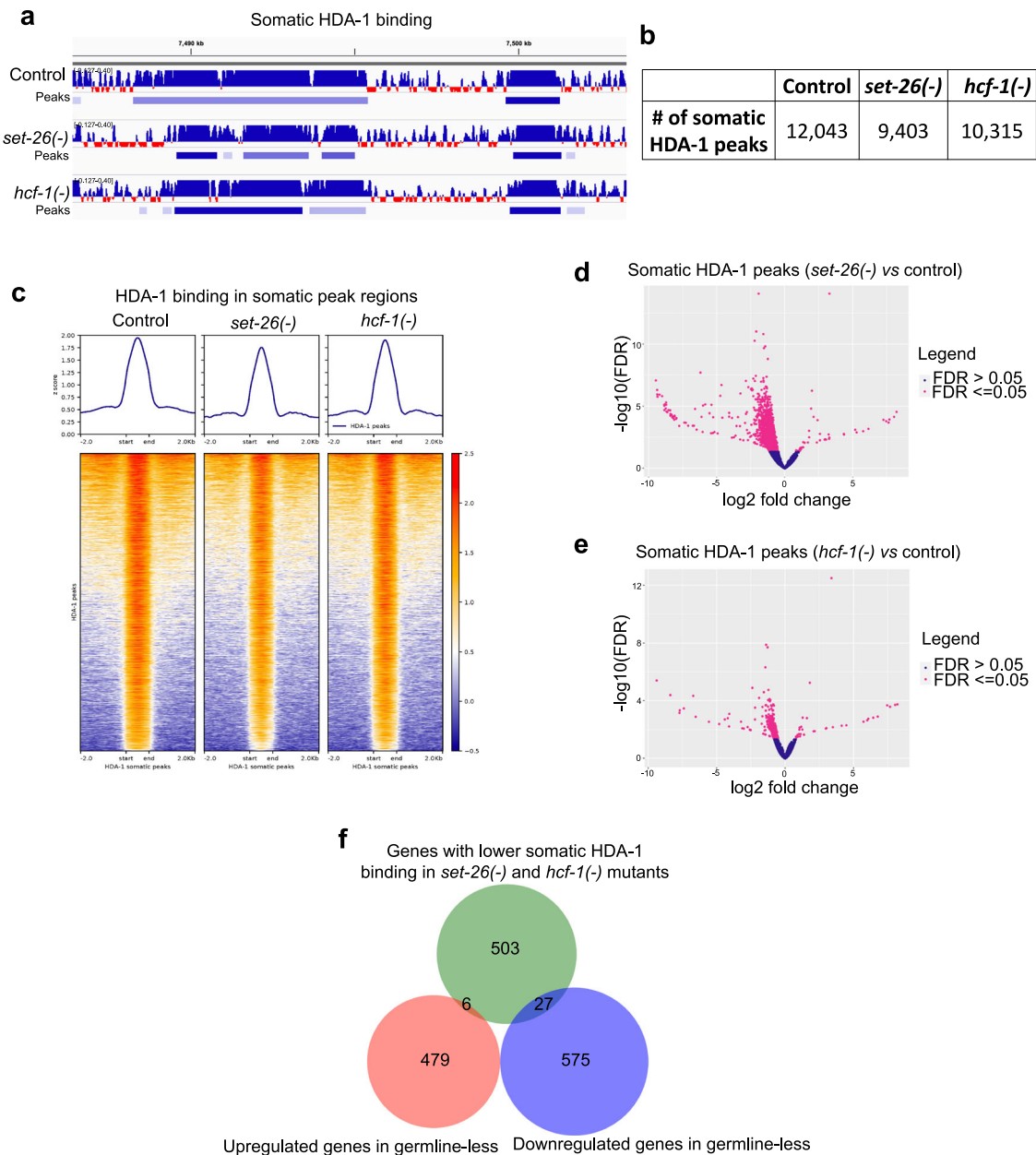

**Fig. 6 | SET-26 and HCF-1 are unlikely to be major players in HDA-1's recruitment to chromatin in somatic cells. a** Screenshot from IGV shows normalized somatic HDA-1 binding and peak calls in a portion of Chromosome I (captured by CUT&RUN of *hda-1::gfp::ha* worms, *N* = 2) in controls, *set-26(-)*, or *hcf-1(-)* mutants grown on *glp-1* RNAi. **b** Number of HDA-1 peaks called in combined replicates of controls, *set-26(-)*, or *hcf-1(-)* mutants grown on *glp-1* RNAi. **c** Metaplot (top) and heatmap (bottom) of z-scores representing normalized HDA-1 signal in somatic HDA-1 binding sites and surrounding 2 kb up- and downstream in either controls, *set-26(-)*, or *hcf-1(-)* mutants grown on *glp-1* RNAi. **d**, **e** Volcano plot of HDA-1 binding

regions determined by DiffBind to be significantly different (pink, FDR < = 0.05) or unchanged (blue, FDR > 0.05) in (**d**) *set-26(-)* or (**e**) *hcf-1(-)* mutants compared to controls grown on *glp-1* RNAi. DiffBind FDR values are calculated using DESeq2. **f** Venn diagram showing the genes with lower somatic HDA-1 binding in *set-26(-)* and *hcf-1(-)* mutants (identified in **d**, **e**) and the 6 and 27 genes that are commonly up- or downregulated, respectively, in *set-26(-)* and *hcf-1(-)* mutants as determined in Fig. 2a, b. Gene sets and differential peaks are provided in Supplementary Data 3 and 4.

gene expression changes. We collected samples at day 3, roughly 48 h after initiating *hda-1* RNAi, and at day 12 of adulthood to examine any age-related changes. We chose day 12 as a late-stage timepoint because it represents a time just before a major wave of death in germline-less controls (Fig. 5b), and because age-related chromatin and RNA expression changes in germline-less *glp-1(-)* mutants have been extensively profiled by our lab at day 12 of adulthood[35,44,45]. As expected, we uncovered many of the same age-related gene expression changes as our lab previously identified[35] (Supplementary Fig. 7a, b).

Surprisingly, we found that gene expression was largely unchanged at day three of adulthood, after two days of *hda-1* RNAi (Supplementary Fig. 7c; Supplementary Data 3). This is despite the marked decrease in HDA-1 protein levels induced at this timepoint, with a decrease of approximately 87% in the wildtype background and 48% in the germline-less background (Supplementary Fig. 7d, e). Although the knockdown in the germline-less background was weaker than the wildtype background when surveying total protein levels in whole worms by western blot, this was likely due to the high proportion of neurons (roughly one third of somatic cells) in the germline-less

mutant, which are resistant to RNAi[46,47]. Indeed, while intestinal cells from germline-less mutants showed a dramatic reduction of HDA-1::GFP::HA levels by day 3 of adulthood after two days of *hda-1* RNAi (Supplementary Fig. 7f, g), the GFP levels in the head were not significantly decreased by *hda-1* RNAi (Supplementary Fig. 7h, i).

By day 12 of adulthood however, *hda-1*-dependent gene expression changes emerged (Supplementary Fig. 7j; Supplementary Data 3), and we therefore focused our analysis on these later-life gene expression changes. We identified 471 and 877 genes that changed in expression in *set-26(-)* and *hcf-1(-)* mutants respectively on *hda-1* RNAi versus empty vector. We found that of the 967 and 1129 genes that are misregulated at baseline in *set-26(-)* and *hcf-1(-)* mutants compared to controls on empty vector at day 12 of adulthood, 114 (12%) and 204 (18%), respectively, were again significantly mis-regulated upon *hda-1* RNAi. Of these genes that are regulated by SET-26 and/or HCF-1 at baseline and are again changed by *hda-1* RNAi, the majority of the genes exhibited an antagonistic change in expression such that the gene expression changes normally observed in the *set-26(-)* and *hcf-1(-)* mutants were reversed when aged on *hda-1* RNAi. Specifically, of the 114 genes that showed significant expression changes in the *set-26(-)* mutant (compared to controls) and also when *set-26(-)* mutants were treated with *hda-1* RNAi (compared to control RNAi), 70% (80 genes) show an antagonistic relationship, to the extent that 50 of these genes that are normally altered in *set-26(-)* mutants no longer reached significance in *set-26(-)* mutants treated with *hda-1* RNAi, and one gene reached significance in the opposite direction from its baseline change. Similarly, of the 204 genes that are differentially expressed in the *hcf-1(-)* mutant (compared to controls) and also when *hcf-1(-)* was treated with *hda-1* RNAi (compared to control RNAi), 68% (138 genes) show an antagonistic relationship, whereupon 83 of these genes were no longer significantly differentially expressed upon *hda-1* RNAi, and two genes reached significance in the opposite direction from their baseline change.

To focus on the gene expression changes most likely to be relevant to the lifespan phenotype, we looked for *hda-1*-dependent gene expression changes unique to *set-26(-)* and *hcf-1(-)* mutants. We reasoned that since *hda-1* RNAi shortens the lifespan of *set-26(-)* and *hcf-1(-)* mutants and not controls, the genes that change in response to *hda-1* RNAi in *set-26(-)* and *hcf-1(-)* mutants and not controls would be most likely to explain the requirement for *hda-1* in *set-26(-)* and *hcf-1(-)* mutant longevity. We identified 813 genes in total that were significantly differentially expressed in either *set-26(-)* or *hcf-1(-)* mutants on *hda-1* RNAi compared to empty vector. We plotted the expression of these genes first at baseline, in *set-26(-)* or *hcf-1(-)* day 12 mutants compared to controls, and then in *set-26(-)* and *hcf-1(-)* mutants on *hda-1* RNAi compared to empty vector (Fig. 7a; Supplementary Data 3).

We noticed that the heatmap naturally clustered into three segments based on different patterns of gene expression behavior, with two of the three clusters exhibiting opposing gene expression in the *set-26(-)* and *hcf-1(-)* mutants at baseline and on *hda-1* RNAi. We were particularly interested in these opposing clusters (cluster 1 with 332 genes and cluster 3 with 281 genes), because of our observation that HDA-1 opposes SET-26 and HCF-1 in lifespan modulation, making genes in these clusters good candidates as modulators of lifespan in the mutants. We found that Cluster 1, which contained genes that tended to increase in *set-26(-)* and *hcf-1(-)* mutants (median fold change 1.25 and 1.40) but decreased on *hda-1* RNAi (median fold change 0.47 and 0.53), was enriched for lipid (26 genes) and mitochondrial metabolism (20 genes), as well as stress response genes (27 genes combined between pathogen, heat, and detoxification categories) and proteolysis genes (5 genes) (Fig. 7b; Supplementary Data 3), while Cluster 3, which contained genes that tended to decrease in *set-26(-)* and *hcf-1(-)* mutants (median fold change 0.70 and 0.59) but increased on *hda-1*

RNAi (median fold change 2.07 and 2.48), was enriched for unassigned processes (110 genes) and collagen (12 genes) (Fig. 7c; Supplementary Data 3).

We were particularly interested in the enrichment in Cluster 1 for mitochondrial metabolism genes, given our findings that SET-26 and HCF-1 stabilize each others' binding on mitochondrial metabolism genes and the expression of these genes are increased in *set-26(-)* and *hcf-1(-)* mutants (Fig. 4h, Supplementary Fig. 4e; Supplementary Data 3). Furthermore, previous studies have demonstrated that HDA-1 is required for activation of the mitoUPR in *C. elegans*[13,14], and the lifespan phenotype of a model of mitoUPR-mediated longevity[13]. Our RNA-seq analysis showed elevated levels of two mitoUPR markers, *hsp-6* (increased 5.46-fold and 3.12-fold) and *hsp-60*[48] (increased 5.82-fold and 2.79-fold), in *set-26(-)* and *hcf-1(-)* young adults (Supplementary Fig. 8a, b), suggesting the mitoUPR is activated at baseline in these mutants. In line with this, we observed increased GFP expression of the classical mitoUPR reporter strain, *hsp-6p::gfp*, in *set-26(-)* and *hcf-1(-)* mutants as young adults (Supplementary Fig. 8c, d). To test if HDA-1 is required for mitoUPR activation in *set-26(-)* and *hcf-1(-)* mutants at baseline, we initiated *hda-1* RNAi at egglay and observed decreased expression of *hsp-6p::gfp* in *set-26(-)* and *hcf-1(-)* mutants as young adults (Supplementary Fig. 8c, d). As a control, we repeated the conditions followed by Shao et al. and observed as expected that *hsp-6p::gfp* activation induced by *atp-2* RNAi was decreased upon *hda-1* RNAi as expected[13] (Supplementary Fig. 8e, f).

Previous analysis from Shao et al. identified a group of 283 "HDA-1-dependent mitoUPR genes", which are mitoUPR genes normally activated upon mitochondrial perturbation and the expression of which decreases on *hda-1* RNAi[13]. We intersected this gene list with our CUT&RUN datasets to obtain 121 "HDA-1-dependent mitoUPR genes" also bound by SET-26, HCF-1, and HDA-1 in somatic cells. We plotted the expression of these 121 genes (Fig. 7d; Supplementary Data 3) in our day 12 adult gene expression data set again at baseline and with *hda-1* RNAi during aging. Although only a small number of these genes (19 and 26) reached the threshold of statistical significance in *set-26(-)* and *hcf-1(-)* mutants versus controls at day 12 of adulthood, respectively, our heatmap revealed that the majority of these 121 genes tended to subtly increase in expression in germline-less *set-26(-)* and *hcf-1(-)* mutants compared to controls, with 94 and 92 out of 121 genes having a fold change higher than one compared to controls in *set-26(-)* and *hcf-1(-)* mutants (median fold change of 1.37 and 1.45, respectively). When exposed to *hda-1* RNAi, the expression of these genes generally decreased in both *set-26(-)* and *hcf-1(-)* mutants, with 102 and 109 genes showing a fold change less than one on *hda-1* RNAi (median fold change of 0.75 and 0.79, respectively), although again the number of genes that reached statistical significance within this group was small (9 in *set-26(-)* mutants and 12 in *hcf-1(-)* mutants on *hda-1* RNAi). Although subtle, the overall trend is consistent with reduction of *hda-1* leading to a mild deactivation of mitoUPR genes in germline-less *set-26(-)* and *hcf-1(-)* mutants. Interestingly, we found that 27 of these direct targets that were "HDA-1-dependent mitoUPR genes" were the same genes identified in Fig. 4h that exhibited de-stabilized SET-26 and HCF-1 binding in the *hcf-1(-)* and *set-26(-)* mutants respectively and were accompanied by increased RNA expression in both *set-26(-)* and *hcf-1(-)* mutants at day 1 of adulthood (median fold change of 4.15 and 3.10 at day 1 of adulthood, respectively) (Fig. 7e; Supplementary Data 3). These include the mitoUPR marker *hsp-6*, the mitochondrial fission factors *mff-2* and *drp-1*, the mitochondrial translocases *timm-23*, *scpl-4*, and *tin-9.1*, and the translational elongation factors *gfm-1*, *tufm-1*, *tufm-2*, and *tsfm-1* (Supplementary Data 3). Taken together, this suggests that SET-26 and HCF-1 stabilize each other and dampen expression of a subset of mitoUPR genes where they compete with HDA-1 for control over gene expression.

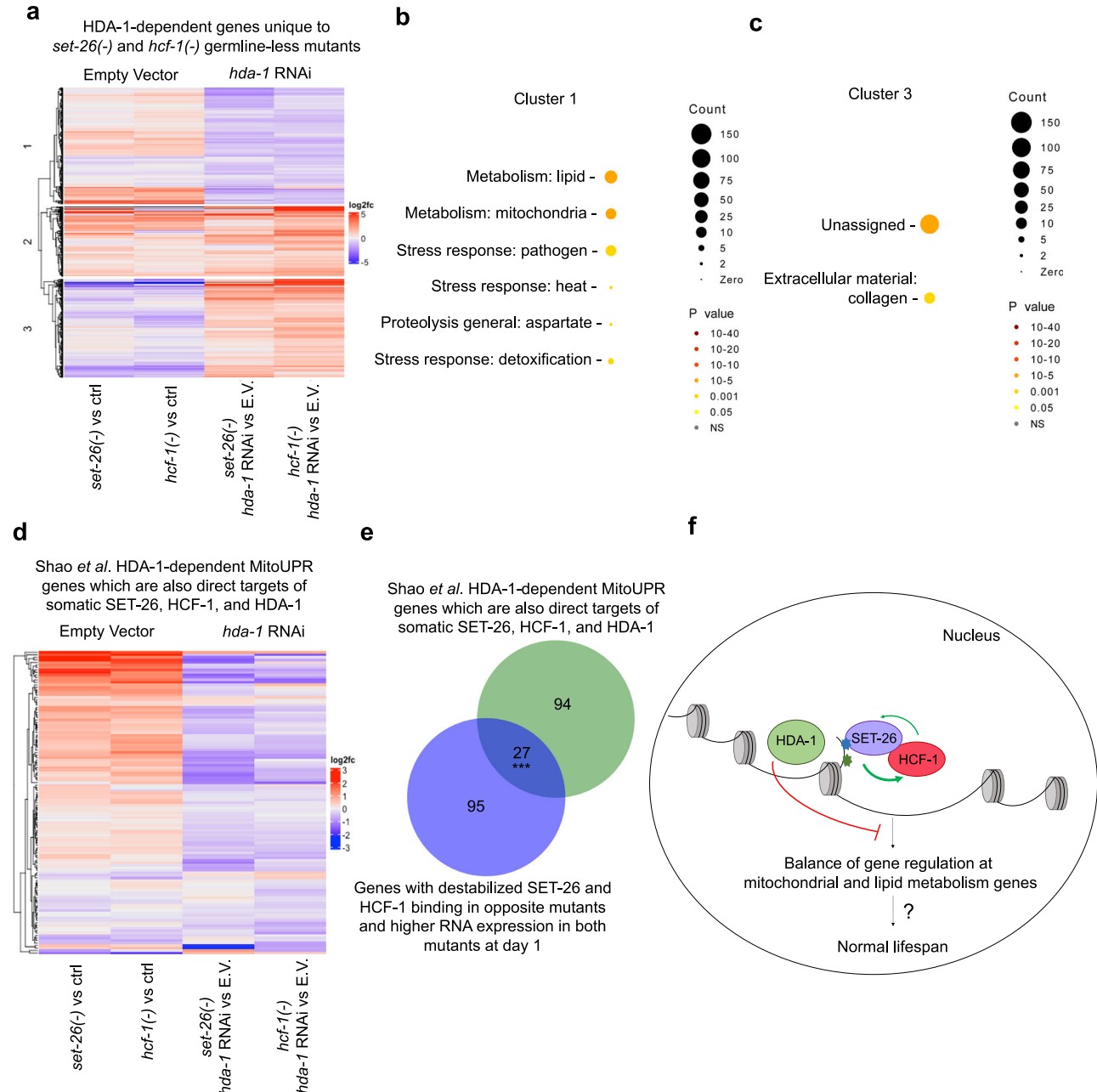

**Fig. 7 | SET-26 and HCF-1 control expression of a subset of genes which are co-regulated by HDA-1 in an antagonistic manner. a** Heatmap showing hierarchical clustering of HDA-1-dependent genes identified as differentially expressed with *hda-1* RNAi during aging in either day 12 adult *glp-1(-);set-26(-)* or *glp-1(-);hcf-1(-)* worms but not *glp-1(-)* single mutants. Log2 fold change of RNA expression for these genes in day 12 samples are indicated by color. Columns 1–2 show expression changes in *glp-1(-);set-26(-)* or *glp-1(-);hcf-1(-)* vs *glp-1(-)*, and columns 3–4 show expression changes in *glp-1(-);set-26(-)* or *glp-1(-);hcf-1(-)* on *hda-1* RNAi vs E.V. during aging. The heatmap is split into three hierarchical clusters representing different behaviors. **b**, **c** Wormcat enrichment analysis of cluster 1 or 3. Wormcat *p* values are determined by one-sided Fisher test with FDR correction. **d** Heatmap showing hierarchical clustering of direct somatic binding targets of SET-26, HCF-1, and HDA-1 that were also identified by ref. 13 as "HDA-1-dependent mitoUPR genes". Log2 fold change of these genes in our day 12 data set are shown by color. As above,

columns 1-2 show expression changes in germline-less *set-26(-)* and *hcf-1(-)* mutants *vs* controls, and columns 3–4 show the expression change in germline-less *set-26(-)* and *hcf-1(-)* mutants on *hda-1* RNAi *vs* E.V. during aging. **e** Venn diagram showing overlap of the common CUT&RUN targets of SET-26, HCF-1, and HDA-1 that are also "HDA-1-dependent mitoUPR genes" as indicated in (**d**) with the 122 genes on which somatic SET-26 and HCF-1 binding stabilize each other and show higher RNA expression in both mutants as identified in Fig. 4h. **f** The proposed model, in which SET-26 recruits HCF-1 to chromatin, HCF-1 stabilizes SET-26, and the pair work together to regulate gene expression. The model proposes that HDA-1 co-regulates a subset of genes with SET-26 and HCF-1, often mitochondrial and lipid metabolism genes, but in the opposite direction. Together, the factors contribute to the balance of gene regulation and longevity. In (**e**), *** indicates $p < 1 \times 10^{-15}$ and the overlap is higher than expected by chance, as calculated by one-sided Fisher's Test. Gene sets are provided in Supplementary Data 3. $N = 2$ for RNA-seq.

## Discussion

In this study, we find that the chromatin factors SET-26 and HCF-1 work together in both lifespan modulation and chromatin regulation in *C. elegans*. We show that SET-26 and HCF-1 operate in the same genetic

pathway to regulate lifespan, and the long lifespan of both mutants requires the histone deacetylase HDA-1 in somatic cells. All three factors have similar binding profiles at chromatin, suggesting they could regulate similar genes. Our CUT&RUN data suggest a more direct

relationship between SET-26 and HCF-1 at chromatin, and we propose that SET-26 is responsible for the majority of HCF-1 recruitment to chromatin in somatic cells. Interestingly, we identified a subset of genes, mainly functioning in mitochondrial and lipid metabolism, on which SET-26 and HCF-1 stabilize each others' binding and act to affect the expression. The genetic relationship between HDA-1 with SET-26 and HCF-1 is antagonistic and our gene expression profiling shows that HDA-1 often impacts gene expression in the opposite direction from SET-26 and HCF-1, especially on mitochondrial and lipid metabolism genes. Taken together, we put forth a model in which SET-26 recruits HCF-1 to chromatin, where HCF-1 helps stabilize SET-26 at a subset of binding sites. HDA-1 binds to these same genes and antagonizes SET-26 and HCF-1 over control of gene expression (Fig. 7f). Together, SET-26, HCF-1, and HDA-1 fine-tune gene expression, particularly of mitochondrial and lipid metabolism genes.

Since previous studies have shown that both SET-26 and HCF-1 operate in somatic cells to regulate lifespan[5,15], we focused the majority of our genomic studies on germline-less mutants to study the tissues of most relevance to the aging phenotype. Indeed, we found that HCF-1 binding to chromatin is severely affected by loss of *set-26* when looking at somatic-only samples but only minorly affected in germline-containing wildtype whole worm samples. One possible explanation for the difference between germline-containing and germline-less worms could be that SET-26 plays a more important role in recruiting HCF-1 in somatic cells than in the germline, however as we have not conducted any tissue-specific analysis of isolated germlines, we can only speculate about this possibility. It is tempting to speculate however, that SET-26 could have a tissue-specific role in recruiting HCF-1 only in somatic cells, or that another protein such as SET-9, a paralog of SET-26 which is only expressed in the germline and modulates reproduction but not lifespan[15], could contribute to HCF-1 recruitment in the germline. It will be interesting for future studies to investigate the role of SET-26 in recruiting HCF-1 in the germline and whether SET-9 plays a role.

As *C. elegans* somatic cells are all terminally differentiated, this study may also provide an incentive to study the role of the human homologs of SET-26 and HCF-1, called MLL5 and HCF-1, in differentiated cells. Most research on MLL5 and HCF-1 has been done in proliferating cell lines to study cell cycle progression and cancer, but recent evidence suggests both MLL5 and HCF-1 can play a role in neuronal development[49,50], and HCF-1 has been shown to play a role in the adult murine liver[31], highlighting the need for additional work on the mammalian homologs in differentiated tissues.

Contrarily, we find a more substantial defect in HDA-1 binding in wildtype whole worm germline-containing samples than in germline-less mutants. We speculate that this could be the result of a tissue-specific role for SET-26 in mediating HDA-1 recruitment in the germline and not in somatic cells, and it will be interesting for future studies to test this hypothesis. In light of our previous observation that SET-9 and SET-26 are required to restrict the spreading of H3K4me3 at SET-9 and SET-26 binding sites only in the germline[15], it will be interesting to investigate whether histone acetylation and HDA-1 recruitment are also altered specifically at these sites. Given that SET-26, HCF-1, and HDA-1 are all well-known regulators of the germline in *C. elegans*[15,18,21] and that in *Drosophila* the SET-26 homolog, UpSET, can recruit the HDA-1 homolog, Rpd3, to chromatin in an embryonic cell line[40], this will be an interesting avenue for future study. It may prove especially interesting given the role of all three proteins in the cell cycle and cancer in humans[16,24,51,52], and the evidence from our whole worm CUT&RUN data that SET-26 and HCF-1 binding is enriched for cell cycle genes in germline-containing worms.

In order to focus on the targets most likely to influence longevity, we characterized SET-26 and HCF-1 binding primarily in somatic cells. Interestingly, SET-26 and HCF-1 both occupy the promoters of many thousands of genes which represent a wide range of basic biological

functions, making them relatively ubiquitous and general factors at promoters. Of note, our CUT&RUN approach utilized whole worms lacking germlines, thus our results are an aggregate of binding signal obtained from all surveyed somatic cell types. Therefore, our current approach is not able to distinguish whether SET-26 and HCF-1 co-occupy the same promoters in the same cell type, or whether they occupy the same promoters independently in different cell types. Given the IP-Mass Spec interaction data between SET-26 and HCF-1 and the requirement for SET-26 in HCF-1 binding, it seems more likely that SET-26 and HCF-1 co-occupy the same promoters together in the same cell types, however future approaches will be required to test this in a tissue-specific manner.

While SET-26 and HCF-1 occupy the promoters of many thousands of genes, only a small subset of these binding sites actually change in expression in *set-26(-)* and *hcf-1(-)* germline-less mutants. Previous studies of HCF-1 in mammalian cells have shown similar findings, revealing HCF-1 to be a factor that binds thousands of active promoters, but influences the expression of only a subset of those promoters in both activating and repressive ways[30,31]. Interestingly, mammalian HCF-1 is well known to be able to recruit either activating or repressive histone-modifying complexes to chromatin at different stages of the cell cycle[16]. Thus, it is possible that SET-26 and HCF-1 interact with multiple chromatin-modifying complexes or transcription factors in the same cell type, or, given that the CUT&RUN and RNA-seq data both represent a mixture of somatic cells, that SET-26 and HCF-1 work with distinct subcomplexes in different cell types. It will be interesting to identify these potential subcomplexes in the future, the cell type they operate in, and which contribute to the longevity phenotype of the mutants. It will also be interesting to determine whether the genes bound by SET-26 and HCF-1 that do not change in expression in our dataset ever show expression changes in different cell types or physiological contexts, and if not, whether SET-26 and HCF-1 are functionally redundant at those sites or if they play a different role in shaping the chromatin landscape.

One of the most interesting findings from our study is that in somatic cells, HCF-1 binding is severely decreased genome-wide upon loss of *set-26*, whereas the majority of somatic SET-26 binding occurs largely independently from HCF-1. Importantly, we have not ruled out an indirect effect caused by loss of *set-26*, and it is entirely possible that loss of *set-26* somehow alters the chromatin environment indirectly in a manner that prevents proper HCF-1 recruitment to chromatin. However, it is tempting to speculate that SET-26 may directly recruit HCF-1 to chromatin. This is especially intriguing given evidence from human cells that mammalian homologs, MLL5 and HCF-1, physically interact[23,38], and recent work that showed MLL5 was required for proper HCF-1 recruitment at certain MLL5 target genes[38]. It will be interesting in the future to further validate whether there is a direct binding relationship between SET-26 and HCF-1, and to determine whether the highly conserved "HCF-1 binding motif" identified in the human MLL5 protein[23] is also required for the interaction between *C. elegans* SET-26 and HCF-1. We posit that SET-26 arrives at chromatin first, recruited to H3K4me3 by its PHD domain[15], and then serves as a major recruiter of HCF-1. This also may explain both the longevity of the *set-26(-)* mutant, caused by a lack of complete HCF-1 binding to chromatin, and the reason why the *set-26(-)* mutant is less long-lived than that *hcf-1(-)* mutant, as even the HCF-1 peaks that show a dramatic reduction in the *set-26(-)* mutant often still have some low level of HCF-1 present.

Although the majority of SET-26 recruitment is not altered by loss of *hcf-1*, there were still a subset of binding sites with decreased SET-26 binding in the *hcf-1(-)* mutant, suggesting that HCF-1 could help stabilize SET-26 at chromatin after being recruited. Interestingly, we found a subset of genes which exhibited lower SET-26 binding in the *hcf-1(-)* mutant, lower HCF-1 binding in the *set-26(-)* mutant, and altered RNA expression in both mutants. The upregulated genes were most

enriched for mitochondrial metabolism, while the downregulated genes were enriched for lipid metabolism, implicating SET-26 and HCF-1 together as important direct co-regulators of these pathways.

In line with a conserved role for SET-26 and HCF-1 in modulating mitochondrial gene expression, evidence from mammals also finds that reduction in HCF-1 binding at chromatin in human cells impacts mitochondrial pathways[39] and loss of MLL5 in murine cells leads to increased mitochondrial membrane potential and ROS levels[53]. In an analysis similar to that shown here, wherein Minocha et al. overlapped their ChIP-seq and RNA-seq data to identify direct targets of HCF-1 that change in expression in mouse hepatocytes, they also identified mitochondrial-related genes as the most significant GO term[31], suggesting that at least HCF-1 has a highly conserved function in directly regulating mitochondrial gene expression from worms to mammals.

We find that HDA-1 co-occupies many of the same promoters as SET-26 and HCF-1, however whether the connection between HDA-1 with SET-26 and HCF-1 is one of direct binding or mediated by the surrounding chromatin environment is unclear. Interestingly, although SET-26 is an H3K4me3 reader, it can only competently bind H3K4me3 when there are nearby histone acetylation marks[15], supporting the likelihood that machinery associated with histone acetylation, such as HDA-1, would occupy similar regions of chromatin.

Indeed, we find that it is unlikely that SET-26 or HCF-1 directly alter HDA-1 recruitment to chromatin in somatic cells, but rather we postulate that the three factors occupy the same promoters, with SET-26 and HCF-1 working to mediate gene expression in one direction, while HDA-1 works against them in the other direction. Genes that match this description within our data are enriched for mitochondrial and lipid metabolism genes, which are of particular interest given our findings that SET-26 and HCF-1 stabilize each other on and affect expression of mitochondrial and lipid metabolism genes, and the wide array of literature connecting mitochondrial perturbation with lifespan modulation[2,54,55]. Furthermore, HDA-1 has been shown to be required for activation of the mitoUPR in *C. elegans* and in human cells[13,14], and to be required for the longevity of a model of mitoUPR-mediated longevity[13]. Thus, it is particularly intriguing to find that many mitoUPR genes are activated in the *set-26(-)* and *hcf-1(-)* long-lived mutants, and their induced expression are reversed in the shorter-lived *set-26(-)* and *hcf-1(-)* mutants on *hda-1* RNAi. It is tempting to speculate that these genes may be promising candidates to explain the long lifespan of the *set-26(-)* and *hcf-1(-)* mutants and the decline in lifespan on *hda-1* RNAi. It will be critical for future studies to determine whether these genes are responsible for the longevity of *set-26(-)* and *hcf-1(-)* mutants in an *hda-1*-dependent manner, and the impact they may have on mitochondrial function in the mutants.

Given the conserved relationship between SET-26 and HCF-1 homologs in humans[23,27,38] and the conservation of SET-26, HCF-1, and HDA-1 homologs in regulating mitochondria in mammals[13,31,39,53], it will be interesting to further explore the role between these three factors in mammalian cells, especially in respect to the regulation of mitochondrial gene expression. As all three factors in humans are implicated in neurodevelopment[49,50,56] and cancer[24,51,52], and HCF-1's function in cancer has already been linked to the regulation of mitochondrial gene expression[39], understanding how MLL5, HCF-1, and HDAC1/2 interact and the pathways they regulate in humans could have broad implications for understanding both aging and human disease.

## Methods

### *C. elegans* maintenance

Strains with a *glp-1(e2141)* mutant background were maintained at 16 °C, whereas all other strains used were maintained at 20 °C. To maintain worms, animals were well fed on 6-cm nematode growth medium (NGM) plates containing 30 µg/mL streptomycin seeded with 200 µl of a 5× concentrated strep-resistance OP50 overnight culture. In all experiments with the exception of RNAi experiments (which utilize the HT115 bacterial strain), worms were fed OP50. All strains used in this study contained the wild-type (N2H) allele of *fln-2*, a gene which has been shown to be mutated in common *C. elegans* strains and can affect lifespan[57]. All worms used for analysis in this paper were hermaphrodites. See Supplementary Data 5 for a list of all strains used in this paper.

### Immunoprecipitation-mass spectrometry (IP-Mass Spec)

Two independent IP-Mass Spec analyses were carried out in single replicate by different labs using the strains IU171 *rwIs3[phcf-1::hcf-1::gfp, unc-119];hcf-1(pk924)* and IU382.1 *rwIs9[phcf-1::hcf-1::gfp pmec-7::rfp]*, both of which contain an integrated HCF-1::GFP array[5,17]. IU171 and IU382.1 were used only for the IP-Mass Spec, whereas in all other cases the tagged HCF-1 strain refers to the CRISPR knock-in created in this study. The first experiment used IU171 and was conducted exactly as previously described[17], using worms grown in liquid culture and an anti-GFP antibody (3E6, Invitrogen). The second experiment was conducted similarly as previously described[58]. Briefly, worms were grown on eight, 9 cm High Growth plates (3 g/L NaCl, 30 g/L agar, 20 g/L peptone, 20 mg/L cholesterol, 1 mM CaCl$_2$,1 mM MgSO$_4$, 25 mM potassium phosphate pH 6.0) seeded with OP50 until the bacteria was nearly, but not yet, eaten. Unsynchronized worms were then harvested, and proteins were extracted by Fastprep in 1x lysis buffer (20 mM Tris-HCl pH 8.0, 150 mM NaCl, 0.1% NP-40, 2 mM EDTA pH 8.0). HCF-1::GFP was immunoprecipitated using 20 µl of GBP beads with 6.5 mL worm lysate supernatant and protease inhibitor cocktail, followed by 5 h incubation at 4 °C. Samples were analyzed by immunoblot followed by an LTQ Orbitrap mass spectrometer (ThermoFisher Scientific). For the experiment using IU171, RAW data from the mass spectrometer were searched using SEQUEST and a database of the *C. elegans* proteome, as described in ref. 59. In both experiments, relative protein abundance for each putative HCF-1 interactor was also measured following immunoprecipitation in untagged N2 worms as a negative control. IP-Mass spec proteomics data from the two independent experiments have been deposited to the ProteomeXchange Consortium via the PRIDE partner repository with the dataset identifier PXD047509 (performed by co-author C.G.R.) and PXD047247 (performed by co-author M.Z.).

### Lifespan

Lifespan analysis was conducted similarly as previously described[15]. Specifically, 3–10 gravid adults were allowed to lay embryos on plates with the appropriate bacteria for 2–5 h at the growth temperature of the strain (16 °C for *glp-1(e2141)* strains or 20 °C for all other strains). The adults were removed and plates containing synchronized embryos were shifted to 25 °C for ~48 h until they reached day 1 of adulthood, which was marked as day 1 for lifespan analysis. 30–35 day 1 adults were then transferred onto 2–3 plates containing either the same bacteria or shifted onto the appropriate bacteria for aging. Specifically, for RNAi experiments using adult *hda-1* RNAi, all worms were grown on the empty vector bacteria "L4440", then worms were split between L4440 and *hda-1* RNAi at day 1 of adulthood. For lifespan experiments conducted on OP50, worms were grown on stock plates (as described above) containing 5X concentrated live OP50, and then transferred at day 1 of adulthood to NGM plates seeded with 200 µl of a 10× concentrated strep-resistant OP50 overnight culture which had been killed by washing twice with a solution of LB containing 100 µg/mL carbenicillin and 15 µg/mL tetracycline as previously described[8]. Worms used for lifespan were transferred to fresh plates every 1–2 days until the end of their reproductive period and monitored for survival every 2 days until all worms died. The chemical 5-fluoro-2'-deoxyuridine was not used. Worms were scored as dead when they failed to respond to a gentle prod on the head. Worms that exploded, experienced internal hatching of offspring (bagged), or crawled onto the side of the plate

were marked as censored on the day of the event. Lifespan data were analyzed with OASIS 2 online survival analysis tool[60]. The Kaplan-Meier estimator was used for re-plotting survival curves in Excel and log-rank tests were utilized to determine whether two lifespans were significantly different. All lifespans were conducted at least twice, and the data displayed in each figure show one representative replicate (See Source Data for all lifespan data).

## RNAi

RNAi was administered by feeding. Briefly, HT115 bacteria expressing double stranded RNA against the gene of interest were obtained from the Ahringer library and verified by Sanger sequencing unless otherwise noted (see "*hda-1 3'UTR* RNAi construction" below). RNAi bacteria and the control empty vector bacteria, L4440, were grown in overnight cultures containing 100 μg/mL carbenicillin and 15 μg/mL tetracycline at 37 °C for 12–16 h. The overnight cultures were diluted 1:20 in LB containing 100 μg/mL carbenicillin and grown at 37 °C for 2–4 h until an optical density of 0.6–0.8 was reached for all cultures. Cultures were then induced with 1 mM total concentration of IPTG, and grown at 37 °C for an additional 2–3 h. After induction, bacteria were spun down at 3000 rpm for 15–20 min at room temperature and concentrated 50-fold. 200 μl of concentrated bacteria were seeded on 6-cm "RNAi plates" and 1–2 mL were seeded on 15-cm "RNAi plates", which were NGM plates lacking streptomycin and with the addition of 100 μg/mL ampicillin, 15 μg/mL tetracycline, and 1 mM IPTG. Unless specifically noted as "*hda-1 3'UTR* RNAi", "*hda-1* RNAi" refers to the use of the Ahringer RNAi construct.

## CRISPR

The *hcf-1::gfp::3xflag* insertion strain was generated using CRISPR-mediated genome editing as previously described[61], such that a flexible linker, GFP and 3X FLAG tags were inserted immediately before the endogenous *hcf-1* stop codon in the N2 background. The Dickinson et al. method[61] was used to generate and inject a repair template and guide RNAs, and to select positive hits that were hygromycin-resistant rollers. After initial selection, the self-excising cassette, containing the roller and hygromycin-resistant markers, was removed and successful removal was verified by Sanger sequencing.

The *set-26::ha* insertion strain was generated using CRISPR-mediated genome editing as previously described[15,62,63], such that an HA tag was inserted immediately before the endogenous *set-26* stop codon in the N2 background.

The *hda-1::gfp::ha* insertion strain was generated by SunyBiotech using CRISPR-mediated genome editing such that the GFP and HA tags were inserted immediately before the endogenous *hda-1* stop codon in the N2 background. All strains generated in this study are available from the authors upon reasonable request.

## CUT&RUN

In experiments in which worms were grown on OP50, embryos were collected from full stock plates as previously described[29]. 3000 embryos were seeded per plate on 15-cm NGM plates containing 1 mL of a 25-times concentrated overnight culture of streptomycin-resistant OP50. In experiments utilizing *glp-1* RNAi, 150–300 synchronized gravid adults were pre-incubated with RNAi plates seeded with *glp-1* RNAi bacteria overnight for 10–14 h at 20 °C. The next day, the gravid adults were transferred to fresh *glp-1* RNAi plates for 3–6 h at 20 °C, after which the resulting synchronized embryos were washed off of the plates, counted, and seeded, either with 3000 worms per plate for 15-cm plates, or 200 worms per plate for 6-cm plates. For consistency, in experiments with *glp-1* RNAi, control worms laying embryos on L4440 were pre-incubated overnight with L4440 in the same way. Enough embryos were prepared such that 3000 worms were available for each CUT&RUN reaction as previously described[29]. After seeding embryos, plates (either OP50 or RNAi) were incubated at 25 °C for ~48–52 h until

the plates were full primarily of young adults. After reaching the young adult stage, worms were washed off plates and CUT&RUN experiments were performed precisely as previously described by us in detail elsewhere[29], using 1 μl of undiluted primary antibody (targeting either HA (Cell Signaling #3724), FLAG (ThermoFisher #MA1-91878) or histone H3 (abcam #ab1791)) in a 100 μl reaction, similarly to other standard CUT&RUN protocols[64,65]. When using the anti-FLAG primary antibody, which was produced in mouse, 1.5 μl of undiluted secondary anti-mouse antibody produced in rabbit (abcam # ab46540) was used in a 150 μl reaction to increase yield as described previously[29].

Sequencing libraries were prepared using the NEBNext Ultra II DNA Library Prep Kit for Illumina (NEB, cat. No E7645S and E7335S) and amplified using 14 PCR cycles following the manufacturer's instructions with slight modifications as previously described by us in detail elsewhere[29]. Libraries were submitted for 2 × 32 paired-end sequencing with an Illumina NextSeq 500 machine. All CUT&RUN experiments were repeated twice, with biological replicates grown and collected separately. See Supplementary Data 5 for a detailed list of antibodies used in CUT&RUN experiments. CUT&RUN sequencing data have been deposited in the NCBI Gene Expression Omnibus under accession code GSE224076 and under SuperSeries GSE224078.

## CUT&RUN, ATAC-seq, and ChIP-seq data analysis

Data analysis was performed similarly as described[45]. For detailed information, see Supplementary Information - Supplementary Methods.

## RNA-seq

Embryos were collected from full stock plates of *glp-1(-)*, *glp-1(-);set-26(-)*, and *glp-1(-);hcf-1(-)* strains as described above for CUT&RUN. Embryos were seeded at ~60 worms per plate on 6-cm RNAi plates containing L4440 empty vector bacteria and were grown at 25 °C for ~48 h until most worms were young adults. Approximately 300 worms were collected as day 1 adults, and the remaining population was washed off of their plates using M9 buffer containing 0.05% Tween-20 and split evenly onto either fresh L4440 plates or *hda-1* RNAi plates. Two days later, approximately 300 worms were collected from each genotype on either L4440 or *hda-1* RNAi as day 3 adults, and the remaining population was grown until day 12 of adulthood, with one transfer at day 7 onto fresh RNAi plates to ensure worms were well fed. At each collection, worms were washed off of plates with M9 buffer containing 0.05% Tween-20. Worms were settled and washed 1–5 times with M9 buffer, and then were snap frozen in liquid nitrogen in 500 μl of Tri Reagent. Samples were alternatively thawed, vortexed, and re-frozen in liquid nitrogen at least five times to break apart worms. 100 μl chloroform was added per tube, and samples were vortexed and then spun at 18,000 rcf for 15 min at 4 °C. The aqueous layer was then transferred to a fresh tube, and 250 μl isopropanol was mixed into each sample. 1 μl GlycoBlue was added and, after a 10 min incubation at RT, samples were spun at 12,000 rcf for 10 min at 4 °C. The supernatant was discarded, and the RNA pellet was washed with 70% ethanol. The pellet was then air dried and dissolved in nuclease-free water. RNA was treated with DNase to remove residual DNA following the instructions from the TURBO DNase kit (Invitrogen, AM1907), and RNA was further purified by ethanol precipitation using 0.1 volumes of 3 M sodium acetate, 2.5 volumes of ice cold 100% ethanol, and 1 μl GlycoBlue overnight at −80 °C. The following morning, samples were spun at 18,000 rcf for 30 min at 4 °C, pellets were washed twice with ice cold 75% ethanol, and the pellet was air dried and dissolved in nuclease-free water. RNA-seq libraries were prepared with the QuantSeq 3' mRNA-Seq Library Prep Kit (Lexogen) following the manufacturer's instructions beginning with 315 ng RNA from each sample and using 13 PCR cycles. Libraries were quantified with Qubit and quality checked using Bioanalyzer and then submitted for single-end 86 bp sequencing with an Illumina NextSeq 500 machine. RNA-seq

data have been deposited in the NCBI GEO database under accession code GSE224075 and under SuperSeries GSE224078.

## RNA-seq data analysis

In the Linux environment, adapter sequences were trimmed and low-quality reads were filtered out from sequencing files using Trim Galore! (v0.6.5), which utilizes Cutadapt (v3.4)[66] and FastQC (v0.11.8), with the settings –q 20 –fastqc. Trimmed sequencing files were then aligned to the ce11/WBcel235 *C. elegans* reference genome using STAR (v2.7.9a)[67] with the options –runThreadN 2 –quantMode GeneCounts –-outFilterMultiMapNmax 1 –outFilterMismatchNmax 2 –outSAMtype BAM SortedByCoordinate. The resulting tab-delimited text files containing read count per gene, with column 3 (the middle column of counts) representing the correct counts for 3′ RNA-seq data, were used to create a matrix of gene expression to compare multiple samples. Matrices contained either all genotypes at the same age on the same bacteria (e.g. *glp-1(-)*, *glp-1(-);set-26(-)*, and *glp1(-);hcf-1(-)* on L4440 at Day 3) or the same genotype at the same age on different bacteria (e.g. *glp-1* on L4440 and *hda-1* RNAi), depending on whether the goal was to identify genes differentially expressed in the longevity mutants or genes dependent on *hda-1* in one genotype. The matrices were uploaded into RStudio and used as input to DESeq2 (v1.34.0)[68]. Genes with low read counts were pre-filtered out before differential analysis, and only genes that had more than 10 read counts in at least 2 samples were kept for analysis. PCA plots were generated using the 'vst' transformation function in DESeq2 and the 'plotPCA' function with default settings, followed by the 'ggplot' function in ggplot2 (v3.3.6)[69]. DESeq2 was used to find the differentially expressed genes either by genotype or condition (*hda-1* RNAi) using the standard DESeq command. Only the genes reaching an adjusted *p* value of less than 0.05 were kept and considered significant. Normalized counts for particular genes of interest were plotted using the 'plotCounts' function in DESeq2. Genes sets in each genotype were compared to each other or to CUT&RUN genes using BioVenn (www.biovenn.nl)[70], and the significance of overlapping genes was determined by Fisher's exact test calculated in RStudio. To compare the genes which are differentially expressed during aging in the current study to Pu et al., (as in Supplementary Fig. 7a, b), differentially expressed genes were downloaded directly from ref. 35 (supplementary information Table S7). WormCat 2.0 (www.wormcat.com) was used for gene ontology enrichment analysis, where the *p* values are determined by Fisher's test with false discovery rate (FDR) correction[71].

To generate heatmaps using RNA-seq data, gene expression matrices were uploaded into DESeq2 as described above, except lowly expressed genes were not pre-filtered. DESeq2 was again used to find differentially expressed genes as described above, and the log2 fold change values were extracted from the DESeq2 result table for each gene to be plotted in the heatmap. Heatmaps were plotted with the 'Heatmap' function in the ComplexHeatmap package (v2.10.0)[72] with the options 'cluster_rows = T, cluster_columns = F, show_row_names = FALSE, row_dend_reorder = TRUE, na_col = "black"'. To split a heatmap, the option 'split = 3' was added and the genes from each clustered section were extracted using the 'row_order' function in ComplexHeatmap.

## Immunoblotting and quantification

300–1000 synchronized adult worms were washed off of plates with M9 buffer containing 0.05% Tween-20. Worms were settled and washed 1–5 times with M9 buffer, and then snap frozen in liquid nitrogen in minimal M9 buffer. Immunoblots were run as previously described[15]. Samples were boiled for 7 min at 100 °C in an equal volume of 2X SDS sample buffer and run through an SDS page gel containing 8% acrylamide. After separation, samples were transferred to a nitrocellulose membrane for 1 h at 4 °C. The membrane was then blocked with Tris-buffered saline containing 0.1% tween (TBST) and 5% BSA, washed with TBST and

incubated overnight at 4 °C with primary antibody (1:1000 for FLAG (ThermoFisher #MA1-91878) or HA (Cell Signaling #3724); 1:2000 for H3 (abcam #ab1791)) diluted in TBST + 5% BSA. The membrane was then washed again with TBST three times, and incubated with a fluorescent secondary antibody (1:5000 dilution; Licor #926-32211 or #926-68070) at RT for 1.5–2.5 h in the dark. Membranes were washed three times with TBST, once with TBS, and then imaged using the ChemiDoc MP Imaging System. Immunoblots were repeated 2–3 times. Fluorescence of the target protein was normalized to H3 fluorescence using the BioRad Image Lab Software, and then fold change was calculated between conditions. Statistical significance of fold change compared to control was analyzed using one-tailed or two-tailed *t* tests as specified in figure legends. See Supplementary Data 5 for a detailed list of antibodies used in immunoblotting experiments.

## *hda-1 3′ UTR* RNAi construction

The *hda-1 3′ UTR* RNAi construct was designed to target a unique 774 bp region in the 3′ UTR of *hda-1*. This region in *hda-1* was amplified from N2 genomic DNA with primers containing restriction digest sites for EcoO109I and SacI-HF restriction enzymes. Purified PCR products and purified DNA from miniprep of the L4440 vector were digested with EcoO109I and SacI-HF in rCutSmart™ (NEB) buffer for 1 h or 16 h at 37 °C. The digested products were purified and ligated together using T4 DNA ligase (NEB). The ligation product was then retransformed into high-efficiency 5-alpha competent *E. coli* cells (NEB) which were grown overnight at 37 °C. Colonies were grown for 12–16 h overnight at 37 °C in LB containing 100 μg/mL carbenicillin and plasmids were purified by miniprep and verified with Sanger sequencing. 1 μl of the correct plasmid was then retransformed into homemade HT115 competent cells and used for RNAi. See Supplementary Data 5 for primer sequences.

## qRT-PCR

For qRT-PCR, 50–300 worms were collected at day 3 of adulthood on L4440 or *hda-1* RNAi, with *hda-1* RNAi initiated on day 1 of adulthood. Samples were frozen and RNA isolated, DNAse-treated, and purified as described for RNA-seq above. cDNA was synthesized using qScript cDNA SuperMix from Quantabio following the manufacturer's instructions, then 1 μl RNase H was added to the reaction and incubated for 20 min at 37 °C to remove residual RNA. qRT-PCR reactions were performed in triplicate on a LightCycler 480 II machine in 10 μl reactions containing 2 μl of tenfold diluted cDNA, 1 μl of 5 μM primers, and 7 μl of Taq mix containing a SybrGreen mix, dNTPs, and hot start Taq. Ct values were obtained using the LightCycler 480 Software high sensitivity setting. Triplicate Ct values were averaged, then normalized to first a housekeeping gene (*ama-1*) and then control samples using the $2^{-\Delta\Delta Ct}$ method[73] to compute fold change of *hda-1* with *hda-1* RNAi compared to empty vector bacteria. Two biological replicates were conducted for each qRT-PCR experiment. The statistical significance of fold change compared to control was analyzed using one-tailed *t* tests. See Supplementary Data 5 for primer sequences.

## Fluorescence imaging and quantification

For detailed information, see Supplementary Information - Supplementary Methods.

## Statistics and reproducibility

Statistical analyses were performed in Microsoft® Excel (v16.66.1) for immunoblotting, imaging, and qRT-PCR experiments, in OASIS 2[60] for lifespan analyses, in WormCat 2.0[71] for gene ontology analyses, and in RStudio (v2022.02.0 + 443) for all other analyses. The R packages used for differential analyses in CUT&RUN and RNA-seq data sets are included in the methods section and Supplementary Methods. All experiments were repeated with at least two biological replicates with similar results. The details of types of statistical analyses and *p* value

cutoffs for each experiment are included in the corresponding figure legend, and the raw data including sample size and statistical analysis for each figure are included in the Source Data file.

## Reporting summary

Further information on research design is available in the Nature Portfolio Reporting Summary linked to this article.

## Data availability

CUT&RUN and RNA-seq raw sequencing data, CUT&RUN bigwig files for visualization, and the RNA-seq read count matrix have been deposited in NCBI's Gene Expression Omnibus and are accessible through GEO series accession number GSE224075 (RNA-seq), GSE224076 (CUT&RUN) and can be found as SuperSeries GSE224078. IP-Mass spec proteomics data from the two independent experiments have been deposited to the ProteomeXchange Consortium via the PRIDE partner repository with the dataset identifier PXD047509 (performed by co-author C.G.R.) and PXD047247 (performed by co-author M.Z.). ChIP-seq data of H3K4me3 and H3 in day 2 adult *glp-1(e2141)* mutants were downloaded from GEO series GSE101964[35]. ATAC-seq data of accessible regions in young adult *glp-1(e2144)* mutants were downloaded from GEO series GSE114439[36]. Differentially expressed genes during aging were downloaded directly from ref. 35. Blacklisted regions were obtained from ENCODE lists[32] and HOT regions were obtained from Chen et al.[33]. For genome alignment, the ce11/WBcel235 *C. elegans* reference genome was downloaded from Ensembl (https://useast.ensembl.org/Caenorhabditis_elegans/Info/Index). Source data are provided with this paper.

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

## Acknowledgements

Thank you to members of the Lee Lab for helpful suggestions and dis-cussions in all aspects of experimental design and data analysis. Thank you to Rada Omanovic for the buffer and plate preparation. Thank you to Dr. Wenke Wang for strain construction. Thank you to the Cornell Uni-versity Institute of Biotechnology and Genomics Facility for Illumina sequencing and computing power. Wormbase.org was used to obtain basic information throughout this study[74]. Some strains were supplied by the *Caenorhabditis* Genetics Center (CGC), and the work was sup-ported by NIH funding R01 AG024425 (to S.S.L.) and NSF GRFP funding DGE-1650441 (to F.J.E.). This work was also supported by grants from the Swedish Research Council (VR) 2017-06088 and 2019-04868 (to C.G.R.), the Swedish Cancer Society (Cancerfonden) 20 1034 Pj (to C.G.R.), the Novo Nordisk Foundation NNF21OC0070427 and NNF22OC0078353 (to C.G.R.), and an ICMC project grant (to C.G.R.).

## Author contributions

F.J.E. and S.S.L. designed the study and wrote the paper. L.Y.L. and C.C. both carried out a lifespan experiment and qRT-PCR experiment. M.Z. and C.G.R. performed the IP-Mass Spec analyses. F.J.E. carried out all other experiments and data analysis. S.S.L. secured funding and supervised the experimental plans.

## Competing interests

The authors declare no competing interests.
