## [Peer Review File · Nature Communications]

The chromatin factors SET-26 and HCF-1 oppose the histone deacetylase HDA-1 in longevity and gene regulation in *C. elegans*Editorial Note: Parts of this Peer Review File have been redacted as indicated to remove third-party material where no permission to publish could be obtained.

REVIEWER COMMENTS

Reviewer #1 (Remarks to the Author):

The authors set out to provide mechanistic insight into how *C. elegans* SET-26 and HCF-1, two conserved chromatin-associated proteins, regulate gene expression and modulate longevity. The premise is that the two proteins physically interact, although data to support this claim is not provided. The authors show widespread binding of these factors to chromatin, and an overall reduction in HCF-1 binding in *set-26* mutants. They also show common up and down-regulation of a subset of genes. Loss of the class I HDAC *hda-1* reduces the increased lifespan of both *set-26* and *hcf-1*. Based on these findings, they suggest that SET-26 recruits HCF-1 to regulate gene expression, and in particular mitochondrial function genes, to modulate longevity, and that HDA-1 opposed this activity.

Overall this study remains quite descriptive, the genome wide analysis does not go far enough in providing significant insight on the action of these widely expressed proteins on chromatin and their impact on longevity. The link to mitochondrial function and how it may impact the longevity phenotype is too speculative.

Specific points

1) this work originates from the purported observation that HCF-1 and SET-26 physically interact, or copurify in proteomics analysis. The authors state that this was shown in previous work (Rizki et al PLoS Genetics), but I could find no trace of these experiments in the cited paper. Assuming that the proteins were indeed found in the same proteomic experiment, the authors have all the tools (endogenously tagged proteins) to at least try to confirm it by co-IP. This is especially important as HCF1 is a notoriously sticky protein (<https://www.uniprot.org/uniprotkb/P51610/entry#interaction>) that copurifies with many chromatin associated complexes eg MLL, SIN3, NSL., and SET-26 has also been found in several MS-IP experiments in *C. elegans* (eg <https://doi.org/10.2144/btn-2021-0075>, DOI: 10.1093/nar/gkz880) . The only evidence presented to support this physical interaction is the presence of an HCF-1 consensus motif in SET-26 (Figure S1) that is found in the human HCF1 interactor MLL5. In reference to this homology, what is the evidence that SET-26 is in fact the *C. elegans* homologue of MLL5? SETD5 appears to be more closely related than MLL5.

2) Figure 2a shows that both *hcf-1* and *set-26* mutants are long lived (as previously published), and the double mutant is not longer lived than either single. This is interpreted as evidence of the two proteins acting together in longevity. While this may well indicate that both genes act in the same pathway, one can also imagine a scenario in which the non-additive effect is due to the two proteins independently targeting "longevity genes" so that their absence leads to misexpression beyond a threshold sufficient to increase longevity; increased misregulation (whether activation or repression) in the double mutant would not necessarily lead to a further increase in longevity. The interpretation of these results should therefore be nuanced.

Chromatin profiles:

3) tracks from cut&tag experiments seem quite noisy on IGV, which is quite surprising given that it is known to significantly reduce background compared to ChIP-seq. Do the authors have an explanation? More than 13,000 SET-26 "peaks" are detected and more than 9000 HCF-1 "peaks"...this is a surprisingly large number. How does the number of peaks detected here compare to previous ChIP-seq which detected SET-26 and SET-9?

The authors show a large degree of overlap between HCF-1 and SET-26 peaks. Given the large number of "peaks", I expect the authors may see a similar overlap with many other chromatin bound factors as well. Did they compare their list of peaks to other factors whose profiles are available through ModEncode? Did the authors check how these overlap with high-occupancy target (HOT) regions? And did they filter out regions from the available black list available through Encode?

I also don't understand the rationale for using histone H3 to normalize, but an expert may be a better judge of the methodology here.

Henikoff described a peak caller specific for Cut&Run analysis : SEACR(Meers MP, Tenenbaum D, Henikoff S. (2019). Peak calling by Sparse Enrichment Analysis for CUT&RUN chromatin profiling. Did the authors compare MACS2 to this method to see whether they obtain a similar number of peaks?

4) In Fig 1e the zoom is on a region of over 80 kb, with SET-26 binding a region of 20kb. It is difficult to reconcile this with gene specific regulation, or promoter peaks centered around the ATG shown in 1b. The authors should zoom in on a smaller region, as in figure S3.

Also, peaks appear between the large blocks, but these are not called by MACS2. Can the authors explain?

5) the authors previously showed SET-26 to be a H3K4me3 reader. How do the binding profiles shown here overlap with H3K4me3, which is mostly confined to promoters?

5) According to Fig 2D the expression of the large majority of genes bound by both HCF-1 and SET-26 (8379) is not altered in either mutant. Where are HCF-1 and SET-26 binding in these 8379 regions? Suppl Figure 1h shows binding just about everywhere (exons, introns, promoters, downstream of promoters...). It looks like these factors are not enriched at regulatory regions, while fig 1c shows that the majority of peaks are upstream of the ATG. I found this confusing
How does this compare to mammalian HCF1, which is mostly restricted at promoter regions?

6) The cut & run experiments suggest that SET-26 is required for most of HCF-1 recruitment to chromatin in the soma. If this is the case wouldn't you expect very similar phenotypes between hcf-1 and set-26 mutants? hcf-1 mutants suppress synMuv phenotypes and ectopic expression of germline genes in the soma...as do chromatin gactors mrg-1, sin-3, dpy-30. Does the set-26 mutant share any of these phenotypes?

Given that HCF-1 copurifies with several chromatin associated factors it hard to imagine that SET-26 is responsible for most of its recruitment throughout the genome. This raises the question that the observed effect may be completely indirect, for example SET-26 alters the expression of another protein or chromatin mark that is required to stabilize HCF-1 on chromatin..

8) RNA-seq -analysis was only carried out on 2 samples/genotype. Although the PCAs look ok, the fact that collagen genes are enriched suggests lack of synchronization, which could impact results.

7) Interaction with HDA-1:

The authors again cite previous work showing HDA-1 was identified with HCF and SET-26 in mass spec experiments. They should validate the interaction by co-IP. An overlap in binding to chromatin does not necessarily show physical interaction or a common function. The authors should provide additional evidence to sustain the claim that HCF-1, SET-26 and HDA-1 physically interact. It is also surprising that published HDA-1 ChIP-seq tracts (ModEncode) show better resolution than HDA-1 cut&tag.

RNA-seq on hda-1(RNAi) : the authors take animals at day 12 of adulthood because they claim that at day 3 no changes in gene expression were observed. This is very surprising. What is the evidence that hda-1(RNAi) is working? Fig S7 only shows a 50% decrease in hda-1RNAi.

In general the reasoning in this whole section is very confusing

8) additional experiments could support the claim that binding to mitoUPR genes is relevant

The authors show increased expression of mitoUPR genes hsp-6 and hsp-60 in set-26(-) and hcf-1(-) young adults. Expression of the widely used reporters hsp-6::GFP and hsp-60::GFP should be used to show that mitoUPR is activated at baseline in these mutants, as they suggest.

hda-1(RNAi) results in decreased expression of hsp-6::GFP following induction of mitochondrial stress. Did the authors test whether SET-26/HCF-1 antagonism of HDA-1 in gene regulation extends to stress induced hsp-6 expression ? This would reinforce the model of opposing function in the regulation of

mitochondria-related functions.

Reviewer #2 (Remarks to the Author):

This manuscript presents findings that establish a relationship between the chromatin-associated proteins SET-26 and HCF-1 with the histone deacetylase HDA-1. The authors build upon their previous studies of SET-26 and HCF-1 by combining insights from genomic approaches like CUT&RUN and RNA-seq with functional genetic tests to probe the interdependence of these chromatin proteins. Long-lived *set-26* mutants and *hcf-1* mutants have similar mis-regulation of somatically-expressed genes, and a significant subset of these genes have both SET-26 and HCF-1 at their promoters, indicating that both proteins co-stabilize each other at specific gene promoters. Loss of HDA-1 suppresses the longevity of both mutants, and HDA-1 was found to bind many of the same promoters as SET-26 and HCF-1. However, HDA-1 binding in somatic cells was not affected by the loss of either chromatin-associated protein, indicating that HDA-1 is not involved in the longevity of *set-26* mutants or *hcf-1* mutants. Because the effects of *hda-1* RNAi in either mutant has an opposite effect on gene expression, the authors propose a model where HDA-1 competes with SET-26 and HCF-1 to control gene expression.

Overall, this is a well-written paper that lays a solid groundwork for the authors' model of how somatic gene expression is mediated by the opposing actions of chromatin regulators. The experiments described in this manuscript are well-conceived and include important controls, including replication of their findings in *glp-1* mutants (to corroborate RNAi results) and in wild-type animals with an intact germline. This work has critical implications for future mammalian studies, as it makes a strong case for investigating the roles of these chromatin-associated proteins in terminally differentiated cells (or even *in vivo*), rather than the current standard of using proliferating cell lines. In conclusion, this study addresses the mechanism of longevity in chromatin mutants, pinpointing their effect on lifespan to mitochondrial stress response genes, and will be of interest to those studying longevity, stress resistance, and how chromatin affects transcription.

The following are minor points provided as constructive feedback for the authors:

1. The relationships identified by peak overlap in the CUT&RUN and corroborated by effects on gene expression in each mutant condition are convincing. However, one drawback of genomic approaches like these is that they reflect changes in a population of cells. In this study, whole animals were used for the genomic approaches, which means that many somatic cell types are represented in the samples. Therefore, it is possible that HCF-1 and SET-26 do not directly co-occupy the same promoters, but rather that HCF-1 regulates those genes in certain cell types, while SET-26 regulates the same genes but in completely different cell types. (The loss of HCF-1 binding in *set-26* mutants makes this alternative interpretation less likely, but it is still formally possible.) Although it is beyond the scope of this paper to experimentally distinguish between this model and the one presented by the authors, this possibility should be addressed in the discussion.

2. Figure 7 includes the interesting result that depletion of *hda-1* in *set-26* mutants or in *hcf-1* mutants reverses some of the gene expression changes observed in the mutants alone. It would be helpful for interpreting these findings to know more details about the analysis. For example, what percentage of mis-expressed genes in the single mutants are suppressed by *hda-1* knockdown? Of the genes that are up-regulated in *set-26* mutants (or in both mutants), how many of these are now unchanged from control, or even down-regulated, in *hda-1* RNAi; *set-26* mutants? Similarly, when combining their CUT&RUN data with existing RNA-seq data from Shao et al 2020, it would help to have a sense of what is meant by overlap: how many genes were originally identified as HDA-1-dependent mitoUPR genes, and of those, how many had HDA-1, HCF-1, and SET-26 present at their promoters? Of this subset of genes with all three chromatin regulators bound at their promoters, what percentage were upregulated/downregulated in *set-26* mutants or *hcf-1* mutants, and how many of these

misregulated genes (which mostly appear to be upregulated?) are now downregulated by hda-1 knockdown?

3. I am intrigued by the authors' model of how HCF-1 and SET-26 work together to counteract the activity of HDA-1, specifically at mitoUPR genes. However, they rely primarily on correlation to dissect the relationships of these chromatin regulators on gene expression – they did not functionally test this interaction at these loci. It is beyond the scope of this particular study to fully validate their model, but the model would be significantly strengthened if they demonstrated a change in chromatin at the promoters of these genes. For example: ChIP-qPCR against H3/H4 acetylation at candidate genes identified in this study in set-26 mutants compared to hda-1 RNAi; set-26 mutants. Alternatively, the gene regulation activity of HDA-1, SET-26, and HCF-1 may independent of direct effects on chromatin state, in which case, future studies will be able to pursue different mechanisms.

Reviewer #3 (Remarks to the Author):

Emerson et al. present a well-design study rich in data that describes how SET-26 and HCF-1 work in opposition of the role of HDA-1 in the regulation of gene expression that influences health and longevity. The authors develop a new model where SET-26 and HCF-1 stabilize each other on chromatin and balance the histone deacetylase activity of HDA-1 that results in a fine focus of gene expression; particularly in longevity-associated genes. Overall, this is an outstanding resource and I only I have one experimental suggestion and two suggestions for the text of the manuscript to make it more impactful. Enthusiasm for this work is very high and I believe it will be well-received and of great interest to the readership of Nature Communications.

Experimental suggestion:

The manuscript lacks measures of healthspan. Although careful detail is taken to study lifespan the manuscript would benefit if the different combinations of genetic and RNAi conditions included an assessment of health parameters (examples, not all are needed, include: movement, intracellular lipid stores, pharyngeal pumping, etc.) at the time points measured in the study (e.g., Day 1 and Day 12).

Text suggestions:

1. The supplementary data includes a large resource that is inadequately described in the text. I recognize that it is impossible to include everything in the text but expanding the results and discussion to include genes and gene families that are of particular importance (include the magnitude of change) would make the text more useful. The authors include statements about mitochondrial genes and cell cycle already, additional examples like this would enhance the text.
2. Statements regarding enrichment and differential activity in the soma and germline need more support (only statements that directly compare soma to germline). The analyses in the context of a germlineless animal are adequately performed, but it is impossible in this experimental set up to account for physiological and cell non-autonomous changes to the animal in the absence of a germline. To make these conclusions what is needed is an analysis of isolated germlines. I am not proposing that this must be performed for the study to be published as there is already a massive amount of interesting data to comb through.

*****I want to be explicit here that the experiments performed are high quality and the data is exciting. The study of germlineless animals is warranted to focus on somatic roles for SET-26 and HCF-1, but statements like "major role in recruiting HCF-1 to chromatin in somatic cells, rather than in the germline" and " Contrarily, we find that HDA-1 binding to chromatin is more affected by loss of set-26 in whole worm samples than in somatic-only samples" make interesting hypothesis, but require additional data; I recommend softening these statements in the discussion.

Reviewer #1 (Remarks to the Author):

The authors set out to provide mechanistic insight into how *C. elegans* SET-26 and HCF-1, two conserved chromatin-associated proteins, regulate gene expression and modulate longevity. The premise is that the two proteins physically interact, although data to support this claim is not provided. The authors show widespread binding of these factors to chromatin, and an overall reduction in HCF-1 binding in *set-26* mutants. They also show common up and down-regulation of a subset of genes. Loss of the class I HDAC *hda-1* reduces the increased lifespan of both *set-26* and *hcf-1*. Based on these findings, they suggest that SET-26 recruits HCF-1 to regulate gene expression, and in particular mitochondrial function genes, to modulate longevity, and that HDA-1 opposed this activity.

Overall this study remains quite descriptive, the genome wide analysis does not go far enough in providing significant insight on the action of these widely expressed proteins on chromatin and their impact on longevity. The link to mitochondrial function and how it may impact the longevity phenotype is too speculative.

Specific points

1) this work originates from the purported observation that HCF-1 and SET-26 physically interact, or copurify in proteomics analysis. The authors state that this was shown in previous work (Rizki et al PloS Genetics), but I could find no trace of these experiments in the cited paper.

Assuming that the proteins were indeed found in the same proteomic experiment, the authors have all the tools (endogenously tagged proteins) to at least try to confirm it by co-IP. This is especially important as HCF1 is a notoriously sticky protein (<https://www.uniprot.org/uniprotkb/P51610/entry#interaction>) that copurifies with many chromatin associated complexes eg MLL, SIN3, NSL..., and SET-26 has also been found in several MS-IP experiments in *C. elegans* (eg <https://doi.org/10.2144/btn-2021-0075>, DOI: 10.1093/nar/gkz880) . The only evidence presented to support this physical interaction is the presence of an HCF-1 consensus motif in SET-26 (Figure S1) that is found in the human HCF1 interactor MLL5. In reference to this homology, what is the evidence that SET-26 is in fact the *C. elegans* homologue of MLL5? SETD5 appears to be more closely related than MLL5.

We thank the reviewer for pointing out that the proteomics experiments should be better documented in our manuscript. The IP-mass spectrometry study we cited from Rizki et al., 2011 did identify SET-26 as a putative HCF-1 interactor, however, the full list of HCF-1 putative interactors was not published in that paper, leading to confusion. In addition to the IP-Mass spec published in Rizki et al., we have also collaborated with another lab which conducted an additional independent second IP-Mass spec analysis using the same HCF-1::GFP-tagged strain (IU382.1 - *rwls9[phcf-1::hcf-1::gfp pmec-7::rfp]*). This second independent analysis also identified SET-26 as a putative interactor. We have now added the full list of putative HCF-1 interactors from both IP-mass spec experiments into Supplementary Data 1, and we have added a reference to this table in the manuscript with the following revised text:

“Analysis of HCF-1 binding partners in *C. elegans* through two independent immunoprecipitation-mass spectrometry (IP-Mass spec) experiments identified SET-26 as a

high confidence interactor (Supplementary Data 1), suggesting the two proteins could be part of the same complex.”

Additionally, we included details for both IP-mass spec experiments in the methods section. Due to the limited time available for revisions (considering the first author is leaving the lab in July 2023), and the substantial time required to perform co-IP analyses (first creating lines from recombination crosses to obtain both SET-26 and HCF-1 tags in the same strain and then optimizing a co-IP protocol which we are not currently using in our lab), we have not performed additional co-IP analyses. However, because we identified SET-26 as an HCF-1-interacting protein from two independent IP-mass spec analyses, we are confident that the interaction is likely to be real. As the reviewer pointed out, it is true that HCF-1 appears in many IP-mass spec experiments, however, evidence from human cells as well as worms suggests that HCF-1 could belong to several complexes at chromatin including both the COMPASS complex and HDAC complexes (Zargar & Tyagi, 2012; Wysocka et al., 2003).

Of the two references provided by the reviewer, the second (Beurton et al., 2019) identifies HCF-1 in a complex with CFP-1, which is also identified in both of our IP-mass spec experiments as an HCF-1 interactor, and with WDR-5.1. Although we did not identify WDR-5.1 as an HCF-1-interactor in our IP-mass spec analyses, literature from human cells suggests that HCF-1 could bridge the gap between the COMPASS complex (which contains WDR-5.1 in worms) and an HDAC complex (which in *C. elegans* contains CFP-1), thus the identification of HCF-1 in this IP-mass spec analysis is consistent with what is known about HCF-1 and does not by itself suggest that HCF-1 is likely to co-IP with unrelated complexes. The first reference mentioned by the reviewer (Baytek et al., 2022) shows the appearance of HCF-1 as an MRG-1 interactor. One of our two IP-mass spec experiments also identified MRG-1 as an HCF-1 interactor, further supporting the likelihood of this interaction. In our view, the appearance of HCF-1 on several IP-mass spec lists does not change the validity of our experiments, and in fact the large number of binding sites we identify for HCF-1 in the *C. elegans* genome increases the likelihood that it would have multiple protein interactors in the worm. Even if SET-26 and HCF-1 are part of a larger protein complex and do not directly bind to each other, the main conclusions of the paper remain unchanged in our view. However, to clarify that the interaction between SET-26 and HCF-1 will require additional validation in future studies, we have edited the following sentence in the discussion to include this information:

“It will be interesting in the future to further validate whether there is a direct binding relationship between SET-26 and HCF-1, and to determine whether the highly conserved “HCF-1 binding motif” identified in the human MLL5 protein²³ is also required for the interaction between *C. elegans* SET-26 and HCF-1.”

In reference to the homology between SET-26 and MLL5 or SETD5, it is true that SETD5 is another potential homolog for SET-26. We focused on MLL5 as the most likely homolog for SET-26, especially in this context, given that both proteins contain the HCF-1-binding motif which is absent in SETD5. While SETD5 and MLL5 both contain a similar SET domain to SET-26, which is thought to be mutated and potentially lacking methyltransferase activity (Li et al., 2023) SETD5 does not contain the PHD domain, which is present in both MLL5 and SET-26 (Li et al., 2023). Previous work from our laboratory revealed that the PHD domain in the *C. elegans* SET-26 is critical for binding the H3K4me3 histone mark (Wang et al., 2018), which is also true of the MLL5 PHD domain (Ali et al., 2013), thus making MLL5 a more likely functional homolog for SET-26 than SETD5. However, based on amino acid similarity, some SETD5 isoforms do have slightly higher similarity to SET-26 than MLL5 (maximum of 24.93% identical over 19% of SET-26 for SETD5 vs 23.70% identical over 16% of SET-26 for MLL5). Interestingly, a report

suggests that human SETD5 can actually also interact with human HCF1 in the context of hematopoietic stem cells (Li et al., 2022), thus both potential homologs of SET-26 show interactions with human HCF1. To address this in the manuscript, we have added the following two edits:

-“SET-26, the *C. elegans* homolog of MLL5 and SETD5,” and

-“Interestingly, SET-26 has two human homologs, MLL5 and SETD5, which both contain similar SET domains as SET-26^{24,25}. MLL5 appears to be more similar to SET-26 in its ability to bind H3K4me3 via its PHD domain²⁶. SETD5, which lacks a PHD domain²⁵, shares a similar degree of sequence similarity to SET-26 and has also been recently reported to interact with human HCF-1 in hematopoietic stem cells²⁷, indicating that both human homologs of SET-26 can interact with human HCF-1.”

2) Figure 2a shows that both *hcf-1* and *set-26* mutants are long lived (as previously published), and the double mutant is not longer lived than either single. This is interpreted as evidence of the two proteins acting together in longevity. While this may well indicate that both genes act in the same pathway, one can also imagine a scenario in which the non-additive effect is due to the two proteins independently targeting “longevity genes” so that their absence leads to misexpression beyond a threshold sufficient to increase longevity; increased misregulation (whether activation or repression) in the double mutant would not necessarily lead to a further increase in longevity. The interpretation of these results should therefore be nuanced.

We agree with the reviewer that longevity epistasis experiments are nuanced and the longevity results of the *set-26(-) hcf-1(-)* double mutant could be interpreted in alternative ways. We have revised the text to reflect this on page 5:

“This non-additive effect is consistent with the hypothesis that the two genes operate in the same genetic pathway to modulate lifespan. However, as it is possible that a non-additive phenotype could be observed in the double mutant due to alternative explanations, such as two separate longevity pathways being fully activated and unable to extend lifespan further, we pursued additional experiments to characterize the actions of SET-26 and HCF-1 directly at chromatin.”

Chromatin profiles:

3) tracks from cut&tag experiments seem quite noisy on IGV, which is quite surprising given that it is known to significantly reduce background compared to ChIP-seq. Do the authors have an explanation?

In our view, the CUT&RUN data are relatively clean when compared to ChIP-seq data in *C. elegans*. In our previous protocol paper (Emerson & Lee, 2022), we directly compared our CUT&RUN data for SET-9/26 to our lab’s previous ChIP-seq data for SET-9/26 using the same GFP-tagged strain and found that the CUT&RUN data were cleaner with less background noise, while similar peak regions were still identified, indicating the technique performed well (Figure 7b of Emerson & Lee, 2022, screenshot below).

[redacted]

Zooming in further, this is even more evident. For example, in the following two screen shots we are showing our CUT&RUN SET-26 binding profile captured in whole worms compared to the SET-26 binding profile captured in whole worms as obtained previously in our lab by Wang et al. (2018). Both data sets were processed in parallel and normalized to antibody background in untagged N2 worms, however it is evident that the resolution of the CUT&RUN data is much higher than the ChIP-seq data. Strong blocks of enrichment are detectable in the CUT&RUN data where this signal is weaker in the ChIP experiments, and the peak boundaries are clearer in the CUT&RUN data, often leading to the detection of higher confidence peaks in the CUT&RUN compared to ChIP-seq.

More than 13,000 SET-26 “peaks” are detected and more than 9000 HCF-1 “peaks”....this is a surprisingly large number.

We agree with the reviewer that the number of peaks identified for SET-26 and HCF-1 are large, likely indicating that these factors are present in many places throughout the genome. However, the number of peaks identified in our study is on par with other studies, as ChIP-seq profiles for mammalian HCF1 identify a similar number of peaks. Specifically, Minocha et al. (2019) identified 7,657 HCF1 peaks by ChIP-seq in mouse hepatocytes, and Michaud et al. (2013) identified 8,097 HCF1 peaks in HeLa cells. The number of peaks we identified for *C. elegans* HCF-1 is slightly higher (9,961), which is likely due to the use of whole worms in our study, which would reveal the HCF-1 binding profiles in a mixture of different cell types.

How does the number of peaks detected here compare to previous ChIP-seq which detected SET-26 and SET-9?

When analyzed in parallel, our lab's ChIP-seq data for SET-9/26 binding (Wang et al., 2018) compared to antibody background identified 4,304 SET-9/26 peaks, 73% of which are also identified in the current CUT&RUN study (see Venn Diagram of overlap below).

* Please note that the numbers represented in the center are not the exact number of peaks for each profile being compared, as the number of peaks that overlap differ between profiles (e.g. 2 ChIP peaks may overlap 1 CUT&RUN peak).

While the number of peaks identified by the previous ChIP-seq is much lower than our current CUT&RUN analysis, we believe this is likely due to the higher resolution data produced by CUT&RUN, as discussed above, which allowed us to uncover additional peak regions. For example, in the below screenshot we are displaying SET-26 binding in CUT&RUN vs ChIP-seq as mentioned in the previous screenshot. In the CUT&RUN track, many more peaks are identified in this region compared to the ChIP-seq tracks. Although both techniques show an enrichment for SET-26 binding in the center region, which is amplified by the second screen shot below, CUT&RUN is able to capture a much larger part of this region as bound by SET-26 compared to ChIP-seq, which is not able to identify much of this region as statistically significantly bound by SET-26 compared to background. Thus, we were able to identify many more peaks via CUT&RUN compared to our previous ChIP-seq analysis.

We do not see the same phenomenon of increasing peak number for every factor or target that we examined, however, indicating that the technique is not over-identifying peaks for every profile.

For example, in our lab's 2018 papers we identified 5,996 peak regions enriched for the histone mark H3K4me3 using N2 larval stage 4 worms (Wang et al. 2018), 6,104 H3K4me3 peaks using *glp-1(e2141)* mutants at the L3 stage, and 7,668 H3K4me3 peaks using *glp-1(e2141)* adults (days 2 and 12) (Pu et al., 2018). In a similar experiment using CUT&RUN to profile H3K4me3 in day 1 adult worms (unpublished data), we obtained 6,573 peaks, a relatively similar number to the ChIP-seq data. The tracks are also similar, as shown in the screenshot below, which compares H3K4me3 in CUT&RUN to the most biologically similar sample, ChIP-seq of L4 N2 worms from Wang et al. (2018).

[redacted]

Importantly, we also found a similar number of peaks between experiments when we processed our HDA-1 CUT&RUN data (in the germline-less *glp-1(e2141)* background) alongside HDA-1 ChIP-seq data (in the germline-less *glp-4(bn2)* background) produced by Shao et al. (2020). Specifically, our CUT&RUN data called 12,112 peaks and Shao et al.'s ChIP-seq data called 12,114 peaks for somatic HDA-1. The tracks are again similar (see screenshot below).

Overall, our analyses indicate that our recent CUT&RUN data allow us to obtain a more in-depth look at the profile of SET-26, which was less-well defined based on our earlier ChIP-seq analysis, while still correctly identifying the binding profiles of several other factors that have been well-defined previously.

The authors show a large degree of overlap between HCF-1 and SET-26 peaks. Given the large number of “peaks”, I expect the authors may see a similar overlap with many other chromatin bound factors as well. Did they compare their list of peaks to other factors whose profiles are available through ModEncode? Did the authors check how these overlap with high-occupancy target (HOT) regions? And did they filter out regions from the available black list available through Encode?

We thank the reviewer for pointing this out. We agree that, given the large number of peaks identified for both SET-26 and HCF-1, they are relatively ubiquitous factors that are likely to overlap with many factors, such as transcription factors, that are present at promoter regions. To test this, we processed ModENCODE data for a set of arbitrarily selected transcription and chromatin factors: ALR-1.2, CEH-44, DPFF-1, HLH-30, HMG-1.2, and KLU-1. When we asked how many peaks obtained from processing these unrelated factors overlapped with somatic SET-26 and HCF-1 peaks by 1bp or more, we found that all factors showed a significant overlap with SET-26 and HCF-1 (see below). While SET-26 and HCF-1 show a significant overlap with

several transcription factors, they are still likely to have an important relationship with each other at chromatin considering the major role we identified for SET-26 in the patterning of HCF-1 at chromatin (Figure 3), wherein the majority of HCF-1 localization at chromatin was disrupted in the *set-26(-)* germline-less mutant. An important test of specificity would be to ask whether loss of SET-26 or HCF-1 also disrupted the recruitment of the above mentioned transcription factors that share overlapping binding sites, however this question is outside the scope of the current study. Of note, since we do not see major defects in somatic HDA-1 recruitment to chromatin in the *set-26(-)* or *hcf-1(-)* mutants, we think it is unlikely that loss of SET-26 uniformly affects recruitment of chromatin factors at binding sites it shares, suggesting there is an important and unique relationship between SET-26 and HCF-1. Considering the data we compared were all derived from whole worms, in future experiments beyond the scope of the current study, it will also be interesting to determine whether the large overlap between SET-26 and HCF-1 and the other transcription factors occur in all cells or whether they occupy the same regions of DNA in different cell types.

* Please note that the Venn diagram is not to scale, and the numbers represented in the center are not the exact number of peaks for each factor being compared, as the number of peaks that overlap differ between factors (e.g. 2 HCF-1 peaks may overlap 1 SET-26 peak).

Our data indicate that SET-26 and HCF-1 are highly ubiquitous factors at open chromatin regions and are enriched at the TSS. Our past data indicated that SET-26 is an H3K4me3 reader and overlaps with H3K4me3 peaks, consistent with this idea. We wanted to verify the overlap with H3K4me3 using our new CUT&RUN data, and also asked if SET-26 and HCF-1 overlap generally with open chromatin regions (defined by accessible ATAC-seq peaks), as one would expect based on the high level of overlap between SET-26 and HCF-1 with other transcription factors and features of active chromatin. To obtain the data closest to our samples, we re-processed our lab's previous H3K4me3 ChIP-seq data using germline-less day 2 adult *glp-1(e2141)* worms (Pu et al., 2018), and called peaks in the same manner as we do for SET-26 and HCF-1. We also downloaded the young adult *glp-1(e2144)* ATAC-seq data published by the Ahringer Lab (Jänes et al., 2018), processed that, and compared accessible regions to our SET-26 and HCF-1 profiles. The result is below.

We found that, as expected, SET-26 and HCF-1 widely overlap with both H3K4me3 and ATAC-seq peak regions (with each pairwise comparison being statistically significant), indicating that they are factors that are generally highly prevalent in active chromatin regions, as the reviewer pointed out. Therefore, to clarify that SET-26 and HCF-1 are broad factors that overlap many active chromatin regions, we have added the following text to the manuscript:

“Interestingly, we found that SET-26 and HCF-1 bind to thousands of regions in the genome (with 13,422 and 9,961 peaks called, respectively), indicating that the factors are highly ubiquitous across the genome. As expected for factors enriched preceding TSSs, SET-26 and HCF-1 profiles largely overlapped with active chromatin, including regions marked by H3K4me3³⁵ and accessible regions identified through ATAC-seq³⁶ (Supplementary Fig. 1h; Supplementary Data 2) which were identified at similar ages in germline-less worms. This indicates that SET-26 and HCF-1 are both highly prevalent factors at active chromatin, and likely also interact with many additional factors that also bind to these sites.”

We also checked whether SET-26 and HCF-1 binding was affected by HOT regions (Chen et al., 2014) or blacklisted regions (Amemiya et al., 2019), as suggested. We found that SET-26 and HCF-1 binding did overlap with HOT regions (see below), which is not surprising given their high number of binding sites and high overlap with other transcription factors.

Importantly, however, even after filtering out both blacklisted genes and HOT regions, the overlap between SET-26 and HCF-1 was still strong (see below) and very significant ($p < 1 \times 10^{-15}$ by hypergeometric test).

This information has also been added to the manuscript as follows:

“Importantly, we found that SET-26 and HCF-1 profiles were highly overlapping even after excluding blacklisted regions³² and highly occupied target (HOT) regions³³ (Supplementary Fig. 1g; Supplementary Data 2), which are regions commonly identified as peaks across many genome-wide binding profiles for transcription factors^{33,34}”

I also don't understand the rationale for using histone H3 to normalize, but an expert may be a better judge of the methodology here.

We thank the reviewer for pointing out that the rationale to normalize to H3 was unclear. The main purpose of normalizing to H3 was so that we could perform a parallel immunoprecipitation (IP) within the same sample and normalize our factor binding to an independent IP. This was to control for any differences in IP itself that could occur between strains, since we were attempting to compare the relative enrichment of factor binding (SET-26, HCF-1, or HDA-1) in mutants versus controls. We felt that this comparison was more appropriate than performing peak calls for these comparisons either with antibody background tracks (i.e. IP performed in a separate genotype of untagged strains) or with no background tracks. Since we did not see major differences in H3 profiles between our mutants and controls, we felt more confident about our factor comparisons because they were done in reference to these unchanging H3 controls. However, to ensure that the H3 normalization did not unduly influence our results, when we conducted our CUT&RUN experiments using the *glp-1(-);hcf-1::gfp::3xflag* and *glp-1(-);set-26(-)hcf-1::gfp::3xflag* strains (Supplementary Figure 3f-h), we had also included parallel controls of antibody background, like the normalizations we used for initially identifying binding sites for our factors in Fig 1 (i.e. FLAG antibody in either single *glp-1(-)* mutants or *glp-1(-);set-26(-)* mutants). The figures below show our results when we ran our differential analysis for HCF-1 binding in the *glp-1(-);set-26(-)* mutant using either H3 normalization (left, as shown in Supplementary Figure 3h), antibody background normalization (middle), or no normalization tracks (right). Using all three approaches, we obtained similar results, overwhelmingly indicating that a large fraction of HCF-1 binding sites are lower in the *set-26(-)* mutant, increasing our confidence in our initial result and our normalization method.

To clarify our rationale in the manuscript, we have added the following sentence to the text: “By surveying H3 in parallel within each genotype, we were able to use H3 as a type of internal control for the immunoprecipitation process itself within each genotype.”

Henikoff described a peak caller specific for Cut&Run analysis : SEACR(Meers MP, Tenenbaum D, Henikoff S. (2019). Peak calling by Sparse Enrichment Analysis for CUT&RUN chromatin profiling. Did the authors compare MACS2 to this method to see whether they obtain a similar number of peaks?

We thank the reviewer for this very good suggestion. We attempted to use SEACR to analyze our CUT&RUN data, but surprisingly found that the peak caller did not perform well for our datasets. As shown below, using SEACR for our somatic SET-26 profile compared to antibody background, we only obtained 4 peaks (top panel), however the regions identified as peaks spanned 3-6kb (middle panel) and in our opinion, did not accurately capture the binding profiles as seen in our data. The reason for this is unclear, as it is evident from the bedgraph tracks for SET-26 and antibody background (bottom panel), which were directly used as input for SEACR, that the enrichment of SET-26 binding is clearly present over background levels in regions that are not identified as peaks by SEACR, but which are identified as peaks by MACS2. One possible explanation for this discrepancy could be that MACS2 is more highly equipped to deal with correction for local background levels compared to SEACR, which expects a large fraction of background to be 0s (Meers et al., 2019). Although in our opinion, our data are higher resolution than our previous ChIP-seq data in *C. elegans*, it is possible that the data are still noisier than standard CUT&RUN data, which is typically produced in mammalian cell cultures. As this publication represents one of the first times in which CUT&RUN has been successfully used in *C. elegans*, it is entirely possible that the data are different enough from the mammalian CUT&RUN that SEACR does not work well.

To further explore whether the peaks called by MACS2 are an accurate representation of the data, we also called peaks with an additional peak caller often used for ChIP-seq analysis, HOMER. We found that HOMER was able to call peaks similarly to MACS2, in a manner that much better represented our data than SEACR. Using the findPeaks command with the parameters `-style factor -minDist 100 -fdr 0.01`, HOMER identified 5,045 peaks for somatic SET-26 binding versus antibody background and 3,782 peaks for somatic HCF-1 binding versus antibody background. While the number of peaks called by HOMER are fewer than by MACS2, both peak callers show that SET-26 and HCF-1 bind to many thousands of regions. Additionally, looking closer at the individual peak calls, we see that MACS2 and HOMER call peaks in similar regions as expected (a and b below), but HOMER occasionally misses calling peaks in regions that are called by MACS2 and visually appear to be true peaks (arrow in c and d below). Based on a visual inspection of the data from all three peak callers, we believe peaks called by MACS2 are the most accurate representation of the data.

4) In Fig 1e the zoom is on a region of over 80 kb, with SET-26 binding a region of 20kb. It is difficult to reconcile this with gene specific regulation, or promoter peaks centered around the ATG shown in 1b. The authors should zoom in on a smaller region, as in figure S3.

We thank the reviewer for pointing out the discrepancy in Fig 1e and we agree that zooming in on a smaller region would be more appropriate. We have replaced Fig 1e with a smaller region as requested, now covering approximately 4kb, as shown here:

Also, peaks appear between the large blocks, but these are not called by MACS2. Can the authors explain?

We agree with the reviewer that the former Fig 1e, which included a screen shot covering over 80kb, a region appeared in which there was abundant HCF-1 binding, but no peak call (see green boxed region below). Upon further zooming into this region, however, it becomes clear that there are interruptions of HCF-1 binding within this region that prevent MACS2 from calling the entire region as a peak. Specifically, the HCF-1 track shows regions of depletion as well as enrichment within the larger overall region of enrichment, thus the overall region would likely not reach the statistical threshold for significance required by MACS2.

5) the authors previously showed SET-26 to be a H3K4me3 reader. How do the binding profiles shown here overlap with H3K4me3, which is mostly confined to promoters?

We thank the reviewer for this thoughtful question. To answer this question (as discussed above), we re-processed our lab's previous H3K4me3 ChIP-seq data from germline-less day 2 adult *glp-1(e2141)* worms and called peaks in the same manner as we do for SET-26 and HCF-1. We then asked how many peaks overlapped between H3K4me3, somatic SET-26, and somatic HCF-1 by 1bp or more. The results are below. We found that 51% of SET-26 peaks (6,781 out of 13,422) overlapped with H3K4me3 peaks, while 65% of HCF-1 peaks (6,477 out of 9,961) overlapped with H3K4me3 peaks. Interestingly, of the common SET-26 peaks that overlapped HCF-1, the vast majority (77%, 6,216 peaks) also overlapped with H3K4me3. Conversely, 68% of H3K4me3 peaks overlapped with SET-26 peaks (6,067 out of 8,959), while 66% of H3K4me3 peaks overlapped with HCF-1 peaks (5,940 out of 8,959). Using a similar analysis of 1bp overlap from SET-9/26 ChIP-seq data, Wang et al. had identified that 75% of SET9/26 peaks overlapped with H3K4me3, while 49% of H3K4me3 peaks overlapped with SET-9/26 (Wang et al., 2018). Although the numbers are slightly different, likely because we

identified additional SET-26 peaks that Wang et al. was not able to uncover using ChIP-seq, our analysis again confirms that the majority of SET-26 peaks, especially those common with HCF-1, overlap with H3K4me3.

* Please note that the Venn diagram is not to scale, and the numbers represented in the center are not the exact number of peaks for each factor being compared, as the number of peaks that overlap differ between factors (e.g. 2 HCF-1 peaks may overlap 1 SET-26 peak).

We also annotated the genomic features overlapping with the day 2 adult *gfp-1(-)* H3K4me3 data to compare the overall features bound by H3K4me3, SET-26, and HCF-1. The results are below. We found, as expected, that the annotations were highly similar between H3K4me3, SET-26, and HCF-1, with 85% of H3K4me3 peaks overlapping with promoters, 73% of SET-26 peaks overlapping with promoters, and 83% of HCF-1 peaks overlapping with promoters. Because the method of annotation counts peaks multiple times if they overlap with more than one feature (see specific point 5 part 2 below for further explanation), we are able to see that even H3K4me3 peaks are not solely confined to promoters, as also examined by Pu et al. (2018).

5) According to Fig 2D the expression of the large majority of genes bound by both HCF-1 and

SET-26 (8379) is not altered in either mutant. Where are HCF-1 and SET-26 binding in these 8379 regions?

We thank the reviewer for this helpful suggestion. To answer this question, we separated out the somatic peaks from either SET-26 or HCF-1 corresponding to genes which change in expression in the *glp-1(-);set-26(-)* mutant (for SET-26 profile) or *glp-1(-);hcf-1(-)* mutant (for HCF-1 profile) at day 1 of adulthood in our RNA-seq data. We compared the annotation of these two types of peaks, those NOT associated with genes that change in RNA expression vs those that do correspond to genes with RNA expression changes in the mutants. The annotation of the total HCF-1 or SET-26 peaks are also included for comparison. The results are below. We found that the percentage of peaks that overlapped with promoters was similar between HCF-1 or SET-26 peaks overall (83% or 73%, respectively) and HCF-1 or SET-26 peaks that did NOT change in RNA expression (81% or 71%, respectively). Interestingly, 100% of the peaks corresponding to genes that did change in expression, however, overlapped with promoters. Therefore, SET-26 and HCF-1 peaks are highly enriched for promoters even for those not associated with genes that change in expression, and are entirely within annotated promoters for those associated with genes with expression change.

Suppl Figure 1h shows binding just about everywhere (exons, introns, promoters, downstream of promoters...). It looks like these factors are not enriched at regulatory regions, while fig 1c shows that the majority of peaks are upstream of the ATG. I found this confusing

We thank the reviewer for pointing this out. We realize that the method of creating Fig 1c and Supplementary Figure 1h were not suitably explained and this likely led to confusion. Fig 1b-c are plotting the number of peaks within 3kb of TSSs, thus the large peak before the TSS in these figures represents that of the peaks that are within 3kb from a TSS, the majority are bound right before the TSS. Supplementary figures 1e-f and h-l on the other hand, represent the percentage of SET-26 or HCF-1 peaks that overlap (by 1bp) with each genomic feature. With the current settings, peaks are counted multiple times if they overlap multiple features, which is

why the percentages on the y axis add up to over 100. For instance, if a peak overlaps with the promoter and also extends into the first exon, then it will be represented in both the promoter and exon category in this chart. We often see that our peaks overlap with multiple feature types, for example, in the following 3 screenshots, we see that SET-26 and HCF-1 overlap with promoters and extend into the gene body as well. Furthermore, from looking at Fig1b-c, we can also see that, although the highest number of SET-26 and HCF-1 peaks are present preceding the TSS, there are still a very high number of peaks that overlap with the TSS itself and immediately after the TSS – these would again be represented in both the promoter and exon categories. As discussed above, the H3K4me3 profile looks similar when annotated this way as well.

To clarify this in the text, we have added the following:

In Supplementary Information:

Under Supplementary Fig 1 e-f and h-i, and Supplementary Fig 5 e and g. A single peak may be annotated to multiple features if it spans more than one feature.

We also slightly edited the sentence describing this result in the main text to read, “SET-26 and HCF-1 peaks were both highly overlapping with promoter regions in the *C. elegans* genome

(Supplementary Fig. 1e, f; Source Data), with high enrichment immediately upstream of the transcription start site (TSS) (Fig. 1b, c)”

How does this compare to mammalian HCF1, which is mostly restricted at promoter regions?

In our CUT&RUN data, we identify 9,961 HCF-1 peaks, and we find that 83% of HCF-1 peaks overlap with a promoter region. In a ChIP-seq analysis of HCF1 genome-wide binding sites in mouse hepatocytes, Minocha et al. (2019) uncovered 7,657 HCF1 peaks, and 74% of these peaks were found to be immediately up or downstream of transcription start sites. Similarly, Michaud et al. (2013) found 8,097 HCF1 peaks in HeLa cells, of which 67% were annotated to promoter regions. Overall then, our analysis reveals that *C. elegans* HCF-1 overlaps with promoters similarly, albeit a bit more frequently, than mammalian HCF1.

6) The cut & run experiments suggest that SET-26 is required for most of HCF-1 recruitment to chromatin in the soma. If this is the case wouldn't you expect very similar phenotypes between *hcf-1* and *set-26* mutants? *hcf-1* mutants suppress synMuv phenotypes and ectopic expression of germline genes in the soma...as do chromatin factors *mrg-1*, *sin-3*, *dpy-30*. Does the *set-26* mutant share any of these phenotypes?

Given that HCF-1 copurifies with several chromatin associated factors it hard to imagine that SET-26 is responsible for most of its recruitment throughout the genome. This raises the question that the observed effect may be completely indirect, for example SET-26 alters the expression of another protein or chromatin mark that is required to stabilize HCF-1 on chromatin..

We thank the reviewer for asking these important questions. Based on our data, we do expect similar, although not necessarily identical, phenotypes for *set-26(-)* and *hcf-1(-)* mutants. Indeed, based on our past experimental data, we know that both *set-26(-)* and *hcf-1(-)* mutants are long-lived, as discussed in our paper, and both mutants are also resistant to heat stress (Li et al., 2008; Wang et al., 2018). As the reviewer pointed out, *hcf-1(-)* mutants suppress the synMuv phenotype (Cui et al., 2006). We also find that *set-26(-)* mutants are able to strongly suppress the synMuv phenotype caused by *lin-15b* RNAi (unpublished data, results below). When exposed to *lin-15b* RNAi, which knocks down both *lin-15a* and *lin-15b*, *eri-1(-)* mutants, which are hypersensitive to RNAi, show a highly penetrant multivulva phenotype (Guang et al., 2008), with an average of 91% of worms showing multivulva from our three trials. In *set-26(-) eri-1(-)* mutants however, an average of only 30% of worms show the multivulva phenotype, indicating that *set-26(-)* can suppress synMuv similarly to what was previously found for the *hcf-1(-)* mutant.

Data displayed are the average of 3 independent biological replicates with error bars representing SEM. ***, $p < 0.001$, two-tailed student's t-test.

Although we do see similar phenotypes for our mutants, we do agree with the reviewer that it is entirely possible that the effect of SET-26 depletion on HCF-1 recruitment to chromatin could be indirect. To make this clearer in our text, we have changed the previous sentence in the discussion of “While we have not completely ruled out an indirect effect caused by loss of *set-26*, it is tempting to speculate that SET-26 may directly recruit HCF-1 to chromatin” to the newly revised sentence, “Importantly, we have not ruled out an indirect effect caused by loss of *set-26*, and it is entirely possible that loss of *set-26* somehow alters the chromatin environment indirectly in a manner that prevents proper HCF-1 recruitment to chromatin. However, it is tempting to speculate that SET-26 may directly recruit HCF-1 to chromatin.”

8) RNA-seq -analysis was only carried out on 2 samples/genotype. Although the PCAs look ok, the fact that collagen genes are enriched suggests lack of synchronization, which could impact results.

Although enrichment of collagen genes can sometimes indicate a lack of synchronization, collagen genes also play important roles in aging itself (Ewald et al., 2015; Morikiri et al., 2018; Palani et al., 2023), thus the enrichment of collagen genes alone to our mind does not necessarily indicate a lack of synchronization. Additionally, we still see an enrichment of collagen genes in upregulated RNA-seq genes in germline-less *set-26(-)* mutants compared to controls at day 3 of adulthood as shown below.

In our view, the persistent upregulation of collagen genes in *set-26(-)* mutants makes it less likely that the enrichment of collagen genes could be due to lack of synchronization and more likely that it is due to a real biological difference between the mutants and controls. Additional evidence that collagen genes identified could be important for our lifespan phenotype comes from our Fig 7a-c (copied below), which identifies collagen genes as differentially impacted in day 12 adults in *set-26(-)* and *hcf-1(-)* mutants aged on empty vector compared to *hda-1* RNAi. Overall, it is our belief that the overrepresentation of collagen genes in our RNA-seq is likely related to a real biological phenotype of our mutants and does not represent a quality control problem.

7) Interaction with HDA-1:

The authors again cite previous work showing HDA-1 was identified with HCF and SET-26 in mass spec experiments. They should validate the interaction by co-IP. An overlap in binding to

chromatin does not necessarily show physical interaction or a common function. The authors should provide additional evidence to sustain the claim that HCF-1, SET-26 and HDA-1 physically interact.

As described above for the interaction between SET-26 and HCF-1, we have added in the two independent IP-Mass spec experiments conducted by our collaborators for *C. elegans* HCF-1 interactors, both of which identified HDA-1 as potential interactors. Similarly to above, due to time and technical limitations, we were not able to perform co-IPs for HDA-1 with SET-26 and HCF-1, however we feel confident that since HDA-1 was identified in two independent IP-Mass spec experiments that the interaction is likely to be real. However, we do agree that an overlap in chromatin binding does not prove a physical interaction or common function, and we tried to be mindful when writing the manuscript to not over-emphasize a physical interaction. For example, in our discussion, instead of concluding from our CUT&RUN that the factors have a physical interaction or common function we have written, “All three factors have similar binding profiles at chromatin, suggesting they could regulate similar genes”. Because of our RNA-seq, we do hypothesize that SET-26 and HCF-1 work cooperatively to regulate gene expression and are opposed by HDA-1, but we do not conclude a common function necessarily from our CUT&RUN data.

It is also surprising that published HDA-1 ChIP-seq tracks (ModEncode) show better resolution than HDA-1 cut&tag.

When we processed the ModEncode HDA-1 ChIP-seq data in parallel to our CUT&RUN for both whole worms (using an N2 background) and somatic samples (using *glp-1(-)* mutants) as well as existing somatic HDA-1 ChIP-seq data (in the *glp-4(bn2)* mutant background) from Shao et al. (2020), we found that all four tracks were highly similar overall, as evidenced by the first screen shot below showing a zoomed out view of many similar peak regions identified from all 4 analyses. The number of peaks called between the samples was slightly different (14,904 peaks called in modENCODE data, 12,114 peaks called in the Shao et al. data, 8,332 peaks for our whole worm CUT&RUN, and 12,112 peaks for our somatic-CUT&RUN). In general, our HDA-1 CUT&RUN data from whole worms was noisier and identified fewer peaks than our somatic-only CUT&RUN data. We aren't sure the reason for the noise in our whole worm HDA-1 data, however we hypothesize it could have to do with the high level of HDA-1 expression in the germline, where HDA-1 could have separate targets from somatic cells and could increase noise in the data given that we used young adult worms for our analysis. Neither the somatic-only data (from our analysis or Shao et al.), or the ModEncode data (which uses L3 larvae) would run into this problem as prominently, due to the absence or underdevelopment of the germline, respectively. While the number of peaks called for our whole worm CUT&RUN was less than the modENCODE data, the number of peaks called for our somatic-only samples was extremely similar to the number of peaks identified by processing the HDA-1 ChIP-seq data from Shao et al (12,112 vs 12,114), which is a fairer comparison given that both samples were germline-less and young adults.

However, even though our whole worm CUT&RUN data was noisier than our somatic samples, we were able to identify several HDA-1 peaks in our whole worm CUT&RUN data that were missed in the ModEncode data (see second screenshot below). We believe these are likely to be real HDA-1 peaks given the presence of peaks in both the Shao et al. ChIP-seq and our somatic-only CUT&RUN. Thus, in some regions, our CUT&RUN data is higher resolution than the modENCODE data, even while the modENCODE data identified more peaks overall.

RNA-seq on *hda-1*(RNAi) : the authors take animals at day 12 of adulthood because they claim that at day 3 no changes in gene expression were observed. This is very surprising. What is the evidence that *hda-1*(RNAi) is working? Fig S7 only shows a 50% decrease in *hda-1* RNAi.

We agree with the reviewer that our RNA-seq results indicating that gene expression is largely unchanged at day 3 of adulthood when *hda-1* RNAi is initiated at day 1 of adulthood is very surprising. We agree that the 50% decrease we show in total HDA-1 total protein levels by western blot in Supplementary Fig 7 appears relatively mild. However, taking into consideration that the worms used for this experiment are germline-less *glp-1*(-) mutants, of which approximately 1/3 of the total cells of the animal are neurons which are generally considered refractory to RNAi in *C. elegans* (Timmons et al., 2001; Kamath et al., 2001), it is our opinion that this 50% decrease is more substantial than it appears. To emphasize this, we used our HDA-1::GFP::HA-tagged strain to visualize HDA-1 knockdown in the *glp-1*(-) mutant background in two different regions – intestinal cells and heads. When we initiated *hda-1* RNAi at day 1 of adulthood in these *glp-1*(-);*hda-1*::*gfp*::*ha* worms and observed GFP levels at day 3 of adulthood, we saw a very substantial decrease in GFP levels in intestinal cells of worms exposed to *hda-1* RNAi versus those on empty vector (E.V.), representing a decrease of around 90% on average. When we surveyed the GFP fluorescence in the head region however, primarily composed of neurons which are refractory to RNAi, we did not see a statistically significant decrease in HDA-1 levels as would be expected, with an average fluorescence decrease of around 20% on *hda-1* RNAi.

Therefore, we believe that the actual knockdown in the cell types most likely to be relevant for our phenotype (i.e. cells not refractory to RNAi) is likely to be higher than 50% by day 3 of adulthood. This information has been added into Supplementary Fig 7, and we have revised the text to include it as follows:

“Surprisingly, we found that gene expression was largely unchanged at day three of adulthood, after two days of *hda-1* RNAi (Supplementary Fig. 7c; Supplementary Data 3). This is despite the marked decrease in HDA-1 protein levels induced at this timepoint, with a decrease of approximately 87% in the wildtype background and 48% in the germline-less background (Supplementary Fig. 7d,e; Source Data). Although the knockdown in the germline-less background was weaker than the wildtype background when surveying total protein levels in whole worms by western blot, this was likely due to the high proportion of neurons (roughly one third of somatic cells) in the germline-less mutant, which are resistant to RNAi^{46, 47}. Indeed, while intestinal cells from germline-less mutants showed a dramatic reduction of HDA-1::GFP::HA levels by day 3 of adulthood after two days of *hda-1* RNAi (Supplementary Fig. 7f,g; Source Data), the GFP levels in the head were not significantly decreased by *hda-1* RNAi (Supplementary Fig. 7h,i; Source Data).”

In general the reasoning in this whole section is very confusing.

To clarify the rationale for selecting the day 12 timepoint, we added the following:

“We chose day 12 as a late-stage timepoint because it represents a time just before a major wave of death in germline-less controls (Fig. 5b), and because age-related chromatin and RNA expression changes in germline-less *glp-1*(-) mutants have been extensively profiled by our lab at day 12 of adulthood^{35,44,45}. As expected, we uncovered many of the same age-related gene expression changes as our lab previously identified³⁵ (Supplementary Fig. 7a,b).”

To clarify our rationale in examining *hda-1*-dependent genes unique to *set-26(-)* and *hcf-1(-)* mutants, we rephrased the following sentences:

“To focus on the gene expression changes most likely to be relevant to the lifespan phenotype, we looked for *hda-1*-dependent gene expression changes unique to *set-26(-)* and *hcf-1(-)* mutants. We reasoned that since *hda-1* RNAi shortens the lifespan of *set-26(-)* and *hcf-1(-)* mutants and not controls, the genes that change in response to *hda-1* RNAi in *set-26(-)* and *hcf-1(-)* mutants and not controls would be most likely to explain the requirement for *hda-1* in *set-26(-)* and *hcf-1(-)* mutant longevity. We identified 813 genes total that were significantly differentially expressed in either *set-26(-)* or *hcf-1(-)* mutants on *hda-1* RNAi compared to empty vector.”

We also added additional details about the *hda-1*-dependent gene expression changes identified in our data as requested by other reviewers to clarify the details of this section.

8) additional experiments could support the claim that binding to mitoUPR genes is relevant. The authors show increased expression of mitoUPR genes *hsp-6* and *hsp-60* in *set-26(-)* and *hcf-1(-)* young adults. Expression of the widely used reporters *hsp-6::GFP* and *hsp-60::GFP* should be used to show that mitoUPR is activated at baseline in these mutants, as they suggest.

hda-1(RNAi) results in decreased expression of *hsp-6::GFP* following induction of mitochondrial stress. Did the authors test whether SET-26/HCF-1 antagonism of HDA-1 in gene regulation extends to stress induced *hsp-6* expression? This would reinforce the model of opposing function in the regulation of mitochondria-related functions.

We thank the reviewer for this suggestion. We have examined the induction of *hsp-6p::gfp* in our *set-26(-)* and *hcf-1(-)* mutants as suggested, and found that, consistent with our RNA-seq, *set-26(-)* and *hcf-1(-)* mutants show elevated levels of *hsp-6p::gfp* (c-d below). To test whether HDA-1 could antagonize SET-26 and HCF-1 in mitochondrial stress-induced *hsp-6* expression, we grew controls, *set-26(-)* mutants, and *hcf-1(-)* mutants on *hda-1* RNAi from egg lay as described in Shao et al. (2020), the paper which originally found HDA-1 to be required for the induction of the mitochondrial UPR in *C. elegans*. Following this protocol, we found that the full induction of *hsp-6p::gfp* in *set-26(-)* and *hcf-1(-)* mutants at day 1 of adulthood did depend on *hda-1*, indicating that HDA-1 could antagonize SET-26 and HCF-1 over induction of the mitoUPR (c-d below). We performed control experiments as described by Shao et al. in parallel, and confirmed that *hsp-6p::gfp* induction caused by *atp-2* RNAi was reduced by *hda-1* RNAi initiated at egg lay (e-f below).

We also conducted preliminary experiments to test whether *hda-1* RNAi initiated at day 1 of adulthood as in our lifespan experiments could decrease the *hsp-6p::gfp* induction in *set-26(-)* and *hcf-1(-)* mutants during aging. Based on our preliminary results, we did not see a striking reduction of *hsp-6p::gfp* in aged *set-26(-)* and *hcf-1(-)* mutants on adult *hda-1* RNAi compared to empty vector, however as our results are preliminary and we are not sure what the precise timing of such a requirement would be, we cannot conclude definitively whether *hda-1* RNAi initiated at day 1 can reduce *hsp-6p::gfp* in our mutants. Our experiments detailing the reduction of *hsp-6p::gfp* in *set-26(-)* and *hcf-1(-)* mutants with *hda-1* RNAi initiated at egglay however, do support the idea that HDA-1 can play an antagonizing role to SET-26 and HCF-1 in mitoUPR induction, and further experimentation in future papers will need to determine whether or not this is relevant for lifespan. The experiments have been added to the text in Supplementary Figure 8, and the following text has been added to the paper:

“In line with this, we observed increased GFP expression of the classical mitoUPR reporter strain, *hsp-6p::gfp*, in *set-26(-)* and *hcf-1(-)* mutants as young adults (Supplementary Fig. 8c,d; Source Data). To test if HDA-1 is required for mitoUPR activation in *set-26(-)* and *hcf-1(-)* mutants at baseline, we initiated *hda-1* RNAi at egglay and observed decreased expression of *hsp-6p::gfp* in *set-26(-)* and *hcf-1(-)* mutants as young adults (Supplementary Fig. 8c,d; Source Data). As a control, we repeated the conditions followed by Shao et al., and observed as expected that *hsp-6p::gfp* activation induced by *atp-2* RNAi was decreased upon *hda-1* RNAi as expected¹³ (Supplementary Fig. 8e,f; Source Data).”

Reviewer #2 (Remarks to the Author):

This manuscript presents findings that establish a relationship between the chromatin-associated proteins SET-26 and HCF-1 with the histone deacetylase HDA-1. The authors build upon their previous studies of SET-26 and HCF-1 by combining insights from genomic approaches like CUT&RUN and RNA-seq with functional genetic tests to probe the interdependence of these chromatin proteins. Long-lived *set-26* mutants and *hcf-1* mutants

have similar mis-regulation of somatically-expressed genes, and a significant subset of these genes have both SET-26 and HCF-1 at their promoters, indicating that both proteins co-stabilize each other at specific gene promoters. Loss of HDA-1 suppresses the longevity of both mutants, and HDA-1 was found to bind many of the same promoters as SET-26 and HCF-1. However, HDA-1 binding in somatic cells was not affected by the loss of either chromatin-associated protein, indicating that HDA-1 is not involved in the longevity of set-26 mutants or hcf-1 mutants. Because the effects of hda-1 RNAi in either mutant has an opposite effect on gene expression, the authors propose a model where HDA-1 competes with SET-26 and HCF-1 to control gene expression.

Overall, this is a well-written paper that lays a solid groundwork for the authors' model of how somatic gene expression is mediated by the opposing actions of chromatin regulators. The experiments described in this manuscript are well-conceived and include important controls, including replication of their findings in glp-1 mutants (to corroborate RNAi results) and in wild-type animals with an intact germline. This work has critical implications for future mammalian studies, as it makes a strong case for investigating the roles of these chromatin-associated proteins in terminally differentiated cells (or even in vivo), rather than the current standard of using proliferating cell lines. In conclusion, this study address the mechanism of longevity in chromatin mutants, pinpointing their effect on lifespan to mitochondrial stress response genes, and will be of interest to those studying longevity, stress resistance, and how chromatin affects transcription.

The following are minor points provided as constructive feedback for the authors:

1. The relationships identified by peak overlap in the CUT&RUN and corroborated by effects on gene expression in each mutant condition are convincing. However, one drawback of genomic approaches like these is that they reflect changes in a population of cells. In this study, whole animals were used for the genomic approaches, which means that many somatic cell types are represented in the samples. Therefore, it is possible that HCF-1 and SET-26 do not directly co-occupy the same promoters, but rather that HCF-1 regulates those genes in certain cell types, while SET-26 regulates the same genes but in completely different cell types. (The loss of HCF-1 binding in set-26 mutants makes this alternative interpretation less likely, but it is still formally possible.) Although it is beyond the scope of this paper to experimentally distinguish between this model and the one presented by the authors, this possibility should be addressed in the discussion.

We thank the reviewer for pointing this out. We agree that our current approach is unable to distinguish between SET-26 and HCF-1 co-occupying the same promoters in the same or different cell types. Therefore, we have added the following sentences to the discussion section:

“Of note, our CUT&RUN approach utilized whole worms lacking germlines, thus our results are an aggregate of binding signal obtained from all surveyed somatic cell types. Therefore, our current approach is not able to distinguish whether SET-26 and HCF-1 co-occupy the same promoters in the same cell type, or whether they occupy the same promoters independently in different cell types. Given the IP-Mass spec interaction data between SET-26 and HCF-1 and the requirement for SET-26 in HCF-1 binding, it seems more likely that SET-26 and HCF-1 co-occupy the same promoters together in the same cell types, however future approaches will be required to test this in a tissue-specific manner.”

2. Figure 7 includes the interesting result that depletion of hda-1 in set-26 mutants or in hcf-1

mutants reverses some of the gene expression changes observed in the mutants alone. It would be helpful for interpreting these findings to know more details about the analysis. For example, what percentage of mis-expressed genes in the single mutants are suppressed by *hda-1* knockdown? Of the genes that are up-regulated in *set-26* mutants (or in both mutants), how many of these are now unchanged from control, or even down-regulated, in *hda-1* RNAi; *set-26* mutants? Similarly, when combining their CUT&RUN data with existing RNA-seq data from Shao et al 2020, it would help to have a sense of what is meant by overlap: how many genes were originally identified as HDA-1-dependent mitoUPR genes, and of those, how many had HDA-1, HCF-1, and SET-26 present at their promoters? Of this subset of genes with all three chromatin regulators bound at their promoters, what percentage were upregulated/downregulated in *set-26* mutants or *hcf-1* mutants, and how many of these misregulated genes (which mostly appear to be upregulated?) are now downregulated by *hda-1* knockdown?

We thank the reviewer for pointing out these gaps in our document. We have added additional details to clarify Figure 7. To address the first half of the questions, we have added the following paragraph:

“By day 12 of adulthood however, *hda-1*-dependent gene expression changes emerged (Supplementary Fig. 7j), and we therefore focused our analysis on these later-life gene expression changes. We identified 471 and 877 genes that changed in expression in *set-26(-)* and *hcf-1(-)* mutants respectively on *hda-1* RNAi versus empty vector. We found that of the 967 and 1,129 genes that are misregulated at baseline in *set-26(-)* and *hcf-1(-)* mutants compared to controls on empty vector at day 12 of adulthood, 114 (12%) and 204 (18%), respectively, were again significantly mis-regulated upon *hda-1* RNAi. Of these genes that are regulated by SET-26 and/or HCF-1 at baseline and are again changed by *hda-1* RNAi, the majority of genes exhibited an antagonistic change in expression such that the gene expression changes normally observed in the *set-26(-)* and *hcf-1(-)* mutants were reversed when aged on *hda-1* RNAi. Specifically, of the 114 genes that showed significant expression changes in the *set-26(-)* mutant (compared to controls) and also when *set-26(-)* mutants were treated with *hda-1* RNAi (compared to control RNAi), 70% (80 genes) show an antagonistic relationship, to the extent that 50 of these genes that are normally altered in *set-26(-)* mutants no longer reached significance in *set-26(-)* mutants treated with *hda-1* RNAi, and one gene reached significance in the opposite direction from its baseline change. Similarly, of the 204 genes that are differentially expressed in the *hcf-1(-)* mutant (compared to controls) and also when *hcf-1(-)* was treated with *hda-1* RNAi (compared to control RNAi), 68% (138 genes) show an antagonistic relationship, whereupon 83 of these genes were no longer significantly differentially expressed upon *hda-1* RNAi, and two genes reached significance in the opposite direction from their baseline change.”

To address the second half of the questions about our Shao et al. (2020) comparison, we have added additional details into the text in the following paragraph:

“Previous analysis from Shao et al. identified a group of 283 “HDA-1-dependent mitoUPR genes”, which are mitoUPR genes normally activated upon mitochondrial perturbation and the expression of which decreases on *hda-1* RNAi¹³. We intersected this gene list with our CUT&RUN datasets to obtain 121 “HDA-1-dependent mitoUPR genes” also bound by SET-26, HCF-1, and HDA-1 in somatic cells. We plotted the expression of these 121 genes (Fig. 7d; Supplementary Data 3) in our day 12 adult gene expression data set again at baseline and with *hda-1* RNAi during aging. Although only a small number of these genes (19 and 26) reached the threshold of statistical significance in *set-26(-)* and *hcf-1(-)* mutants versus controls at day 12 of adulthood, respectively, our heatmap revealed that the majority of these 121 genes tended to

subtly increase in expression in germline-less *set-26(-)* and *hcf-1(-)* mutants compared to controls, with 94 and 92 out of 121 genes having a fold change higher than one compared to controls in *set-26(-)* and *hcf-1(-)* mutants (median fold change of 1.37 and 1.45, respectively). When exposed to *hda-1* RNAi, the expression of these genes generally decreased in both *set-26(-)* and *hcf-1(-)* mutants, with 102 and 109 genes showing a fold change less than one on *hda-1* RNAi (median fold change of 0.75 and 0.79, respectively), although again the number of genes that reached statistical significance within this group was small (9 in *set-26(-)* mutants and 12 in *hcf-1(-)* mutants on *hda-1* RNAi). Although subtle, the overall trend is consistent with reduction of *hda-1* leading to a mild deactivation of mitoUPR genes in germline-less *set-26(-)* and *hcf-1(-)* mutants. Interestingly, we found that 27 of these direct targets that were “HDA-1-dependent mitoUPR genes” were the same genes identified in Fig. 4h that exhibited destabilized SET-26 and HCF-1 binding in the *hcf-1(-)* and *set-26(-)* mutants respectively and were accompanied by increased RNA expression in both *set-26(-)* and *hcf-1(-)* mutants at day 1 of adulthood (median fold change of 4.15 and 3.10 at day 1 of adulthood, respectively) (Fig. 7e; Supplementary Data 3).”

3. I am intrigued by the authors' model of how HCF-1 and SET-26 work together to counteract the activity of HDA-1, specifically at mitoUPR genes. However, they rely primarily on correlation to dissect the relationships of these chromatin regulators on gene expression – they did not functionally test this interaction at these loci. It is beyond the scope of this particular study to fully validate their model, but the model would be significantly strengthened if they demonstrated a change in chromatin at the promoters of these genes. For example: ChIP-qPCR against H3/H4 acetylation at candidate genes identified in this study in *set-26* mutants compared to *hda-1* RNAi; *set-26* mutants. Alternatively, the gene regulation activity of HDA-1, SET-26, and HCF-1 may independent of direct effects on chromatin state, in which case, future studies will be able to pursue different mechanisms.

We thank the reviewer for this thoughtful suggestion. We agree that it would be very interesting to survey histone acetylation at candidate genes in the *set-26(-)* and *hcf-1(-)* mutants with and without *hda-1* RNAi, and that this analysis would help to further the mechanistic understanding of the potential antagonistic interaction between HDA-1 with SET-26 and HCF-1. However, we agree with the reviewer that these experiments are beyond the scope of the current study, and we hope to pursue them in future studies to better understand the relationship, particularly at mitoUPR genes.

Reviewer #3 (Remarks to the Author):

Emerson et al. present a well-design study rich in data that describes how SET-26 and HCF-1 work in opposition of the role of HDA-1 in the regulation of gene expression that influences health and longevity. The authors develop a new model where SET-26 and HCF-1 stabilize each other on chromatin and balance the histone deacetylase activity of HDA-1 that results in a fine focus of gene expression; particularly in longevity-associated genes. Overall, this is an outstanding resource and I only I have one experimental suggestion and two suggestions for the text of the manuscript to make it more impactful. Enthusiasm for this work is very high and I believe it will be well-received and of great interest to the readership of Nature Communications.

Experimental suggestion:

The manuscript lacks measures of healthspan. Although careful detail is taken to study lifespan the manuscript would benefit if the different combinations of genetic and RNAi conditions included an assessment of health parameters (examples, not all are needed, include:

movement, intracellular lipid stores, pharyngeal pumping, etc.) at the time points measured in the study (e.g., Day 1 and Day 12).

We thank the reviewer for this thoughtful suggestion, and we agree that our study would benefit from a thorough analysis of healthspan in our mutants with and without *hda-1* RNAi during aging. Unfortunately, a thorough analysis across healthspan measures encompassing multiple measurements is beyond the scope of the current study, so we chose to conduct motility assays to survey if crawling differed in our different conditions. As a first attempt to assay whether the motility of germline-less *set-26(-)* and *hcf-1(-)* mutants differed from germline-less *glp-1(-)* controls at baseline (day 1) or during aging (day 12) on *hda-1* RNAi compared to empty vector (E.V.), we used the WMicrotracker SMART machine from PhylumTECH which captures 30 second videos to assay average speed, travelled distance, and rotation index of populations of worms. We used 2-4 plates of worms per condition, with at least 12 worms per condition per biological replicate and 2 independent biological replicates. To assess the significance between conditions, we used two-tailed student t-tests, and display any $p < 0.05$ with an * below. Error bars represent standard error. As shown below, the results unexpectedly revealed that at day 1 of adulthood, germline-less *set-26(-)* and *hcf-1(-)* mutants trended toward lower motility scores in average speed, travelled distance, and rotation index compared to controls (*glp-1(-)* single mutants), although only rotation index for *hcf-1(-)* mutants reached the level of statistical significance. By day 12, *hcf-1(-)* mutants had comparable motility scores to controls, while *set-26(-)* mutants still showed somewhat lower scores for average speed and travelled distance. Interestingly, worms aged on *hda-1* RNAi from day 1 of adulthood tended to have lower motility scores by day 12, although surprisingly this only reached the level of statistical significance when comparing the controls on empty vector to *hda-1* RNAi in travelled distance and rotation index.

Although these results are unexpected, it is true that healthspan and lifespan are not perfectly linked. Additionally, having assayed only one measure of healthspan at two timepoints, it is difficult to conclude the overall health status of our mutants. It is known from previous studies that both *set-26(-)* (Wang et al., 2018) and *hcf-1(-)* (Li et al., 2008) mutants have improved heat stress resistance as young adults, marking them as healthier in at least one measure. It will be interesting in future studies to determine whether the beneficial (i.e. lifespan extension, heat stress resistance) and undesirable (i.e. decreased motility) effects of *set-26* and *hcf-1* mutation occur through different mechanisms, for example utilizing distinct sets of target genes and/or operating in separate cell types. A more thorough investigation of healthspan with various measurements at different timepoints would greatly benefit our understanding of the relationship between SET-26, HCF-1, and HDA-1, however experiments of this size are outside the scope of the current study. As our current study is focused on the relationship between these factors in lifespan and at chromatin, we feel that our conclusions remain sound.

Text suggestions:

1. The supplementary data includes a large resource that is inadequately described in the text. I recognize that it is impossible to include everything in the text but expanding the results and discussion to include genes and gene families that are of particular importance (include the magnitude of change) would make the text more useful. The authors include statements about mitochondrial genes and cell cycle already, additional examples like this would enhance the text.

We thank the reviewer for this thoughtful suggestion. To make the text more useful, we ensured that each time a WormCat GO enrichment was mentioned in the text, at least 2-3 relevant biological categories were mentioned, along with the number of genes present within each biological category in our data and the median fold change in both *set-26(-)* and *hcf-1(-)* mutants for each biological category mentioned. This included the addition of several interesting biological categories, including stress response and proteolysis in the discussion for cluster 1 of Figure 7b, as well as transcription: chromatin structure, snRNAs, and mRNA binding for whole worm-enriched SET-26 and HCF-1 binding in Supplementary Fig. 10,p.

We hesitate to list individual gene names too often, as we have not individually tested many genes, and it is difficult to say which genes are of particular importance to our phenotype given the vast number of genes on which SET-26 and HCF-1 bind and the still hundreds on which they could influence expression. However, we do include details about *hsp-6* and *hsp-60* expression in Supplementary Fig 8a-b and have added the magnitude of change to the main text. Additionally, as Fig 7e shows an overlap of only 27 genes, we were able to pull out some genes that may be particularly interesting and include those in the text as follows: “These include the mitoUPR marker *hsp-6*, the mitochondrial fission factors *mff-2* and *drp-1*, the mitochondrial translocases *timmm-23*, *scpl-4*, and *tin-9.1*, and the translational elongation factors *gfm-1*, *tufm-1*, *tufm-2*, and *tsfm-1* (Supplementary Data 3).” Additionally, we added references to the supplementary data and source data into the main text alongside our figure references wherever relevant so that readers can easily access the relevant data if they desire.

2. Statements regarding enrichment and differential activity in the soma and germline need more support (only statements that directly compare soma to germline). The analyses in the context of a germlineless animal are adequately performed, but it is impossible in this experimental set up to account for physiological and cell non-autonomous changes to the animal in the absence of a germline. To make these conclusions what is needed is an analysis of isolated germlines. I am not proposing that this must be performed for the study to be published as there is already a massive amount of interesting data to comb through.

*****I want to be explicit here that the experiments performed are high quality and the data is exciting. The study of germlineless animals is warranted to focus on somatic roles for SET-26 and HCF-1, but statements like “major role in recruiting HCF-1 to chromatin in somatic cells, rather than in the germline” and “Contrarily, we find that HDA-1 binding to chromatin is more affected by loss of *set-26* in whole worm samples than in somatic-only samples” make interesting hypothesis, but require additional data; I recommend softening these statements in the discussion.

We thank the reviewer for pointing out the discrepancy in our experimental design and our phrasing for conclusions about the germline. We agree that analysis of isolated germlines would be required to make solid conclusions about the action of SET-26 and HCF-1 in the germline, and have therefore edited our discussion in the following places:

Original: “Indeed, we found that HCF-1 binding to chromatin is severely affected by loss of *set-26* when looking at somatic-only samples but only minorly affected in whole worm samples, leading us to conclude that SET-26 could play a major role in recruiting HCF-1 to chromatin in somatic cells, rather than in the germline.”

Revised: “Indeed, we found that HCF-1 binding to chromatin is severely affected by loss of *set-26* when looking at somatic-only samples but only minorly affected in germline-containing wildtype whole worm samples. One possible explanation for the difference between germline-containing and germline-less worms could be that SET-26 plays a more important role in recruiting HCF-1 in somatic cells than in the germline, however as we have not conducted any tissue-specific analysis of isolated germlines, we can only speculate about this possibility. It is tempting to speculate however, that SET-26 could have a tissue-specific role in recruiting HCF-1 only in somatic cells, or that another protein such as SET-9, a paralog of SET-26 which is only expressed in the germline and modulates reproduction but not lifespan¹⁵, could contribute to

HCF-1 recruitment in the germline. It will be interesting for future studies to investigate the role of SET-26 in recruiting HCF-1 in the germline and whether SET-9 plays a role.”

Original: “Contrarily, we find that HDA-1 binding to chromatin is more affected by loss of *set-26* in whole worm samples than in somatic-only samples, suggesting a possible role for germline SET-26 in mediating HDA-1 recruitment.”

Revised: “Contrarily, we find a more substantial defect in HDA-1 binding in wildtype whole worm germline-containing samples than in germline-less mutants. We speculate that this could be the result of a tissue-specific role for SET-26 in mediating HDA-1 recruitment in the germline and not in somatic cells, and it will be interesting for future studies to test this hypothesis.”

Original: “Our data suggest that SET-26 works primarily in the germline to affect HDA-1 recruitment, and is unlikely to be related to the longevity phenotype of the *set-26(-)* mutant.”

Revised: “The data raise the intriguing possibility that germline SET-26 could have a role in HDA-1 recruitment. However, given that SET-26 operates in somatic cells to regulate lifespan, our data suggest that any HDA-1 recruitment defect observed here is unlikely to be related to the longevity phenotype of the *set-26(-)* mutant.”

References

Ali, M. *et al.* Molecular basis for chromatin binding and regulation of MLL5. *Proc. Natl. Acad. Sci. U. S. A.* **110**, 11296–11301 (2013).

Amemiya, H. M., Kundaje, A. & Boyle, A. P. The ENCODE Blacklist: Identification of Problematic Regions of the Genome. *Sci. Rep.* **9**, 9354 (2019).

Baytek, G. *et al.* SUMOylation of the chromodomain factor MRG-1 in *C. elegans* affects chromatin-regulatory dynamics. *Biotechniques* **73**, 5–17 (2022).

Beurton, F. *et al.* Physical and functional interaction between SET1/COMPASS complex component CFP-1 and a Sin3S HDAC complex in *C. elegans*. *Nucleic Acids Res.* **47**, 11164–11180 (2019).

Chen, R. A.-J. *et al.* Extreme HOT regions are CpG-dense promoters in *C. elegans* and humans. *Genome Res.* **24**, 1138–1146 (2014).

Cui, M., Kim, E. B. & Han, M. Diverse chromatin remodeling genes antagonize the Rb-involved SynMuv pathways in *C. elegans*. *PLoS Genet.* **2**, e74 (2006).

Emerson, F. J. & Lee, S. S. CUT&RUN for Chromatin Profiling in *Caenorhabditis elegans*. *Curr Protoc* **2**, e445 (2022).

Ewald, C. Y., Landis, J. N., Porter Abate, J., Murphy, C. T. & Blackwell, T. K. Dauer-independent insulin/IGF-1-signalling implicates collagen remodelling in longevity. *Nature* **519**, 97–101 (2015).

Guang, S. *et al.* An Argonaute transports siRNAs from the cytoplasm to the nucleus. *Science* **321**, 537–541 (2008).

Jänes, J. *et al.* Chromatin accessibility dynamics across *C. elegans* development and ageing. *eLife* vol. 7 Preprint at <https://doi.org/10.7554/elife.37344> (2018).

Kamath, R. S., Martinez-Campos, M., Zipperlen, P., Fraser, A. G. & Ahringer, J. Effectiveness of specific RNA-mediated interference through ingested double-stranded RNA in *Caenorhabditis elegans*. *Genome Biol.* **2**, RESEARCH0002 (2001).

Li, J. *et al.* *Caenorhabditis elegans* HCF-1 functions in longevity maintenance as a DAF-16 regulator. *PLoS Biol.* **6**, e233 (2008).

Li, M. *et al.* SETD5 modulates homeostasis of hematopoietic stem cells by mediating RNA Polymerase II pausing in cooperation with HCF-1. *Leukemia* **36**, 1111–1122 (2022).

Li, M. *et al.* Structure, activity and function of the lysine methyltransferase SETD5. *Front. Endocrinol.* **14**, 1089527 (2023).

Meers, M. P., Tenenbaum, D. & Henikoff, S. Peak calling by Sparse Enrichment Analysis for CUT&RUN chromatin profiling. *Epigenetics Chromatin* **12**, 42 (2019).

Michaud, J. *et al.* HCFC1 is a common component of active human CpG-island promoters and coincides with ZNF143, THAP11, YY1, and GABP transcription factor occupancy. *Genome Res.* **23**, 907–916 (2013).

Minocha, S. *et al.* Rapid Recapitulation of Nonalcoholic Steatohepatitis upon Loss of Host Cell Factor 1 Function in Mouse Hepatocytes. *Mol. Cell. Biol.* **39**, (2019).

Morikiri, Y., Matsuta, E. & Inoue, H. The collagen-derived compound collagen tripeptide induces collagen expression and extends lifespan via a conserved p38 mitogen-activated protein kinase cascade. *Biochem. Biophys. Res. Commun.* **505**, 1168–1173 (2018).

Palani, S. N., Sellegounder, D., Wibisono, P. & Liu, Y. The longevity response to warm temperature is neurally controlled via the regulation of collagen genes. *Aging Cell* **22**, e13815 (2023).

Pu, M., Wang, M., Wang, W., Velayudhan, S. S. & Lee, S. S. Unique patterns of trimethylation of histone H3 lysine 4 are prone to changes during aging in *Caenorhabditis elegans* somatic cells. *PLoS Genet.* **14**, e1007466 (2018).

Rizki, G. et al. The evolutionarily conserved longevity determinants HCF-1 and SIR-2.1/SIRT1 collaborate to regulate DAF-16/FOXO. *PLoS Genet.* **7**, e1002235 (2011).

Shao, L.-W. *et al.* Histone deacetylase HDA-1 modulates mitochondrial stress response and longevity. *Nat. Commun.* **11**, 4639 (2020).

Timmons, L., Court, D. L. & Fire, A. Ingestion of bacterially expressed dsRNAs can produce specific and potent genetic interference in *Caenorhabditis elegans*. *Gene* **263**, 103–112 (2001).

Wang, W. *et al.* SET-9 and SET-26 are H3K4me3 readers and play critical roles in germline development and longevity. *Elife* **7**, (2018).

Wysocka, J., Myers, M. P., Laherty, C. D., Eisenman, R. N. & Herr, W. Human Sin3 deacetylase and trithorax-related Set1/Ash2 histone H3-K4 methyltransferase are tethered together selectively by the cell-proliferation factor HCF-1. *Genes Dev.* **17**, 896–911 (2003).

Zargar, Z. & Tyagi, S. Role of host cell factor-1 in cell cycle regulation. *Transcription* **3**, 187–192 (2012).

REVIEWER COMMENTS

Reviewer #1 (Remarks to the Author):

Although the authors made considerable efforts to address the points raised by this reviewer, two outstanding points remain concerning the physical interaction between SET-26 and HCF-1, and the genomic profiling of these two factors. Because of this, I feel this work does not provide sufficient novelty or insight on the action of these factors on chromatin to support the proposed model and warrant publication in this journal.

Major points

Physical interaction

This reviewer still feels that co-IP experiments are a minimal requirement to sustain a physical interaction between HCF-1 and SET-26 (direct or not). Co-IP is a routine experiment, especially if carried out on embryos (as shown by numerous publications). I do not challenge the conclusion that HCF-1 is found in several chromatin associated complexes, but this should be formally shown for publication in this journal, especially considering that the proposed model, as written in the abstract, is that SET-26 recruits HCF-1 to chromatin in somatic cells. Since the authors claim to provide mechanistic insight, this interaction should be validated. The authors now provide a list of HCF-1 interactors isolated in two previous independent co-IP, including SET-26. This could eventually replace a co-IP if it were shown that these are high confidence hits (number of unique peptides isolated vs control, as is standard practice).

Chromatin profiles:

The authors made a considerable effort to further analyze the cut & run data, and provide plausible explanations and clarification of the data. While this reviewer appreciates the effort and is now convinced of the quality of the cut&run experiments, which are quite challenging, the widespread binding of both HCF-1 and SET-26, that the authors now show often overlap with accessible regions bound by many other chromatin factors makes it difficult to make any conclusion as to the mechanism of action of these two chromatin associated factors, their relation to each other, and, more importantly, the specificity of their interaction on chromatin. With 13,422 and 9,961 peaks called for SET-26 and HCF-1, respectively, this means that at least half of the genes are likely to have binding of one or both of these factors in one or more tissues. In addition it is not surprising that of the 9,961 HCF-1 peaks 83% of them overlap with a promoter region, given the compactness of the *C. elegans* genome. Not surprisingly binding does not correlate to the transcription of the underlying gene.

As the authors point out "SET-26 and HCF-1 are both highly prevalent factors at active chromatin, and likely also interact with many additional factors". Unfortunately the present study does not provide any insight into how specificity is achieved. Many chromatin factors are found to colocalize when using whole animals, as clearly shown by the ModEncode tracts.

These experiments were carried out on entire worms, so the profiles represent an average of many different cell types. While I am convinced that SET-26 and HCF-1 are likely to bind at least some of the same genomic regions, the present data does not address the question of whether both proteins are found at the same genomic region in a given cell type.

the authors have revised the text to take into account the possibility that the effect of SET-26 depletion on HCF-1 recruitment to chromatin could be indirect ie a general effect on chromatin structure, or deregulation of one or more chromatin factors in the mutant

"Although we do see similar phenotypes for our mutants, we do agree with the reviewer that it is entirely possible that the effect of SET-26 depletion on HCF-1 recruitment to chromatin could be

indirect. To make this clearer in our text, we have changed the previous sentence in the discussion of "While we have not completely ruled out an indirect effect caused by loss of set26, it is tempting to speculate that SET-26 may directly recruit HCF-1 to chromatin" to the newly revised sentence, "Importantly, we have not ruled out an indirect effect caused by loss of set26, and it is entirely possible that loss of set-26 somehow alters the chromatin environment indirectly in a manner that prevents proper HCF-1 recruitment to chromatin. However, it is tempting to speculate that SET-26 may directly recruit HCF-1 to chromatin."

As already mentioned, in light of all the above considerations, this reviewer still feels that a model in which SET-26 recruits HCF-1 to chromatin in somatic cells to mediate transcriptional changes related to longevity, and that the histone deacetylase HDA-1 opposes their activity, as stated in the abstract, is not sufficiently supported by the present data.

Reviewer #2 (Remarks to the Author):

The original submission was well-founded and generated a wealth of new data to support the authors' model that SET-26 and HCF-1 oppose HDA-1 to regulate somatic gene expression during aging. The additions and clarifications to the text provide a more tempered approach to discussing their findings and significantly strengthen the paper.

Reviewer #3 (Remarks to the Author):

The authors have addressed my concerns. Congrats on a very nice manuscript and overall research study.

Response to Reviewers
Emerson et al.
10/4/2023

REVIEWER COMMENTS

Reviewer #1 (Remarks to the Author):

Although the authors made considerable efforts to address the points raised by this reviewer, two outstanding points remain concerning the physical interaction between SET-26 and HCF-1, and the genomic profiling of these two factors. Because of this, I feel this work does not provide sufficient novelty or insight on the action of these factors on chromatin to support the proposed model and warrant publication in this journal.

Major points

Physical interaction

This reviewer still feels that co-IP experiments are a minimal requirement to sustain a physical interaction between HCF-1 and SET-26 (direct or not). Co-IP is a routine experiment, especially if carried out on embryos (as shown by numerous publications). I do not challenge the conclusion that HCF-1 is found in several chromatin associated complexes, but this should be formally shown for publication in this journal, especially considering that the proposed model, as written in the abstract, is that SET-26 recruits HCF-1 to chromatin in somatic cells. Since the authors claim to provide mechanistic insight, this interaction should be validated. The authors now provide a list of HCF-1 interactors isolated in two previous independent co-IP, including SET-26. This could eventually replace a co-IP if it were shown that these are high confidence hits (number of unique peptides isolated vs control, as is standard practice).

To address the reviewer’s concern regarding the strength of the interaction between SET-26 and HCF-1, we have now added additional details of the IP-mass spec to Supplementary Data 1 to support the claim that SET-26 is a high confidence hit for an HCF-1-interactor. Specifically, both of the two independent IP-Mass spec experiments that were conducted included a negative control in which GFP was immunoprecipitated in an untagged (N2) background. Supplementary Data 1 shows the abundance of SET-26 and HDA-1 identified in either 1) HCF-1 IP or 2) control IP. The table is reproduced here for convenience. Both SET-26 and HDA-1 are identified specifically in the HCF-1 IP and not in the control purification in both independent experiments, supporting the idea that they are both high confidence hits.

		Experiment 1 - conducted by co-author C.G.R.		Experiment 2 - conducted by co-author M.Z.	
Protein Interactor	Wormbase ID	Number of unique peptides in HCF-1 IP	Number of unique peptides in control purification*	Spectral counts in HCF-1 IP	Spectral counts in control purification*
SET-26	WBGene00013106	14	0	46	0
HDA-1	WBGene00001834	2	0	2	0
* anti-GFP immunoprecipitation from untagged (N2) animals					

Chromatin profiles:

The authors made a considerable effort to further analyze the cut & run data, and provide plausible explanations and clarification of the data. While this reviewer appreciates the effort and is now convinced of the quality of the cut&run experiments, which are quite challenging, the widespread binding of both HCF-1 and SET-26, that the authors now show often overlap with accessible regions bound by many other chromatin factors makes it difficult to make any conclusion as to the mechanism of action of these two chromatin associated factors, their relation to each other, and, more importantly, the specificity of their interaction on chromatin. With 13,422 and 9,961 peaks called for SET-26 and HCF-1, respectively, this means that at least half of the genes are likely to have binding of one or both of these factors in one or more tissues. In addition it is not surprising that of the 9,961 HCF-1 peaks 83% of them overlap with a promoter region, given the compactness of the *C. elegans* genome. Not surprisingly binding does not correlate to the transcription of the underlying gene.

As the authors point out "SET-26 and HCF-1 are both highly prevalent factors at active chromatin, and likely also interact with many additional factors". Unfortunately the present study does not provide any insight into how specificity is achieved. Many chromatin factors are found to colocalize when using whole animals, as clearly shown by the ModEncode tracts. These experiments were carried out on entire worms, so the profiles represent an average of many different cell types. While I am convinced that SET-26 and HCF-1 are likely to bind at least some of the same genomic regions, the present data does not address the question of whether both proteins are found at the same genomic region in a given cell type.

the authors have revised the text to take into account the possibility that the effect of SET-26 depletion on HCF-1 recruitment to chromatin could be indirect ie a general effect on chromatin structure, or deregulation of one or more chromatin factors in the mutant

"Although we do see similar phenotypes for our mutants, we do agree with the reviewer that it is entirely possible that the effect of SET-26 depletion on HCF-1 recruitment to chromatin could be indirect. To make this clearer in our text, we have changed the previous sentence in the discussion of "While we have not completely ruled out an indirect effect caused by loss of set26, it is tempting to speculate that SET-26 may directly recruit HCF-1 to chromatin" to the newly revised sentence, " Importantly, we have not ruled out an indirect effect caused by loss of set26, and it is entirely possible that loss of set-26 somehow alters the chromatin environment indirectly in a manner that prevents proper HCF-1 recruitment to chromatin. However, it is tempting to speculate that SET-26 may directly recruit HCF-1 to chromatin."

As already mentioned, in light of all the above considerations, this reviewer still feels that a model in which SET-26 recruits HCF-1 to chromatin in somatic cells to mediate transcriptional changes related to longevity, and that the histone deacetylase HDA-1 opposes their activity, as stated in the abstract, is not sufficiently supported by the present data.

We appreciate the feedback from the reviewer and we agree that due to the use of whole worms, we have not directly shown that SET-26 and HCF-1 occupy the same binding sites in the same cell types. Importantly, this comment was made by Reviewer 2 in the initial review of our manuscript, and during the revision process we did add several sentences to the text in the discussion section to address this limitation as follows:

"Of note, our CUT&RUN approach utilized whole worms lacking germlines, thus our results are an aggregate of binding signal obtained from all surveyed somatic cell types. Therefore, our

current approach is not able to distinguish whether SET-26 and HCF-1 co-occupy the same promoters in the same cell type, or whether they occupy the same promoters independently in different cell types. Given the IP-Mass spec interaction data between SET-26 and HCF-1 and the requirement for SET-26 in HCF-1 binding, it seems more likely that SET-26 and HCF-1 co-occupy the same promoters together in the same cell types, however future approaches will be required to test this in a tissue-specific manner.”

As mentioned in the initial review, we agree with the reviewer that we have not definitively shown that SET-26 recruits HCF-1 to chromatin. However, based on our data (both IP-Mass Spec and CUT&RUN) and the high conservation of the interaction between homologs of SET-26 and HCF-1 in humans (including evidence that MLL5 recruits HCF1 to chromatin in Lee et al., 2020, PMID: [33050986](https://pubmed.ncbi.nlm.nih.gov/33050986/)), we still think a direct model involving an interaction between SET-26 and HCF-1 is the most likely explanation of our data. Therefore, we have not entirely removed mention of our model that SET-26 could recruit HCF-1 as we feel that the inclusion of this model adds more context and an important testable hypothesis for future work, but we have toned down the phrasing in several places to make it clear that this interpretation is a hypothesis based on our data. Specifically, we have changed the phrasing in the following places:

In the abstract, instead of the previous phrasing of: “We propose a model in which SET-26 recruits HCF-1 to chromatin in somatic cells, where they stabilize each other at a subset of genes, particularly mitochondrial and lipid metabolism genes, and regulate their expression”, we have now rephrased it to say, “HCF-1 localization at chromatin is largely dependent on functional SET-26, whereas SET-26 is only minorly affected by loss of HCF-1, suggesting that SET-26 could recruit HCF-1 to chromatin.”

At the end of the introduction, instead of the previous phrasing of: “Our data suggest that SET-26 plays a major role in recruiting HCF-1 to chromatin in *C. elegans* somatic cells, where, at a subset of binding sites, HCF-1 plays a minor role in stabilizing SET-26 binding as well”, we have now rephrased it to say, “Our data show that SET-26 is largely required for HCF-1 binding to chromatin in *C. elegans* somatic cells, whereas HCF-1 is dispensable for most, but not all, SET-26 binding to chromatin. We therefore hypothesize that SET-26 recruits HCF-1 to chromatin, and, at a subset of binding sites, we hypothesize that HCF-1 plays a minor role in stabilizing SET-26 binding as well.”

As pointed out by the reviewer, we had previously included updated the following sentence in the discussion after the first review to make this clear as follows: “Importantly, we have not ruled out an indirect effect caused by loss of *set-26*, and it is entirely possible that loss of *set-26* somehow alters the chromatin environment indirectly in a manner that prevents proper HCF-1 recruitment to chromatin. However, it is tempting to speculate that SET-26 may directly recruit HCF-1 to chromatin.”

With these updates, we hope it will be clear to readers throughout the paper that the model in which SET-26 recruits HCF-1 to chromatin is our working hypothesis and has not been definitively proven.

REVIEWER COMMENTS

Reviewer #1 (Remarks to the Author):

I appreciate the authors efforts, but rather than carrying out doable additional experiments, they have watered down their interpretation of the results, reflecting inherent weaknesses in the chromatin profiling and biochemical data.

Concerning the biochemistry 2 hits in a co-IP (for HDA-1) is next to background from my experience. These low hits have to be validated biochemically with available tools.

Response to Reviewer
Emerson et al.
12/8/2023

REVIEWER COMMENTS

Reviewer #1 (Remarks to the Author):

I appreciate the authors efforts, but rather than carrying out doable additional experiments, they have watered down their interpretation of the results, reflecting inherent weaknesses in the chromatin profiling and biochemical data.

The choice to revise our text arose directly from a comment we received from the Editor upon receipt of our previous reviewer comments, which suggested that we tone down our language. However, importantly, the conclusions and interpretations of our data from our manuscript did not change with our dampened language. Since our original revision, we have always agreed with the reviewer that we have not definitively shown that SET-26 recruits HCF-1 to chromatin, but that it remains the most likely model based on our data. We therefore maintained our model, and minorly edited our language to make this clear. Importantly, since our initial revision, we have always included the following sentence in our discussion section,

“Importantly, we have not ruled out an indirect effect caused by loss of *set-26*, and it is entirely possible that loss of *set-26* somehow alters the chromatin environment indirectly in a manner that prevents proper HCF-1 recruitment to chromatin. However, it is tempting to speculate that SET-26 may directly recruit HCF-1 to chromatin.”

This makes it clear that we have always agreed with the reviewer about the interpretation of our data, and our subsequent minor revisions of three sentences only serve to clarify this throughout the text.

Additionally, even if we carried out the suggested additional experiment of co-IPs between SET-26 and HCF-1 to further validate our already strong IP-Mass Spec experiments, this would not be able to definitively prove that SET-26 recruits HCF-1 to chromatin. Rather, it would further support that the proteins physically interact, but not specify where in the cell this occurs or in what manner each protein arrives at chromatin. As an experiment to definitively prove SET-26 recruits HCF-1 to chromatin is beyond the scope of this study, we felt it was appropriate to be transparent with readers as to what our data directly show and what we think the most likely explanation of those data are.

Concerning the biochemistry 2 hits in a co-IP (for HDA-1) is next to background from my experience. These low hits have to be validated biochemically with available tools.

We agree with the reviewer that the IP-Mass Spec data suggest only a weak possible interaction between HCF-1 and HDA-1. We used the IP-Mass Spec data as initiative to

pursue a functional relationship between HCF-1 and HDA-1, however based on the remainder of the data we collected, we have always proposed a model, which has remained unchanged from our initial submission, that HDA-1 is unlikely to have a strong physical interaction with HCF-1 at chromatin in our particular system.

Specifically, the main findings of our manuscript, which can be graphically summarized in our model figure (Fig. 7f, copied below), and which has not changed since the initial submission of this manuscript suggest: (i) *set-26* and *hcf-1* act in the same genetic pathway to limit lifespan, whereas *hda-1* antagonizes their role in longevity. (ii) SET-26 and HCF-1 co-occupy common promoters and SET-26 is required for HCF-1 recruitment to chromatin. (iii) HDA-1, while it localizes to many of the same promoters, gets to chromatin independently of SET-26 and HCF-1. (iv) SET-26 and HCF-1 regulate gene expression similarly on many common target genes, whereas HDA-1 appears to have the opposing effect on gene expression among those common targets.

f

As is evident from our original conclusions and model, we actually suggest that HDA-1 is nearby HCF-1 at chromatin, particularly in somatic cells which was the focus of our study. This is perhaps why it was picked up at a low level on the HCF-1 IP-Mass Spec, but it is unlikely to have a strong physical relationship with HCF-1 at the level of chromatin in our experimental condition. Therefore, the weak interacting data are

unsurprising and in no way affect our overall conclusions, which have not changed since the initial submission of this manuscript. We returned to the text of our manuscript to ensure that our reasoning and conclusions were clear throughout the text, and realized that we had a lead-in paragraph where we seemed to suggest HDA-1 could form a complex with HCF-1 and SET-26 based on the IP-MS data, even though that was not our conclusion at the end. Therefore, to make it clear to readers that the IP-MS data between HCF-1 and HDA-1 are weak, we have revised this paragraph as follows:

“HDA-1 is required for the full longevity of *set-26(-)* and *hcf-1(-)* mutants

Because SET-26 and HCF-1 homologs are well known to work with additional chromatin factors^{23,30,38,39}, we wondered whether we could identify additional protein factors that might work together with SET-26 and HCF-1 at chromatin in *C. elegans*. We noted that the histone deacetylase HDA-1, homolog of mammalian HDAC1 and HDAC2, was a possible interactor of HCF-1 based on the IP-Mass Spec data (Supplementary Data 1), even though the data suggested only a weak interaction. Homologs of SET-26 and HCF-1 in flies and humans, respectively, have been suggested to recruit HDAC1 homologs to chromatin^{16,40}, supporting a possible conserved functional interaction between these three proteins in various species.”

Apart from this single paragraph, we do not reference the IP-MS data of HCF-1 and HDA-1 interactions in the manuscript, and do not conclude that HCF-1 and HDA-1 form a physical interaction. Therefore, additional co-IP experiments would not even necessarily expect to find a strong physical interaction between HCF-1 and HDA-1. Importantly, the results, whether positive or negative, would not add to the conclusions or overall model of our paper which are based off of the genetics and genomics data that we collected and again, do not conclude a physical interaction between HCF-1 and HDA-1. We hope the reviewer would agree with our rationale; Considering the time and staff required to generate the appropriate *C. elegans* strains and optimize the protocol in our laboratory, and that the first author of the paper has now graduated and left the lab, it is neither feasible nor beneficial for us to conduct additional co-IP experiments.

However, to provide additional transparency with our IP-Mass Spec data, we have now submitted the data from both independent experiments to ProteomeXchange, and provided the relevant identifiers and reviewer token in the IP-Mass Spec methods section as follows:

“IP-Mass spec proteomics data from the two independent experiments have been deposited to the ProteomeXchange Consortium via the PRIDE partner repository with the dataset identifier PXD047509 (performed by co-author C.G.R.) and PXD047247 (performed by co-author M.Z., with reviewer tokens as follows: <https://repository.jpostdb.org/preview/2007299207656401fb90254>, access key 9270).”

We have also provided additional data within our manuscript for the IP-Mass Spec experiments. Specifically, we now include all hits from both independent IP-Mass Spec experiments which probed for HCF-1 interactors. We include the relative abundance of all hits obtained either from the experimental (IP of HCF-1) or control (IP of GFP in

wildtype worms) immunoprecipitations within Supplementary Data 1, so that readers are able to further explore and evaluate the data themselves.